# Local Convergence of Gradient Methods for Min-Max Games: Partial Curvature Generically Suffices

**Guillaume Wang**
Institute of Mathematics
École polytechnique fédérale de Lausanne
Station Z, CH-1015 Lausanne
guillaume.wang@epfl.ch

**Lénaïc Chizat**
Institute of Mathematics
École polytechnique fédérale de Lausanne
Station Z, CH-1015 Lausanne
lenaic.chizat@epfl.ch

## Abstract

We study the convergence to local Nash equilibria of gradient methods for two-player zero-sum differentiable games. It is well-known that such dynamics converge locally when $S \succ 0$ and may diverge when $S = 0$, where $S \succeq 0$ is the symmetric part of the Jacobian at equilibrium that accounts for the "potential" component of the game. We show that these dynamics also converge as soon as $S$ is nonzero (*partial curvature*) and the eigenvectors of the antisymmetric part $A$ are in general position with respect to the kernel of $S$. We then study the convergence rates when $S \ll A$ and prove that they typically depend on the *average* of the eigenvalues of $S$, instead of the minimum as an analogy with minimization problems would suggest. To illustrate our results, we consider the problem of computing mixed Nash equilibria of continuous games. We show that, thanks to partial curvature, conic particle methods – which optimize over both weights and supports of the mixed strategies – generically converge faster than fixed-support methods. For min-max games, it is thus beneficial to add degrees of freedom "with curvature": this can be interpreted as yet another benefit of over-parameterization.

## 1 Introduction

Min-max optimization is notoriously subtler than minimization, even in convex-concave settings. While many of the proof techniques for minimization have a natural equivalent in the min-max world, some common intuitions fail to transfer. The picture is clear for strongly convex-strongly concave (SC-SC) min-max games: all the classical gradient methods converge exponentially (for small enough step-sizes) with worst-case convergence rates dependent on the strong convexity and strong concavity parameters $\mu_x$, $\mu_y$. But, most famously perhaps, for bilinear min-max games the last iterate of simultaneous Gradient Descent-Ascent (GDA) diverges, while the Proximal Point (PP) method converges, and alternating GDA and the continuous-time limit of all of these algorithms – Gradient Flow (GF) – exhibit a cycling behavior.

Based on those two extreme cases, it still seems that part of our intuition from minimization, where the last-iterate convergence rate is indeed determined by the strong convexity parameter, is preserved. In this paper, we argue that this intuition is in fact overly pessimistic, and that gradient methods behave in general more favorably in the min-max setting than in the minimization setting.

Let us first look at a basic example. Consider a bilinear min-max game regularized by a quadratic term *only in one scalar variable*:

$$\min_{x \in \mathbb{R}^d} \max_{y \in \mathbb{R}^d} x^\top P y + \frac{\alpha}{2} x_1^2.$$

37th Conference on Neural Information Processing Systems (NeurIPS 2023).

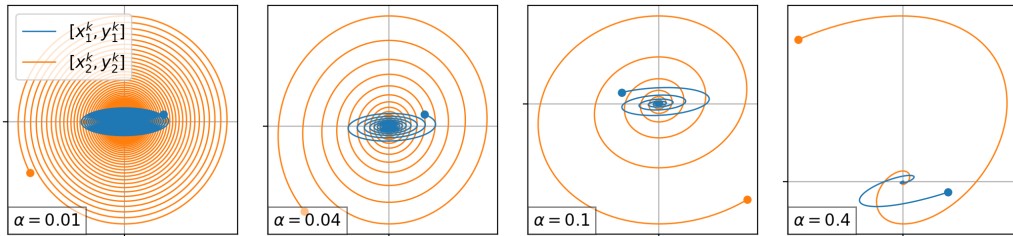

(a) Local convergence of the GF iterates for a fixed draw of $P \in \mathbb{R}^{2\times 2}$ and various values of $\alpha$. Only the final phase of the dynamics is shown, so here we see the iterates evolve along the "dominant" eigenspace of $M$ (the subspace along which convergence is the slowest).

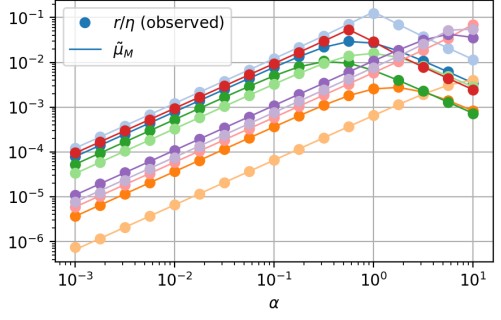

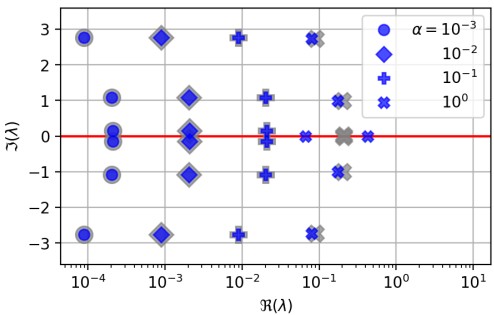

(b) Observed and predicted ($\tilde{\mu}_M$) normalized convergence rate $r/\eta$ of GDA with a small step-size $\eta$, i.e., $\|x^k\| + \|y^k\| = \Theta((1-r)^k)$, vs. regularization strength $\alpha$ (the higher the faster). Each color represents one draw of $P \in \mathbb{R}^{2\times 2}$.

(c) Spectrum of the Jacobian $M$ for a fixed draw of $P \in \mathbb{R}^{3\times 3}$ (green) and its approximation by Equation (3.1) (gray)

Figure 1: Convergence of gradient methods on a random bilinear game regularized by $\frac{\alpha}{2}x_1^2$ for small step-sizes. The fact that $\tilde{\mu}_M$, and so $r/\eta$, scale linearly with $\alpha$ (for small $\alpha$) is explained by Prop. 3.1.

Here there is no strong convexity in either player (so $\mu_x = \mu_y = 0$), yet as observed on Fig. 1a, Fig. 1b, when $P \in \mathbb{R}^{d\times d}$ is random with independent standard normal entries, GDA (with a small step-size) typically converges at an exponential rate that scales linearly with $\alpha$.[1]

As we will see, this phenomenon is a consequence of the existing theory for general smooth min-max games

$$\min_{x\in\mathbb{R}^n} \max_{y\in\mathbb{R}^m} f(x,y) \tag{1.1}$$

with a local Nash equilibrium (NE), or local saddle point, $z^* = (x^*, y^*)$. It is well-known from the dynamical systems literature that GF converges locally to $z^*$ if

$$\tilde{\mu}_M := \min_{\lambda\in\mathrm{Sp}(M)} \Re(\lambda) > 0, \qquad \text{with } M = \begin{bmatrix} \nabla^2_{xx}f & \nabla^2_{xy}f \\ -\nabla^2_{xy}f^\top & -\nabla^2_{yy}f \end{bmatrix}(z^*) \in \mathbb{R}^{(n+m)\times(n+m)}$$

the Jacobian of the skewed gradient field at $z^*$, and where $\mathrm{Sp}(\cdot)$ denotes the spectrum, i.e., the set of eigenvalues. For our starting basic example, this quantity[2] can be visualized in Fig. 1c. Moreover, $\tilde{\mu}_M$ also characterizes the convergence behavior of gradient methods in the leading order in the step-size $\eta$: they all converge to $z^*$ with the rate $\eta\tilde{\mu}_M + O(\eta^2)$, see the Appendix for a review.[3]

While the condition $\tilde{\mu}_M > 0$ is very general and tight, it is not obvious how to control or interpret it. Our purpose with this paper is to explore what this condition entails and to emphasize some of its surprising consequences.

---

[1] Julia code for the numerical experiments is available at https://github.com/guillaumew16/local_cvgce_minmax.

[2] In numerical analysis and stability theory, the quantity $-\tilde{\mu}_M$ is known as the *spectral abscissa* of $-M$.

[3] More precisely, gradient methods converge locally (for small enough step-sizes) if $\min_{\lambda\in\mathrm{Sp}(M)} \Re(\lambda) > 0$, and only if $\min_{\lambda\in\mathrm{Sp}(M)\setminus\{0\}} \Re(\lambda) \geq 0$. Throughout this paper we will assume for simplicity $M$ invertible so that those two quantities are equal.

The rest of the paper is organized as follows. In Sec. 2 we show that, generically, $\tilde{\mu}_M > 0$ as long as the problem has *partial curvature*, i.e. the diagonal blocks of $M$ are non-zero. In Sec. 3, we study more precisely the case of games where the "interaction" component dominates the "potential" component, i.e., when $\nabla^2_{xx} f(z^*), \nabla^2_{yy} f(z^*) \ll \nabla^2_{xy} f(z^*)$. For such games, in a certain random setting, we show that $\tilde{\mu}_M$ scales as the *average* of the eigenvalues of the potential part, instead of the minimum as an analogy with minimization problems would suggest. In Sec. 4 we consider the computation of mixed Nash equilibria of continuous games using particle methods, a setting where the convergence under partial curvature has a striking consequence. By optimizing over both the weights and supports of the mixed strategies, one obtains dynamics that generically converge faster than fixed-support methods, even when the latter are using the optimal supports.

For ease of exposition, in most of the paper we focus on the local convergence of GF. We discuss the convergence of discrete time algorithms in Sec. 3.2.

## 1.1 Related work

Throughout this paper, by "convergence" we mean convergence of the last iterate to a (local) NE, while a different line of work also considers convergence of the averaged iterate [Nem04], and another considers convergence to any critical point [ALW21]. Also to prevent possible confusion, let us mention a related but distinct line of work [DH19; Doa22] that considers using different step-sizes to update the $x$ and the $y$ variables. The resulting two-timescale dynamics can be analyzed globally assuming a min-max analog of the Polyak-Łojasiewicz inequality, using quite different considerations than the ones developed in this paper.

**Analysis of GDA and PP for SC-SC, bilinear or convex-SC games.** The fact that PP converges for SC-SC and bilinear min-max games is well-established since at least [Roc76]; for a modern reformulation, see e.g. [MOP20]. The convergence of simultaneous GDA for SC-SC games and its divergence for bilinear games is also a classical fact, see e.g. [LS19; Lu22]. The cycling behavior of alternating GDA for bilinear games is proved in [BGP20, Theorem 4].

As shown more recently in [NK17, Appendix G] and [ZWLG22, Theorem 6], convexity in $x$, strong concavity in $y$ and non-degeneracy of the interaction component are in fact sufficient to ensure $\tilde{\mu}_M > 0$, and so local convergence of GDA. The setting of the former work corresponds to $\nabla^2_{xx} f(z^*) = 0$, $-\nabla^2_{yy} f(z^*) \succ 0$ and $\nabla^2_{xy} f(z^*)$ full-row-rank, extended in the latter work to $\nabla^2_{xx} f(z^*) \succeq 0$. Those two works also provide bounds on $\tilde{\mu}_M$ for those specific cases in terms of the least eigenvalues of $-\nabla^2_{yy} f(z^*)$ and of $\nabla^2_{xy} f(z^*) \nabla^2_{xy} f(z^*)^\top$.

**Spectral analysis-based convergence analyses.** Our approach to analyzing min-max algorithms is to directly study the properties of the update operator (e.g., of $T(z) = z - \begin{pmatrix} \nabla_x f \\ -\nabla_y f \end{pmatrix}(z)$ for simultaneous GDA). Specifically, by a classical result on discrete-time dynamical systems, the local exponential convergence is characterized by the spectral radius of that operator's Jacobian. This approach is used for example in [Gid+19] to analyze the local convergence behavior of alternating GDA with negative momentum, and in [AMLG20] to derive tight convergence bounds – dependent explicitly on $\eta$ and $\text{Sp}(M)$ – for simultaneous GDA and multi-step Extra-Gradient.

**Average-case analysis of the (local) convergence rate.** In [PS20; DPS21], the authors analyze the convergence of gradient methods for affine operator problems – i.e., for finding $z$ such that $M(z - z^*) = 0$ – when $M$ is a random normal matrix with a known spectral distribution. They derive average-case optimal methods for this setting. Our analysis in Sec. 3 also has an average-case flavor, but our random model is different: we assume that the symmetric $S$ and antisymmetric parts $A$ of $M$ are independent and that $S \ll A$. In particular we do not require $M$ to be normal (i.e., $S$ and $A$ do not commute a priori, which is the typical case in min-max optimization).

**Hypocoercivity.** In the context of partial differential equations (PDEs), the phenomenon that a linear PDE $\partial_t u_t = -L u_t$ may exhibit linear convergence to $0$ even when the generator $L$ is not coercive, is called hypocoercivity and is studied in detail in [Vil09]. This is precisely the infinite-dimensional analog of the phenomenon studied in the present paper: still denoting by $S, A$ the symmetric resp. antisymmetric parts of $M$, coercivity of $L$ corresponds to $S \succ 0$, while hypocoercivity corresponds to

$\tilde{\mu}_M > 0$. Specifically, [Vil09] shows how to construct Lyapunov functions to establish hypocoercivity and convergence rates of certain PDEs, by exploiting properties of the iterated commutators of $S$ and $A$. By contrast our focus is on the finite-dimensional case, where it is easier and more natural to directly study the spectrum of $M$.

## 1.2 Notation

For any matrix $T \in \mathbb{R}^{d \times d}$ or $\mathbb{C}^{d \times d}$, denote by $\mathrm{Sp}(T) \subset \mathbb{C}$ its spectrum, i.e., the set of its eigenvalues, and by $\rho(T) = \max_{\lambda \in \mathrm{Sp}(T)} |\lambda|$ its spectral radius. Recall that the spectral radius is distinct from the operator norm, which is the largest singular value, although they coincide for Hermitian matrices. Denote eigenspaces as $E_\lambda(T) = \{z \in \mathbb{C}^d;\ Tz = \lambda z\}$ and let $\mathrm{Eigvecs}(T) = \bigcup_{\lambda \in \mathrm{Sp}(T)} E_\lambda(T)$ the set of all (complex) eigenvectors. $\|\cdot\|$ denotes Euclidean or Hermitian norm. For a collection of square matrices (resp. scalars) $(C_k)_k$, $\mathrm{Diag}((C_k))$ denotes the (block-)diagonal matrix with blocks (resp. coefficients) $(C_k)_k$.

## 2 Characterization of local convergence

Consider the general smooth min-max game of Equation (1.1) and assume $M$ is invertible. We decompose $M$ into its symmetric and antisymmetric parts as

$$M = \begin{bmatrix} \nabla_{xx}^2 f & \nabla_{xy}^2 f \\ -\nabla_{xy}^2 f^\top & -\nabla_{yy}^2 f \end{bmatrix}(z^*) =: \begin{bmatrix} Q & P \\ -P^\top & R \end{bmatrix}, \qquad S := \begin{bmatrix} Q & 0 \\ 0^\top & R \end{bmatrix}, \qquad A := \begin{bmatrix} 0 & P \\ -P^\top & 0 \end{bmatrix}.$$

By the second-order optimality condition in the definition of NE, $Q \succeq 0$ and $R \succeq 0$.

Following [Let+19], $S$ and $A$ can be thought of intuitively as the "potential" resp. "interaction" (or Hamiltonian) components of the two-player zero-sum game (1.1). Indeed, consider the quadratic

game $\quad \min_{x \in \mathbb{R}^n} \max_{y \in \mathbb{R}^m} \left\{ \frac{1}{2} \begin{pmatrix} x \\ y \end{pmatrix}^\top \nabla^2 f(z^*) \begin{pmatrix} x \\ y \end{pmatrix} = \frac{1}{2} x^\top Q x - \frac{1}{2} y^\top R y + x^\top P y \right\}$, which is

essentially sufficient to understand the local behavior of GF for (1.1) around $z^*$. Then we can interpret the terms in $Q$ and $R$ as quadratic potentials to be optimized independently by each player, with the bilinear term in $P$ capturing all of the interaction between the players.

**Partial curvature generically suffices.** Let us recall that $\tilde{\mu}_M = \min_{\lambda \in \mathrm{Sp}(M)} \Re(\lambda)$ governs the local convergence of GF around the local NE $z^*$. It is a general fact that this quantity is nonnegative (the proof is included below). In the following theorem, we give necessary and sufficient conditions for it to be positive, in terms of geometric conditions on $Q$, $R$ and $P$.

**Theorem 2.1.** *The following conditions are equivalent:*

$(i)$ $\tilde{\mu}_M > 0$.

$(ii)$ $\mathrm{Eigvecs}(A) \cap \mathrm{Ker}\, S = \{0\}$.

$(iii)$ *For any eigenvector $x$ of $PP^\top$ (i.e., left-singular vector of $P$), $x \notin \mathrm{Ker}\, Q$ or $P^\top x \notin \mathrm{Ker}\, R$.*

$(iv)$ *For any eigenvector $y$ of $P^\top P$ (i.e., right-singular vector of $P$), $Py \notin \mathrm{Ker}\, Q$ or $y \notin \mathrm{Ker}\, R$.*

As a consequence of Thm. 2.1, the condition $\tilde{\mu}_M > 0$ holds generically in the following sense: for any fixed $S \neq 0$, the set of matrices $P$ such that $\tilde{\mu}_M > 0$ is dense and open in $\mathbb{R}^{n \times m}$. In particular, this property holds with probability 1 if $P$ is drawn from an absolutely continuous distribution and is independent from $Q$ and $R$, as in the experiment of Fig. 1.

*Proof.* Let $\lambda \in \mathrm{Sp}(M)$ and $z \in \mathbb{C}^{n+m}$ non-zero such that $Mz = (S + A)z = \lambda z$. Since $S$ and $A$ are real and symmetric resp. antisymmetric,

$$\overline{\bar{z}^\top S z} = z^\top S \bar{z} = (z^\top S \bar{z})^\top = \bar{z}^\top S z, \ \text{ so } \ \bar{z}^\top S z \in \mathbb{R}$$

$$\text{and } \ \overline{\bar{z}^\top A z} = z^\top A \bar{z} = (z^\top A \bar{z})^\top = -\bar{z}^\top A z, \ \text{ so } \ \bar{z}^\top A z \in i\mathbb{R}.$$

So by taking the real part in $\bar{z}^\top (S + A) z = \lambda \|z\|^2$,

$$\Re(\lambda) \|z\|^2 = \bar{z}^\top S z = \Re(z)^\top S \Re(z) + \Im(z)^\top S \Im(z) \geq 0$$

since $S = \mathrm{Diag}(Q, R) \succeq 0$. This shows that $\tilde{\mu}_M \geq 0$.

Now let us show the equivalence of the conditions.

$(ii) \implies (i)$: By contraposition, suppose there exists $\lambda \in \mathrm{Sp}(M)$ with $\Re(\lambda) = 0$, and let $z$ non-zero such that $Mz = \lambda z$. Then $\overline{z}^\top S z = \Re(\lambda) \|z\|^2 = 0$, so $z \in \mathrm{Ker}\, S$. So $Mz = Az = \lambda z$, and $z \in (E_\lambda(A) \cap \mathrm{Ker}\, S) \setminus \{0\}$.

$(i) \implies (ii)$: By contraposition, suppose there exists $\lambda \in \mathrm{Sp}(A)$ and a non-zero $z \in E_\lambda(A) \cap \mathrm{Ker}\, S$. Since $A$ is antisymmetric, then $\Re(\lambda) = 0$. On the other hand, $Mz = (S + A)z = Az = \lambda z$, i.e., $\lambda \in \mathrm{Sp}(M)$. So $\tilde{\mu}_M \leq \Re(\lambda) = 0$.

$(i) \implies (iii), (iv)$: By contraposition, suppose there exists an eigenvector $x$ of $PP^\top$ such that $x \in \mathrm{Ker}\, Q$ and $P^\top x \in \mathrm{Ker}\, R$, and denote $\sigma \in \mathbb{R}$ such that $PP^\top x = \sigma^2 x$ (since $PP^\top \succeq 0$). Then

$$M \begin{pmatrix} i\sigma\, x \\ P^\top x \end{pmatrix} = \begin{bmatrix} Q & P \\ -P^\top & R \end{bmatrix} \begin{pmatrix} i\sigma\, x \\ P^\top x \end{pmatrix} = \begin{pmatrix} \sigma^2 x \\ -i\sigma P^\top x \end{pmatrix} = -i\sigma \begin{pmatrix} i\sigma\, x \\ P^\top x \end{pmatrix}$$

and so $-i\sigma \in \mathrm{Sp}(M)$. This shows $(i) \implies (iii)$, and $(i) \implies (iv)$ follows analogously.

$(iii), (iv) \implies (ii)$: By contraposition, suppose there exists $\lambda = i\sigma \in \mathrm{Sp}(A)$ and a non-zero $z = (x, y) \in E_\lambda(A) \cap \mathrm{Ker}\, S$. Expanding the blocks in $Az = \lambda z$,

$$\begin{cases} Py = i\sigma x \\ -P^\top x = i\sigma y \end{cases} \quad \text{and so} \quad \begin{cases} P^\top P y = \sigma^2 y \\ PP^\top x = \sigma^2 x. \end{cases}$$

Moreover, this implies that $x = 0 \iff y = 0$, and since $z \neq 0$, then both $x \neq 0$ and $y \neq 0$. So $x$ is an eigenvector of $PP^\top$ and, since $Sz = 0$, $x \in \mathrm{Ker}\, Q$ and $P^\top x = -i\sigma y \in \mathrm{Ker}\, R$, which contradicts $(iii)$. Likewise, $y$ is an eigenvector of $P^\top P$ and $y \in \mathrm{Ker}\, R$ and $Py = i\sigma x \in \mathrm{Ker}\, Q$, which contradicts $(iv)$. $\qquad\square$

**Geometric interpretation using real vectors.** We draw the attention of the reader to the fact that $(ii)$ involves complex eigenvectors. For a rephrasing in terms of real objects, note that if $z \in E_{i\sigma}(A)$ with $\sigma \in \mathbb{R}$, then $A\,[\Re(z) \quad \Im(z)] = [\Re(z) \quad \Im(z)] \begin{bmatrix} 0 & \sigma \\ -\sigma & 0 \end{bmatrix}$; geometrically, $F_{i\sigma}(A) = \mathrm{span}(\Re(z), \Im(z))$ is a "rotation plane" of the GF for the bilinear game $\min_x \max_y x^\top P y$, in the sense that the projection of GF on $F_{i\sigma}(A)$ is a circular motion with constant speed $\sigma$. Condition $(ii)$ expresses that for each such $F_{i\sigma}(A)$, there exists an eigenspace $E_\mu(S)$ of $S$ (for a $\mu > 0$) that is not orthogonal to it. This causes the GF for $\min_x \max_y \frac{1}{2} x^\top Q x - \frac{1}{2} y^\top R y + x^\top P y$ projected on $F_{i\sigma}(A)$ to spiral down to 0 instead of cycling around it.

One may naturally wonder whether a notion of non-orthogonality between the potential $(S)$ and interaction components $(A)$ can be used to bound the convergence quantitatively; this is developed in the next section in the particular case $S \ll A$.

## 3 Convergence rate when interaction dominates ($S \ll A$)

Let us now discuss the case of games with a small symmetric part, that is, whose Jacobian at optimum is $M_\alpha = A + \alpha S$ for some symmetric $S = \begin{bmatrix} Q & 0 \\ 0^\top & R \end{bmatrix}$, antisymmetric $A = \begin{bmatrix} 0 & P \\ -P^\top & 0 \end{bmatrix}$ and some small $\alpha > 0$. In this section we assume $n = m$ (the general case is technically more challenging as Prop. 3.1 requires $A$ to have distinct eigenvalues, which requires $|n - m| \leq 1$).

### 3.1 Convergence rate of Gradient Flow

As discussed previously, the normalized local exponential convergence rate $r/\eta$ of gradient methods in the asymptotic regime $\eta \to 0$ – or equivalently, the convergence rate of GF – for a game with Jacobian at optimum $M_\alpha$ is equal to $\tilde{\mu}_{M_\alpha}$. We can estimate this quantity using the standard formula for the asymptotic expansion of the eigenvalues of a perturbed matrix, which takes an interesting form in our context.

**Proposition 3.1.** *Suppose that $P$ is full-rank and has distinct singular values, and let $P = U\Sigma V^\top = \sum_{j=1}^n \sigma_j u_j v_j^\top$ be its singular value decomposition. Then it holds*

$$\tilde{\mu}_{M_\alpha} = \frac{1}{2}\alpha\left(\min_{1\leq j\leq n} u_j^\top Q u_j + v_j^\top R v_j\right) + O(\alpha^3).$$

This expansion explains in particular why the normalized convergence rate $r/\eta \sim \tilde{\mu}_{M_\alpha}$ is approximately proportional to $\alpha$ in Fig. 1. Interestingly, the error term is $O(\alpha^3)$, which suggests that the approximation can be reasonably accurate even for quite large values of $\alpha$, as illustrated in Fig. 1c.

*Proof.* Here $M_0 = A$ has distinct eigenvalues $\{is\sigma_j, s \in \{-1, 1\}, 1 \leq j \leq n\}$, with unit-norm eigenvectors $A\begin{pmatrix} -isu_j/\sqrt{2} \\ v_j/\sqrt{2} \end{pmatrix} = is\sigma_j \begin{pmatrix} -isu_j/\sqrt{2} \\ v_j/\sqrt{2} \end{pmatrix}$. By the calculation of the eigenvalue derivatives from [Tao08], we obtain the following expansion for $\mathrm{Sp}(M_\alpha)$:

$$\mathrm{Sp}(M_\alpha) = \left\{ is\sigma_j + \frac{1}{2}\alpha\left(u_j^\top Q u_j + v_j^\top R v_j\right) + \frac{1}{4}\alpha^2 \sum_{(s',j')\neq(s,j)} \frac{1}{is\sigma_j - is'\sigma_{j'}} \left(ss' u_{j'}^\top Q u_j + v_{j'}^\top R v_j\right)^2 \right.$$

$$\left. + O(\alpha^3), \quad s \in \{-1,1\}, 1 \leq j \leq n \right\}. \quad (3.1)$$

Now the zeroth- and second-order terms are all in $i\mathbb{R}$, hence the announced expansion for $\tilde{\mu}_{M_\alpha}$. $\square$

**Estimate of the leading term under a probabilistic model.** Assuming the singular vectors $(u_1, ..., u_n)$, $(v_1, ..., v_n)$ of $P$ are distributed uniformly at random – which is the case for example if $P$ has i.i.d. Gaussian entries by rotational invariance[4] –, the leading term in the expansion of $\tilde{\mu}_{M_\alpha}$ can be estimated in expectation as follows. The proof is placed in the Appendix, where we also include a high-probability version of the estimate.

**Proposition 3.2.** *Suppose $Q, R$ are fixed and $U, V$ are independently distributed uniformly on the set of $n \times n$ orthonormal matrices. Then*

$$\frac{\mathrm{Tr}(S)}{n}\left(1 - 2\frac{\|S\|_F}{\mathrm{Tr}(S)}\sqrt{\log n}\right) \leq \mathbb{E}\left[\min_{1\leq j\leq n} u_j^\top Q u_j + v_j^\top R v_j\right] \leq \frac{\mathrm{Tr}(S)}{n}$$

*where $\|\cdot\|_F$ denotes the Frobenius norm. In particular, provided that $\frac{\mathrm{Tr}(S)}{\|S\|_F} \geq 2\sqrt{\log n}(1 + c)$ for some fixed $c > 0$, we have $\mathbb{E}\left[\min_{1\leq j\leq n} u_j^\top Q u_j + v_j^\top R v_j\right] \asymp \frac{\mathrm{Tr}(S)}{n}$ as $n \to \infty$.*

Note that $\frac{\mathrm{Tr}(S)}{\|S\|_F}$, which always lies in the interval $[1, \sqrt{2n}]$, is a measure of the effective sparsity of the spectrum of $S$ (larger meaning less sparse), so the condition $\frac{\mathrm{Tr}(S)}{\|S\|_F} \geq 2\sqrt{\log n}(1 + c)$ means the spectrum of $S$ is well spread-out. So the proposition shows that **the exponential convergence rate depends on the *average* of the eigenvalues of $S$,** when $\alpha S \ll A$ and the spectrum of $S$ is well spread-out. This fact should be contrasted with the case of minimization, where the convergence rate scales as the *minimum* eigenvalue of the Hessian.

Interestingly, when the spectrum of $S$ is sparse, the typical behavior of the leading term in the expansion of $\tilde{\mu}_{M_\alpha}$, $\min_j u_j^\top Q u_j + v_j^\top R v_j$, is quite different. In this case, that quantity depends on the geometric mean of the non-zero eigenvalues of $S$, rather than the arithmetic mean, as formalized in the following proposition. The proof is placed in the Appendix, along with a high-probability version of the estimate.

**Proposition 3.3.** *Suppose $Q, R$ are fixed and $U, V$ are independently distributed uniformly on the set of $n \times n$ orthonormal matrices. Let $s_1 \geq ... \geq s_r > 0 = s_{r+1} = ... = s_{2n}$ the eigenvalues of $S$.*

---

[4]If $P = U\Sigma V^\top$ has i.i.d. Gaussian entries, then $P$ has the same law as $\widetilde{U}^\top P \widetilde{V}$, for any $\widetilde{U}, \widetilde{V} \in \mathcal{O}_n$ the set of $n \times n$ orthonormal matrices. So $(U, V)$ has the same law as $(\widetilde{U}^\top U, \widetilde{V}^\top V)$. This shows that $(U, V)$ is distributed according to the Haar measure on the product group $\mathcal{O}_n \times \mathcal{O}_n$, that is, $U$ and $V$ are independently distributed uniformly on $\mathcal{O}_n$.

Table 1: Expansions of $\rho(\nabla T(z^*))$ in $\alpha$ and $\eta$ for classical gradient methods

| Algorithm | $T(z)$ | $\nabla T(z^*)$ | $\rho(\nabla T(z^*))^2 = \max_{j \leq 2n}[...]$ $+ O(\eta\alpha^3 + \eta^2\alpha^2)$ |
|---|---|---|---|
| Sim-GDA | $z - \eta g(z)$ | $I - \eta M$ | $1 - 2\alpha\eta \left(\overline{w}_j^\top S w_j\right) + \eta^2\sigma_j^2$ |
| Alt-GDA | see text | $I - \eta(I - \frac{\eta}{2}A)M + O(\eta^3)$ | $1 - 2\alpha\eta \left(\overline{w}_j^\top S w_j\right) + O(\eta^3)$ |
| EG | $z - \eta g(z - \eta g(z))$ | $I - \eta(I - \eta M)M$ | $1 - 2\alpha\eta \left(\overline{w}_j^\top S w_j\right) - \eta^2\sigma_j^2 + O(\eta^3)$ |

*Then*

$$\mathbb{E}\left[\min_{j \leq n} u_j^\top Q u_j + v_j^\top R v_j\right] \geq \max_{\mathcal{S} \subset \{1,\ldots,r\}} \frac{1}{e} \frac{|\mathcal{S}|}{n} n^{-\frac{2}{|\mathcal{S}|}} \left[\prod_{l \in \mathcal{S}} s_l\right]^{\frac{1}{|\mathcal{S}|}}.$$

*In particular,* $\mathbb{E}\left[\min_{j \leq n} u_j^\top Q u_j + v_j^\top R v_j\right] \gtrsim n^{-\frac{2}{r}-1}$ *when* $n \to \infty$ *and* $r$ *and* $s \in \mathbb{R}^r$ *are fixed.*

Numerically, the quantity $\min_{j \leq n} u_j^\top Q u_j + v_j^\top R v_j$ indeed scales as $n^{-\frac{2}{r}-1}$ under the conditions of the proposition, suggesting that our lower estimate could be tight.

## 3.2 Convergence rate of discrete-time algorithms

Prop. 3.1 gave an expansion of the normalized convergence rate $r/\eta$ of gradient methods (for a game with Jacobian at optimum $M_\alpha$), non-asymptotically in $\alpha$ and in the asymptotic limit $\eta \to 0$. In this subsection, we give expansions of the convergence rate $r$ that are non-asymptotic in $\alpha$ and $\eta$.

The algorithms we will consider can be written in the form $z^{k+1} = T(z^k)$ with the update operator $T$ dependent only on $\nabla f$ and on step-size $\eta$, and satisfying $z^* = T(z^*)$. It is well-known that local convergence of such methods is determined by $\rho(\nabla T(z^*))$, where $\nabla T$ is the Jacobian of $T$ and $\rho(\cdot)$ denotes spectral radius. Namely, if $\rho(\nabla T(z^*)) < 1$ then the iterates converge locally with $\|z^k - z^*\| = O\left((\rho(\nabla T(z^*)) + \varepsilon)^k\right)$, with $\varepsilon > 0$ an arbitrarily small constant [Ber97, Proposition 4.4.1].

In Table 1 we give an expansion for $\rho(\nabla T(z^*))$ for three classical gradient methods: simultaneous GDA (Sim-GDA), alternating GDA (Alt-GDA), and Extra-Gradient (EG). Let us clarify immediately that throughout this paper, statements made about "GDA" without further specification apply to both Sim-GDA and Alt-GDA. In the second column we denoted by $g(z) = \begin{pmatrix} \nabla_x f \\ -\nabla_y f \end{pmatrix}(z)$ the skewed gradient field of the game, and in the third column we wrote $M$ instead of $M_\alpha$ for concision. In the fourth column, $\pm i\sigma_j$ denotes the eigenvalues of $A$ assumed distinct and $w_j$ are the associated eigenvectors – equivalently, the singular value decomposition of $P$ is $P = \sum_{j=1}^n \sigma_j u_j v_j^\top$ and for each $j \leq n$, $\overline{w}_j^\top S w_j = \overline{w}_{j+n}^\top S w_{j+n} = \frac{1}{2}\left(u_j^\top Q u_j + v_j^\top R v_j\right)$. We refer to the Appendix for the derivation of this table and for explicit bounds on the "$O(\cdot)$" terms.

Informally, as one can directly see from the fourth column, Sim-GDA requires a very small step-size for the first term in $\eta$ to overcome the terms $+\eta^2\sigma_j^2$, while for EG those terms actually appear with a favorable sign, and Alt-GDA neither benefits nor suffers from those terms. We also see that Alt-GDA is quite faithful to GF, in that their normalized convergence rates coincide up to terms of order $\eta^3 + \alpha^4$. All of these insights are in line with common intuition in the min-max optimization literature [BGP20; Lu22], as well as with our numerical experiments for the next section, Fig. 2.

**A symmetrized formulation of Alt-GDA.** In order to derive the rate for Alt-GDA, we used the following symmetrized formulation of it: we let $(x^0, y^{1/2}) \in \mathbb{R}^d \times \mathbb{R}^d$ and

$$\begin{cases} \forall k \in \mathbb{N}, & x^{k+1} = x^k - \eta\nabla_x f(x^k, y^{k+1/2}) \\ \forall k \in \mathbb{N} + 1/2, & y^{k+1} = y^k + \eta\nabla_y f(x^{k+1/2}, y^k) \end{cases} \quad \text{and} \quad \begin{cases} \forall k \in \mathbb{N}, & x^{k+1/2} = \frac{x^{k+1}+x^k}{2} \\ \forall k \in \mathbb{N} + 1/2, & y^{k+1/2} = \frac{y^{k+1}+y^k}{2}. \end{cases}$$

That is, $x$ gets updated with the gradient rule at integer time-steps, $y$ at half-integer time-steps, and we define $x^k, y^k$ at non-updating time-steps as the average of the preceding and following updating time-steps. Assuming $\eta \leq \|\nabla_{xx}^2 f\|_\infty^{-1} \wedge \|\nabla_{yy}^2 f\|_\infty^{-1}$, we show in the Appendix that $z^{k+1}$ is indeed

entirely determined by $z^k = (x^k, y^k)$ for each $k \in \mathbb{N}$, and that the associated update operator $T$ satisfies

$$T(z) = z - \eta g(z) + \frac{\eta^2}{2} A(z) g(z) + O(\eta^3 \|g(z)\|) \quad \text{where} \quad A(z) = \frac{\nabla g(z) - \nabla g(z)^\top}{2}.$$

For comparison, the usual formulation of Alt-GDA considers as the iterates $(\tilde{x}^k, \tilde{y}^k) = (x^k, y^{k+1/2})$. We emphasize that $(\tilde{x}^k, \tilde{y}^k)$ and $(x^k, y^k)$ have the same convergence rate if they converge exponentially, as one can check directly from the definition.

## 4 Illustration: sparse mixed Nash equilibria of continuous games

In this section we apply the above considerations to a particular class of min-max problems, which is of its own interest in game theory. Namely we consider the classical problem of finding the mixed Nash equilibria (MNE) of two-player zero-sum games, that is, given strategy spaces $\mathcal{X}, \mathcal{Y}$ and a payoff function $f : \mathcal{X} \times \mathcal{Y} \to \mathbb{R}$, solving the min-max problem

$$\min_{\mu \in \mathcal{P}(\mathcal{X})} \max_{\nu \in \mathcal{P}(\mathcal{Y})} \{ F(\mu, \nu) = \mathbb{E}_{x \sim \mu, y \sim \nu}[f(x, y)] \}.$$

Here $\mathcal{P}(\mathcal{X})$ denotes the space of probability measures – representing mixed strategies – over $\mathcal{X}$. Let us focus on cases where $\mathcal{X}$ and $\mathcal{Y}$ are continuous sets, say, $\mathcal{X} = \mathcal{Y} = \mathbb{T}^1$ the one-dimensional Euclidean torus,[5] and $f : \mathbb{T}^1 \times \mathbb{T}^1 \to \mathbb{R}$ is smooth. The above min-max problem is then infinite-dimensional, and algorithms to solve it explicitly must be based on reparameterized formulations. More specifically, suppose that the MNE $(\mu^*, \nu^*)$ is unique and "sparse", i.e., has finite supports: $\text{supp}(\mu^*) = \{x_I^*, 1 \le I \le N\}$, $\text{supp}(\nu^*) = \{y_J^*, 1 \le J \le M\}$ (this is the case for example if $f$ is a sum of separable functions [SOP08]).

In this setting there are two natural reparameterizations and associated algorithms:

1. If the optimal support points $\{x_I^*\}_I$, $\{y_J^*\}_J$ are known, then we may reparameterize by $\mu = \sum_{I=1}^N a_I \delta_{x_I^*}$, $\nu = \sum_{J=1}^M b_J \delta_{y_J^*}$ and optimize over the $a_I, b_J$. The problem reduces to a constrained bilinear game

$$\min_{a \in \Delta_N} \max_{b \in \Delta_M} \left\{ F_1(a, b) = \sum_{I=1}^N \sum_{J=1}^M a_I b_J \, f(x_I^*, y_J^*) =: a^\top P b \right\}$$

   where $\Delta_N$ denotes the standard simplex. A classical approach is then to apply Mirror Prox (MP) with entropy link function [Nem04] (MP is the Bregman-geometry analog of EG).

2. If only the number of optimal support points is known, then we may reparameterize by $\mu = \sum_{I=1}^N a_I \delta_{x_I}$, $\nu = \sum_{J=1}^M b_J \delta_{y_J}$ and optimize over both the weights $(a_I, b_J)$ and the support points $(x_I, y_J)$. The problem reduces to

$$\min_{(a,x) \in \Delta_N \times (\mathbb{T}^1)^N} \max_{(b,y) \in \Delta_M \times (\mathbb{T}^1)^M} \left\{ F_2(a, x, b, y) = \sum_{I=1}^N \sum_{J=1}^M a_I b_J \, f(x_I, y_J) \right\}.$$

   A possible approach is to iteratively update (simultaneously) the $a, b$ using MP steps with step-size $\eta$ and the $x, y$ using EG steps with step-size $\gamma\eta$, for some parameter $\gamma > 0$. This algorithm is called Conic Particle Mirror Prox in [WC22], but for concision we will refer to it simply as "EG" in this section. Note that the main challenge in that reference is to deal with the case where $N$ and $M$ are unknown, but here we assume they are known for the sake of simplicity.

In Fig. 2, we show the dependency on $\eta$ and $\gamma$ of the local convergence rate of MP and EG, as well as that of the analogs of Sim-GDA and Alt-GDA and GF for the problem $\min_{(a,x)} \max_{(b,y)} F_2(a, x, b, y)$ in order to illustrate the insights from Sec. 3.2. We used a randomly generated payoff function $f$, and the convergence is measured by the iterates' $\ell_2$ distance to the solution; see the Appendix for details.

---

[5]The choice of $\mathcal{X} = \mathcal{Y} = \mathbb{T}^1$ is made for simplicity of exposition. The discussion below extends straightforwardly to toruses of any dimension, and could be extended to $\mathcal{X}, \mathcal{Y}$ Riemannian manifolds without boundaries at the cost of more technical notation.

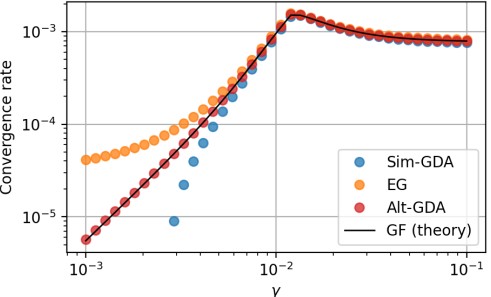 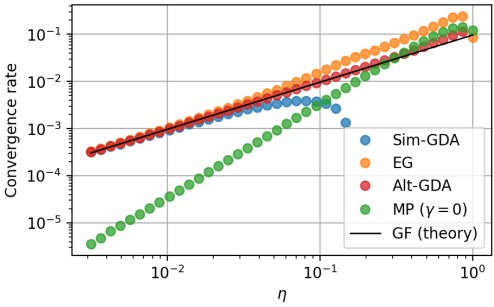

(a) Convergence rate $r$ vs. $\gamma$ (fixed $\eta = 10^{-2}$)  (b) Convergence rate vs. $\eta$ (fixed $\gamma = 10^{-2}$)

Figure 2: Observed local convergence rates $r$ (i.e., $\left\| z^k - z^* \right\| = \Theta((1-r)^k)$) for various conic particle methods using step-size $\eta$ for the weights $(a, b)$ and $\gamma\eta$ for the positions $(x, y)$ and for a fixed random draw of $f$. (left) Fixing $\eta = 10^{-2}$, we observe a rate for GF (we plot $\eta \cdot \tilde{\mu}_\gamma$) scaling as $\gamma^2$ as predicted in Prop. 4.1. Interestingly, Alt-GDA has exactly the same behavior, while the behavior of Sim-GDA and EG differ for small $\gamma$, due to the corrective terms shown in Table 1. (right) Fixing $\gamma = 10^{-2}$, we observe a rate scaling as $\eta$ for EG and its variants, and as $\eta^2$ for MP; the convergence of EG is mostly faster than MP.

MP (the algorithm based on the reparameterization $F_1$) converges to the MNE for any small enough $\eta$ with an exponential rate at least proportional to $\eta^2$ [WLZL21], and this scaling is tight numerically (green dots in Fig. 2b). Its explicit variant (Mirror Descent-Ascent) and its continuous-time flow are known to diverge for any $\eta$ [BP18] [MPP18].

EG is known to converge locally to the MNE for any small enough $\eta$ (and any $\gamma > 0$) with an exponential rate at least proportional to $\eta^2$, despite the non-convexity of $F_2$, under some non-degeneracy assumptions [WC22, Sec. 3.1]. However numerically the convergence rate of EG typically scales as $\eta$, not $\eta^2$ (orange dots in Fig. 2b) – a fact which we will explain below. The explicit variant of EG and its continuous-time flow have not previously been analyzed; the discussion below will give a characterization of when they converge locally.

Note that, at least for the particular $f$ and $\gamma$ used for Fig. 2b, EG converges locally faster than MP for the same $\eta$ (in number of iterations, the per-iteration costs differing only by a constant factor), even though the former does not use the knowledge of the $\{x_I^*\}_I, \{y_J^*\}_J$! In other words, even when the optimal support points are known, it is beneficial to use the overparameterized formulation $F_2$ where we also vary the support points.

**Overparameterization induces partial curvature.** Let us inspect the "Jacobians" at optimum for the two dynamics, MP vs. EG with parameter $\gamma$. Due to the simplex constraints and the non-Euclidean nature of the updates for $a, b$, the relevant matrices are $M_{\mathrm{MP}}$ and $M_\gamma$ defined below, in the sense that the exponential convergence rate of each algorithm is $\tilde{\mu}_M \eta + O(\eta^2)$ (see the Appendix). Namely, omitting half of the antisymmetric off-diagonals for readability,

$$M_{\mathrm{MP}} = \begin{bmatrix} \mathbf{0} & \Pi_a D_a P D_b \Pi_b^\top \\ -(*)^\top & \mathbf{0} \end{bmatrix}$$

where $D_a = \mathrm{Diag}(\sqrt{a^*})$ (square roots being taken component-wise) and $\Pi_a \in \mathbb{R}^{(N-1)\times N}$ is any matrix such that $\Pi_a \Pi_a^\top = I_{N-1}$ and $\Pi_a^\top \Pi_a = I_N - \sqrt{a^*}\sqrt{a^*}^\top$, and likewise for $D_b$ and $\Pi_b \in \mathbb{R}^{(M-1)\times M}$; and for EG,

$$M_\gamma = \begin{bmatrix} \mathbf{0} & \mathbf{0} & \Pi_a D_a P D_b \Pi_b^\top & \sqrt{\gamma}\,\Pi_a D_a [\partial_y P] D_b \\ \mathbf{0} & \gamma\mathrm{Diag}(\partial_{xx}^2 P b^*) & \sqrt{\gamma} D_a [\partial_x P] D_b \Pi_b^\top & \gamma D_a [\partial_{xy}^2 P] D_b \\ -(*)^\top & -(*)^\top & \mathbf{0} & \mathbf{0} \\ -(*)^\top & -(*)^\top & \mathbf{0} & -\gamma\mathrm{Diag}(\partial_{yy}^2 P^\top a^*) \end{bmatrix}$$

where $[\partial_x P]_{IJ} = \partial_x f(x_I^*, y_J^*)$, and likewise for $\partial_y P, \partial_{xx}^2 P, \partial_{yy}^2 P, \partial_{xy}^2 P$.

For MP, it is clear that the equivalent conditions of Thm. 2.1 are violated for any payoff function $f$, and so $\tilde{\mu}_{M_{\mathrm{MP}}} = 0$. For EG, depending on $f$ and $\gamma$, they may or may not be violated. For all of the random payoff functions we considered in our experiments, we observed that $\tilde{\mu}_{M_\gamma} > 0$, suggesting

that the conditions hold generically. They are violated for certain $f$'s however, as shown in the Appendix, so that the scaling in $\eta^2$ of the convergence rate proved in [WC22] is tight in the worst case.

More precisely as we show in the next proposition, assuming that the blocks of $M_\gamma$ are in general position, we expect $\tilde{\mu}_{M_\gamma}$ to scale as $\gamma^2$, which is indeed observed in the numerical experiment reported in Fig. 2a. The proof, placed in the Appendix, relies on the same tools as Prop. 3.1, that is, on the asymptotic expansions of the eigenvalues of perturbed matrices.

**Proposition 4.1.** *Let $S_2$ symmetric and $A_0, A_1, A_2$ antisymmetric real matrices of the form*

$$
S_2 = \left[\begin{array}{c|c} \mathbf{0} & \\ \quad * & \\ \hline & \mathbf{0} \\ & \quad * \end{array}\right], \ A_0 = \left[\begin{array}{cc|c} * & 0 & \\ 0 & 0 & \\ \hline * & 0 & \\ 0 & 0 & \end{array}\right], \ A_1 = \left[\begin{array}{c|cc} & 0 & * \\ & * & 0 \\ \hline 0 & * & \\ * & 0 & \end{array}\right], \ A_2 = \left[\begin{array}{c|cc} & 0 & 0 \\ & 0 & * \\ \hline 0 & 0 & \\ 0 & * & \end{array}\right]
$$

*and $M_\gamma = \gamma S_2 + A_0 + \sqrt{\gamma} A_1 + \gamma A_2$ for all $\gamma > 0$. Then $\tilde{\mu}_{M_\gamma} = O(\gamma^2)$ as $\gamma \to 0$.*

## 5 Conclusion

We have investigated the local convergence of gradient methods for min-max games and found that they converge generically under partial curvature. In more specific settings, we have obtained quantitative estimates of the local convergence rate which exhibit the *average* of the eigenvalues of $S$ as the driving quantity for typical problems. For the computation of mixed Nash equilibria of continuous games, this leads to a behavior of conic particle gradient methods that is more favorable than that described by the worst-case bounds.

More generally, our analysis leads to the following insights: (i) worst-case bounds might be looser in min-max optimization (compared to minimization) as they fail to capture the interplay between interaction and potential parts; (ii) the addition of new degrees of freedom with curvature typically accelerates the local convergence, as we illustrated in Sec. 4.

We note that the phenomenon described in this paper is fundamentally a consequence of the fact that the skewed gradient field's Jacobian at optimum has a positive-semidefinite symmetric part. This property is satisfied at local Nash equilibria of min-max games, i.e., of two-player zero-sum differentiable games, but is not true for general differentiable games.

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
