Table 2 summarizes the local convergence rates of three classical gradient methods, up to third order in $\eta$. The last two columns of the table are valid for $\eta \leq \eta_{\max}$ (given by the second column), and the $O(\cdot)$'s hide only universal constants. In the last column, we denoted $\tilde{\mu} = \tilde{\mu}_M$ for concision, and $L = \rho(M)$. "$k$-EG" stands for $k$-step Extra-Gradient. The table was extracted from the proofs of [AMLG20, Appendix E].

As shown in the fourth column, the quantity $\tilde{\mu}_M$ appears naturally in the bounds on the local convergence rate $1 - \rho(\nabla T(z^*))$, with all methods benefitting from $\tilde{\mu}_M$ being larger. Moreover, the rate depends linearly on $\tilde{\mu}_M$ in the asymptotic regime of small $\eta$, as can be seen in the third column, since $1 - \rho(\nabla T(z^*))^2 \sim 2\eta \min_{\lambda \in \mathrm{Sp}(M)} \Re(\lambda) = 2\eta \tilde{\mu}_M$.[6] This asymptotic equivalence corresponds to the fact that GF, the continuous-time flow of all of the classical gradient methods, has local exponential convergence rate equal to $\tilde{\mu}_M$. (Of course using a large $\eta$, or following a different continuous-time flow than GF, may lead to faster convergence, but it may require knowledge of problem parameters or the use of more complex adaptive schemes.)

Interestingly, as can be seen in the third column, EG may have a slower convergence rate than GDA if $|\Im(\lambda)| < \Re(\lambda)$ for $\lambda = \arg\min_{\mathrm{Sp}(M)} \Re(\cdot)$, when $\eta$ is small.

Table 2: Moduli of eigenvalues of $\nabla T(z^*)$ for classical gradient methods

| Algorithm | $\eta_{\max}$ | $\left\{ \|\nu\|^2, \nu \in \mathrm{Sp}(\nabla T(z^*)) \right\}$ | Upper bound on $\rho(\nabla T(z^*))^2$ |
|---|---|---|---|
| Sim-GDA | $\infty$ | $\left\{ 1 - 2\eta\Re(\lambda) + \eta^2 \|\lambda\|^2, \lambda \in \mathrm{Sp}(M) \right\}$ | $1 - 2\eta\tilde{\mu} + \eta^2 L^2$ |
| $k$-EG $(k \geq 2)$ | $(1-c)/L$ $(\forall c > 0)$ | $\left\{ 1 - 2\eta\Re(\lambda) - \eta^2 \left( \|\lambda\|^2 - 4\Re(\lambda)^2 \right) + O(\frac{1}{1-c}\eta^3 \|\lambda\|^3), \lambda \in \mathrm{Sp}(M) \right\}$ | $1 - \frac{1}{(2-c)^2} \max\left( 2\eta\tilde{\mu}, \eta^2 L^2 \right)$ |
| PP | $\infty$ | $\left\{ \frac{1}{1 + 2\eta\Re(\lambda) + \eta^2 \|\lambda\|^2}, \lambda \in \mathrm{Sp}(M) \right\}$ | $1 - \max\left( \frac{2\eta\tilde{\mu}}{1+2\eta\tilde{\mu}}, \frac{\eta^2 L^2}{1+\eta^2 L^2} \right)$ |

## B  Details for Sec. 3.1

### B.1  Proof of Prop. 3.2

For ease of reference, we restate the proposition below.

**Proposition 3.2.** *Suppose $Q, R$ are fixed and $U, V$ are independently distributed uniformly on the set of $n \times n$ orthonormal matrices. Then*

$$\frac{\mathrm{Tr}(S)}{n} \left( 1 - 2\frac{\|S\|_F}{\mathrm{Tr}(S)} \sqrt{\log n} \right) \leq \mathbb{E}\left[ \min_{1 \leq j \leq n} u_j^\top Q u_j + v_j^\top R v_j \right] \leq \frac{\mathrm{Tr}(S)}{n}$$

*where $\|\cdot\|_F$ denotes the Frobenius norm. In particular, provided that $\frac{\mathrm{Tr}(S)}{\|S\|_F} \geq 2\sqrt{\log n}(1 + c)$ for some fixed $c > 0$, we have $\mathbb{E}\left[ \min_{1 \leq j \leq n} u_j^\top Q u_j + v_j^\top R v_j \right] \asymp \frac{\mathrm{Tr}(S)}{n}$ as $n \to \infty$.*

---

[6] $1 - \rho(\nabla T(z^*))^2$ is the exponential convergence rate for $\|z^k - z^*\|^2$, and $1 - \rho(\nabla T(z^*))$ is the one for $\|z^k - z^*\|$. Hence those two quantities differ by a factor 2 at first order in $\eta$. Equivalently, the additional factor 2 comes from the fact that $\rho^2 = [1 - (1 - \rho)]^2 \approx 1 - 2(1 - \rho)$ when $\rho$ is close to 1.

For the sake of concision, let $W = \begin{bmatrix} U \\ V \end{bmatrix}$ and $w_j = W_{\bullet j} = \begin{pmatrix} u_j \\ v_j \end{pmatrix}$, and (in this appendix only) $\mu :=$ $\min_{j \leq n} u_j^\top Q u_j + v_j^\top R v_j = \min_{j \leq n} w_j^\top S w_j$ the quantity which we want to estimate. Furthermore, denote $s_1 \geq ... \geq s_r > 0 = s_{r+1} = ... = s_{2n}$ the eigenvalues of $S$, and assume $S$ is diagonal w.l.o.g. Also let, for all $j \leq n$, $N_1, ..., N_n, M_1, ..., M_n \sim \chi_n^2$ i.i.d. and independent of $U$ and $V$, and pose

$$a_j = \begin{pmatrix} \sqrt{N_j}\, u_j \\ \sqrt{M_j}\, v_j \end{pmatrix}.$$

Note that for each $j$, $a_j \sim \mathcal{N}(0, I_{2n})$. Indeed, $\sqrt{N_j}\, u_j \sim \mathcal{N}(0, I_n)$ since it is isotropic and its norm has the correct distribution, and likewise for $\sqrt{M_j}\, v_j$; so $a_j \sim \mathcal{N}(0, I_{2n})$ as the concatenation of two independent standard Gaussian vectors.

**Lemma B.1.** *We have $\frac{1}{n}\mathbb{E}\left[\min_{j \leq n} a_j^\top S a_j\right] \leq \mathbb{E}\mu \leq \frac{\mathrm{Tr}(S)}{n}$.*

*Proof.* Let $J := \arg\min_{j \leq n} w_j^\top S w_j$. Then $a_J^\top S a_J = N_J u_J^\top Q u_J + M_J v_J^\top R v_J$. Now $J$ is a deterministic function of the $(w_j)_{j \leq n}$, and $(N_j, M_j)_{j \leq n}$ are i.i.d. and independent of $(w_j)_{j \leq n}$, so we have the conditional expectation $\mathbb{E}[N_J | (w_j)_{j \leq n}] = \mathbb{E}N_1 = \mathbb{E}\chi_n^2 = n$ and likewise for $M_J$. Thus,

$$\mathbb{E}\left[a_J^\top S a_J \big| (w_j)_{j \leq n}\right] = \mathbb{E}\left[N_J | (w_j)_{j \leq n}\right] u_J^\top Q u_J + \mathbb{E}\left[M_J | (w_j)_{j \leq n}\right] v_J^\top R v_J = n\mu,$$

and the first inequality follows by taking total expectations.

For the second inequality, $\mathbb{E}\mu \leq \mathbb{E}u_1^\top Q u_1 + v_1^\top R v_1$. Now letting $a_1^{(1)} = \sqrt{N_1}\, u_1 \sim \mathcal{N}(0, I_n)$, $N_1 u_1^\top Q u_1 = (a_1^{(1)})^\top Q a_1^{(1)}$, so taking expectations on both sides, $n\mathbb{E}u_1^\top Q u_1 = \mathrm{Tr}(Q)$. So $\mathbb{E}\mu \leq \frac{\mathrm{Tr}(Q) + \mathrm{Tr}(R)}{n} = \frac{\mathrm{Tr}(S)}{n}$. $\square$

Let for concision $\zeta_j = a_j^\top S a_j$ for each $j \leq n$.

**Lemma B.2.** *The moment-generating function of $\zeta_1 (\sim \zeta_2 \sim ... \sim \zeta_n)$ is*

$$\mathbb{E}e^{t\zeta_1} = \prod_{l=1}^{r}(1 - 2s_l t)^{-1/2} \quad \text{for all} \quad t < \frac{1}{2\max_l s_l}.$$

*Proof.* Since $\zeta_1 = \sum_{l=1}^{2n} s_l a_1[l]^2$ and $a_1[l]^2 \sim \chi^2$ i.i.d., we have $\mathbb{E}e^{ta_1[l]^2} = (1 - 2t)^{-1/2}$ for all $t < \frac{1}{2}$ and $\mathbb{E}e^{t\zeta_1} = \prod_{l=1}^{r}\mathbb{E}e^{ts_l a_1[l]^2} = \prod_{l=1}^{r}(1 - 2s_l t)^{-1/2}$ for all $t < \frac{1}{2\max_l s_l}$. $\square$

We now lower-bound the expectation of $\min_{j \leq n} \zeta_j$ using a Chernoff bound-type argument, which we note does not require independence.

**Lemma B.3.** *We have*

$$\mathbb{E}\min_{j \leq n} \zeta_j \geq \mathrm{Tr}(S)\left(1 - 2\frac{\|S\|_F}{\mathrm{Tr}(S)}\sqrt{\log n}\right)$$

*where $\|\cdot\|_F$ denotes Frobenius norm, i.e., $\ell_2$-norm of the vector of eigenvalues.*

*Proof.* By Jensen's inequality and monotonicity of $\exp(\cdot)$,

$$\forall t > 0, \quad \exp\left(t\, \mathbb{E}\max_{j \leq n} -\zeta_j\right) \leq \mathbb{E}\exp\left(t\max_{j \leq n} -\zeta_j\right) = \mathbb{E}\max_{j \leq n}\exp\left(-t\zeta_j\right).$$

So, taking $\log$ and optimizing the bound,

$$\begin{aligned}
\mathbb{E}\max_{j \leq n}(-\zeta_j) &\leq \inf_{t > 0}\frac{1}{t}\log \mathbb{E}\max_{j \leq n}\exp\left(t(-\zeta_j)\right) \\
&\leq \inf_{t > 0}\frac{1}{t}\log\left[n\mathbb{E}e^{t(-\zeta_1)}\right] = \inf_{t > 0}\frac{1}{t}\log\left[n\prod_{l=1}^{r}(1 + 2s_l t)^{-1/2}\right] \\
&= \inf_{t > 0}\frac{1}{t}\left(\log(n) - \frac{1}{2}\sum_{l=1}^{r}\log(1 + 2s_l t)\right) =: \inf_{t > 0} g(t).
\end{aligned} \tag{B.1}$$

By calculating we find that $g'(t) > 0 \iff \sum_{l=1}^r \left[ \log(1 + 2s_l t) - \frac{2s_l t}{1+2s_l t} \right] > 2 \log n$. With the case $r \gg \log n$ in mind, let us evaluate at $t^{(1)}$ defined by $\sum_{l=1}^r 2(s_l t)^2 = 2 \log n$, i.e., $t^{(1)} = \sqrt{\log n} / \|S\|_F$. (This choice is obtained by Taylor expansion for $t \to 0$ of the condition $g'(t) = 0$.) Using that $\log(1 + y) \geq y - \frac{1}{2} y^2$ for $y \geq 0$, we get

$$
\begin{aligned}
g(t^{(1)}) &= \frac{\|S\|_F}{\sqrt{\log n}} \left( \log n - \frac{1}{2} \sum_{l=1}^r \log \left( 1 + 2s_l \frac{\sqrt{\log n}}{\|S\|_F} \right) \right) \\
&\leq \frac{\|S\|_F}{\sqrt{\log n}} \left( \log n - \frac{1}{2} \sum_{l=1}^r \left( 2s_l \frac{\sqrt{\log n}}{\|S\|_F} - \frac{1}{2} 4 s_l^2 \frac{\log n}{\|S\|_F^2} \right) \right) \\
&= \frac{\|S\|_F}{\sqrt{\log n}} \left( \log n - \frac{\mathrm{Tr}(S) \sqrt{\log n}}{\|S\|_F} + \log n \right) \\
&= - \mathrm{Tr}(S) + 2 \|S\|_F \sqrt{\log n}.
\end{aligned}
$$

Thus, $\mathbb{E} \min_{j \leq n} \zeta_j \geq -g(t^{(1)}) \geq \mathrm{Tr}(S) - 2 \|S\|_F \sqrt{\log n} = \mathrm{Tr}(S) \left( 1 - 2 \frac{\|S\|_F}{\mathrm{Tr}(S)} \sqrt{\log n} \right)$. $\qquad\square$

The upper bound of Prop. 3.2 is shown in Lemma B.1, and the lower bound follows immediately from substituting Lemma B.3 into Lemma B.1.

## B.2   Proof of Prop. 3.3

For ease of reference, we restate the proposition below.

**Proposition 3.3.** *Suppose $Q, R$ are fixed and $U, V$ are independently distributed uniformly on the set of $n \times n$ orthonormal matrices. Let $s_1 \geq ... \geq s_r > 0 = s_{r+1} = ... = s_{2n}$ the eigenvalues of $S$. Then*

$$
\mathbb{E} \left[ \min_{j \leq n} u_j^\top Q u_j + v_j^\top R v_j \right] \geq \max_{\mathcal{S} \subset \{1,...,r\}} \frac{1}{e} \frac{|\mathcal{S}|}{n} n^{-\frac{2}{|\mathcal{S}|}} \left[ \prod_{l \in \mathcal{S}} s_l \right]^{\frac{1}{|\mathcal{S}|}}.
$$

*In particular, $\mathbb{E} \left[ \min_{j \leq n} u_j^\top Q u_j + v_j^\top R v_j \right] \gtrsim n^{-\frac{2}{r}-1}$ when $n \to \infty$ and $r$ and $s \in \mathbb{R}^r$ are fixed.*

We are still exactly in the same setting as for Prop. 3.2, so all the lemmas of Sec. B.1 apply. We also reuse notations from that subsection: $\mu = \min_{j \leq n} u_j^\top Q u_j + v_j^\top R v_j$, $a_j = \begin{pmatrix} \sqrt{N_j}\, u_j \\ \sqrt{M_j}\, v_j \end{pmatrix}$ where $N_1, ..., N_n, M_1, ..., M_n \sim \chi_n^2$ i.i.d. and independent of $U$ and $V$, and $\zeta_j = a_j^\top S a_j$ for $j \leq n$. We also assume $S$ diagonal w.l.o.g.

Recall from Lemma B.1 that $\mathbb{E}\mu \geq \frac{1}{n} \mathbb{E} \min_{j \leq n} \zeta_j$. The proposition then follows immediately from the following lemma.

**Lemma B.4.** *Let any $\mathcal{S} \subset \{1, ..., r\}$, denote $r_\mathcal{S} = |\mathcal{S}|$ and $G_\mathcal{S} = \left[ \prod_{l \in \mathcal{S}} s_l \right]^{1/r_\mathcal{S}}$. Then we have $\mathbb{E} \min_{j \leq n} \zeta_j \geq \frac{1}{e} r_\mathcal{S} \, n^{-\frac{2}{r_\mathcal{S}}} G_\mathcal{S}$.*

*Proof.* Recall from (B.1) that

$$
\mathbb{E} \max_{j \leq n} (-\zeta_j) \leq \inf_{t > 0} \frac{1}{t} \left( \log(n) - \frac{1}{2} \sum_{l=1}^r \log(1 + 2s_l t) \right) =: \inf_{t > 0} g(t)
$$

and that $g'(t) > 0 \iff \sum_{l=1}^r \left[ \log(1 + 2s_l t) - \frac{2s_l t}{1 + 2s_l t} \right] > 2 \log n$.

With the case $r \ll \log n$ in mind, let us evaluate at $t^{(2)}$ defined by $\sum_{l=1}^r [\log(1 + 2s_l t) - 1] = 2 \log n$, i.e., $\prod_{l=1}^r (1 + 2s_l t) = e^{2 \log n + r} = n^2 e^r$ – more precisely, let $t^{(2)}$ be the smallest positive such $t$ (since there may be several solutions to that polynomial equation). Then $g(t^{(2)}) = \frac{1}{t^{(2)}} (-r/2)$.

Let us further upper-bound $t^{(2)}$ by some $u > 0$. Since $t^{(2)}$ is defined as the smallest positive root of the polynomial $P(t) = \prod_{l=1}^r (1 + 2s_l t) - n^2 e^r$, and since $P(0) < 0$, it suffices to pick

any $u > 0$ that $P(u) \geq 0$. Since $P(t) \geq \left[\prod_{l=1}^{r} 2s_l\right] t^r - n^2 e^r$ for any $t > 0$,[7] we may choose $u = (n^2 e^r)^{\frac{1}{r}} / \left[\prod_{l=1}^{r} 2s_l\right]^{\frac{1}{r}}$.

Thus we have

$$-\mathbb{E} \min_{j \leq n} \zeta_j \leq g(t^{(2)}) = \frac{1}{t^{(2)}}(-r/2) \leq \frac{1}{u}(-r/2) = \frac{\left[\prod_{l=1}^{r} 2s_l\right]^{1/r}}{(n^2 e^r)^{1/r}}(-r/2)$$

$$\mathbb{E} \min_{j \leq n} \zeta_j \geq r \frac{\left[\prod_{l=1}^{r} s_l\right]^{1/r}}{e \cdot n^{2/r}}.$$

This shows the lemma in the case $\mathcal{S} = \{1, ..., r\}$.

For the case of arbitrary $\mathcal{S} \subset \{1, ..., r\}$, drop some terms in the Chernoff bound-type inequality; that is, upper-bound $-\sum_{l \notin \mathcal{S}} \log(1 + 2s_l t)$ by zero:

$$-\mathbb{E} \min_{j \leq n} \zeta_j \leq \inf_{t>0} \frac{1}{t}\left(\log(n) - \frac{1}{2}\sum_{l=1}^{r} \log(1 + 2s_l t)\right) \leq \inf_{t>0} \underbrace{\frac{1}{t}\left(\log(n) - \frac{1}{2}\sum_{l \in \mathcal{S}} \log(1 + 2s_l t)\right)}_{=:g_{\mathcal{S}}(t)}.$$

Thereafter, going through exactly the same considerations as above, we obtain the inequality where we restricted attention to the components $l \in \mathcal{S}$. $\qquad\square$

### B.3 High-probability bounds for the spread-out spectrum case

In this subsection we provide a high-probability counterpart to the expectation estimate from Prop. 3.2, where we showed $\mathbb{E}\left[\min_{j \leq n} u_j^\top Q u_j + v_j^\top R v_j\right] \sim \frac{\mathrm{Tr}(S)}{n}$ when $\frac{\mathrm{Tr}(S)}{\|S\|_F} \geq \Omega(\sqrt{\log n})$.

**Proposition B.5.** *Suppose $Q, R$ are fixed and $U, V$ are independently distributed uniformly on the set of $n \times n$ orthonormal matrices. Then, denoting $\mu = \min_{j \leq n} u_j^\top Q u_j + v_j^\top R v_j$,*

$$\forall 0 \leq \gamma \leq 1, \ \mathbb{P}\left(\mu \geq \frac{\mathrm{Tr}(S)}{2n}(1 - \gamma)\right) \geq 1 - n e^{-\frac{\mathrm{Tr}(S)^2}{4\|S\|_F^2}\gamma^2} - 2e^{-\frac{n}{8}}$$

$$and \quad \forall \gamma \geq 0, \ \mathbb{P}\left(\mu \leq \frac{4\,\mathrm{Tr}(S)}{n}(1 + \gamma)\right) \geq \begin{cases} 1 - e^{-\frac{\mathrm{Tr}(S)^2}{8\|S\|_F^2}\gamma^2} - 2e^{-\frac{n}{8}} & if\ \gamma \leq \frac{\|S\|_F^2}{\mathrm{Tr}(S)\|\!|S|\!\|} \\ 1 - e^{-\frac{\mathrm{Tr}(S)}{8\|\!|S|\!\|}\gamma} - 2e^{-\frac{n}{8}} & otherwise \end{cases}$$

*where $\|\cdot\|_F$ denotes the Frobenius norm and $\|\!|\cdot|\!\|$ denotes the operator norm.*

The remainder of this subsection is dedicated to proving the above proposition. Let as in Sec. B.1 $a_j = \begin{pmatrix} \sqrt{N_j}\, u_j \\ \sqrt{M_j}\, v_j \end{pmatrix}$ where $N_1, ..., N_n, M_1, ..., M_n \sim \chi_n^2$ i.i.d. and independent of $U$ and $V$, and $\zeta_j = a_j^\top S a_j$ for $j \leq n$. Also denote $s_1 \geq ... \geq s_r > 0 = s_{r+1} = ... s_{2n}$ the eigenvalues of $S$, and assume $S$ diagonal w.l.o.g.

**Lemma B.6.** *For each $j$, $\mathbb{P}\left(\frac{1}{2n} \leq \frac{1}{N_j}\right) \geq 1 - e^{-\frac{n}{8}}$ and $\mathbb{P}\left(\frac{1}{N_j} \leq \frac{4}{n}\right) \geq 1 - e^{-\frac{n}{8}}$.*

*Proof.* Since $N_j \sim \chi_n^2$, we have the classical concentration bounds [LM00, Equations (4.3), (4.4)]

$$\forall t > 0, \ \mathbb{P}\left(N_j - n \leq 2\sqrt{nt} + 2t\right) \geq 1 - e^{-t} \quad \text{and} \quad \mathbb{P}\left(N_j - n \geq -2\sqrt{nt}\right) \geq 1 - e^{-t}.$$

Evaluating the first inequality at $t = \frac{n}{8}$ yields $N_j \leq n + \frac{n}{\sqrt{2}} + \frac{n}{4} \leq 2n$ with probability $\geq 1 - e^{-\frac{n}{8}}$. Evaluating the second inequality at $t = \frac{n}{8}$ yields $N_j \geq n - \frac{n}{\sqrt{2}} \geq \frac{n}{4}$ with probability $\geq 1 - e^{-\frac{n}{8}}$. $\quad\square$

---

[7]And this approximation of $P(t)$ intuitively makes sense to do since our choice of $t^{(2)}$ was guided by the Ansatz that $\frac{2s_l t}{1 + 2s_l t} \approx 1$, i.e. $s_l t \gg 1$, for all $l \leq r$.

Letting the random variable $J = \arg\min_{j \leq n} u_j^\top Q u_j + v_j^\top R v_j$ that is only dependent on $U$ and $V$, so independent of the $N_j$, $M_j$, we have

$$\mu = \frac{1}{N_J} \cdot \sqrt{N_J} u_J^\top Q \sqrt{N_J} u_J + \frac{1}{M_J} \cdot \sqrt{M_J} v_J^\top R \sqrt{M_J} v_J \geq \left( \frac{1}{N_J} \wedge \frac{1}{M_J} \right) a_J^\top S a_J$$

$$\geq \left( \frac{1}{N_J} \wedge \frac{1}{M_J} \right) \min_{j \leq n} a_j^\top S a_j$$

and so by union bound, using that $N_J \sim N_1$ and $M_J \sim M_1$ by independence,

$$\forall c \geq 0, \ \mathbb{P}\left( \mu \geq c \right) \geq \mathbb{P}\left( \min_{j \leq n} a_j^\top S a_j \geq 2nc \right) - 2e^{-\frac{n}{8}}. \tag{B.2}$$

Let for concision $\zeta_j = a_j^\top S a_j$ for each $j \leq n$. Recall from Lemma B.2 that the moment-generating function of $\zeta_1 (\sim \zeta_2 \sim ... \sim \zeta_n)$ is

$$\mathbb{E} e^{t\zeta_1} = \prod_{l=1}^r (1 - 2s_l t)^{-1/2} \quad \text{for all} \quad t < \frac{1}{2 \max_l s_l}.$$

We can now use union bound and Chernoff's bound to lower-bound $\min_{j \leq n} \zeta_j$ with high probability. The derivation is essentially an instantiation of the general subexponential tail bound [Wai19, Sec. 2.1.3] using our precise knowledge of the moment-generating function of the $\zeta_j$.

**Lemma B.7.** *For any $0 \leq \gamma \leq 1$, we have*

$$\mathbb{P}\left( \min_{j \leq n} \zeta_j \geq \frac{\mathrm{Tr}(S)}{2n}(1 - \gamma) \right) \geq 1 - n e^{-\frac{\mathrm{Tr}(S)^2}{4\|S\|_F^2}\gamma^2}$$

*where $\|\cdot\|_F$ denotes Frobenius norm, i.e., $\ell_2$-norm of the vector of eigenvalues.*

*Proof.* By union bound, since the $\zeta_j$ are identically distributed, $\mathbb{P}\left( \min_j \zeta_j \leq x \right) \leq n\mathbb{P}\left( \zeta_1 \leq x \right)$. By Markov's inequality,

$$\mathbb{P}\left( \zeta_1 \leq x \right) = \mathbb{P}\left( -\zeta_1 \geq -x \right) \leq \inf_{t>0} \frac{1}{e^{-tx}} \mathbb{E} e^{-t\zeta_1} = \inf_{t>0} e^{tx} \prod_{l=1}^r (1 + 2s_l t)^{-1/2}$$

$$\log \mathbb{P}\left( \zeta_1 \leq x \right) \leq \inf_{t>0} tx - \frac{1}{2} \sum_{l=1}^r \log(1 + 2s_l t) =: \inf_{t>0} g_x(t).$$

By calculating we find that $g_x'(t) > 0 \iff 2x > \sum_{l=1}^r \frac{2s_l t}{1+2s_l t}$. With the case $x \approx \mathbb{E}\zeta_1 = \mathrm{Tr}(S) \ll r$ in mind, let us evaluate at $t_x^{(1)}$ defined by $2x = \sum_{l=1}^r 2s_l(1 - 2s_l t) = 2\mathrm{Tr}(S) - 4\|S\|_F^2 t$, i.e., $t_x^{(1)} = \frac{\mathrm{Tr}(S)-x}{2\|S\|_F^2}$. Assume henceforth that $x \leq \mathrm{Tr}(S)$ so that $t_x^{(1)} \geq 0$. Using that $\log(1 + y) \geq y - \frac{1}{2}y^2$ for $y \geq 0$, we get

$$g_x(t_x^{(1)}) = \frac{\mathrm{Tr}(S) - x}{2\|S\|_F^2} x - \frac{1}{2} \sum_{l=1}^r \log\left( 1 + 2s_l \frac{\mathrm{Tr}(S) - x}{2\|S\|_F^2} \right)$$

$$\leq \frac{\mathrm{Tr}(S)x - x^2}{2\|S\|_F^2} - \frac{1}{2} \sum_{l=1}^r 2s_l \frac{\mathrm{Tr}(S) - x}{2\|S\|_F^2} + \frac{1}{4} \sum_{l=1}^r 4s_l^2 \left( \frac{\mathrm{Tr}(S) - x}{2\|S\|_F^2} \right)^2$$

$$= \frac{\mathrm{Tr}(S)x - x^2}{2\|S\|_F^2} - \mathrm{Tr}(S) \frac{\mathrm{Tr}(S) - x}{2\|S\|_F^2} + \|S\|_F^2 \left( \frac{\mathrm{Tr}(S) - x}{2\|S\|_F^2} \right)^2$$

$$= \frac{1}{2\|S\|_F^2} \left[ \mathrm{Tr}(S)x - x^2 - \mathrm{Tr}(S)^2 + \mathrm{Tr}(S)x + \frac{1}{2}(\mathrm{Tr}(S) - x)^2 \right]$$

$$= -\frac{1}{4\|S\|_F^2} [\mathrm{Tr}(S) - x]^2.$$

Hence we have shown

$$\forall x \leq \mathrm{Tr}(S), \; \mathbb{P}\left(\min_j \zeta_j \leq x\right) \leq n\mathbb{P}\left(\zeta_1 \leq x\right) \leq n \exp\left(-\frac{1}{4\left\|S\right\|_F^2}\left[\mathrm{Tr}(S) - x\right]^2\right)$$

and the announced bound follows by a change of variables $\gamma = 1 - \frac{x}{\mathrm{Tr}(S)}$. $\qquad\square$

Combining (B.2) with Lemma B.7 yields the high-probability lower bound

$$\forall 0 \leq \gamma \leq 1, \quad \mathbb{P}\left(\mu \geq \frac{\mathrm{Tr}(S)}{2n}(1 - \gamma)\right) \geq 1 - ne^{-\frac{\mathrm{Tr}(S)^2}{4\left\|S\right\|_F^2}\gamma^2} - 2e^{-\frac{n}{8}}$$

which is exactly the lower bound of Prop. B.5.

For a high-probability upper bound on $\mu = \min_{j \leq n} w_j^\top S w_j$, in the regime of interest in this subsection it is sufficient to start from

$$\mu \leq w_1^\top S w_1 = \frac{1}{N_1} \cdot \sqrt{N_1} u_1^\top Q \sqrt{N_1} u_1 + \frac{1}{M_1} \cdot \sqrt{M_1} v_1^\top R \sqrt{M_1} v_1 \leq \left(\frac{1}{N_1} \vee \frac{1}{M_1}\right) a_1^\top S a_1$$

so that, by union bound and Lemma B.6,

$$\forall C \geq 0, \; \mathbb{P}(\mu \leq C) \geq \mathbb{P}\left(a_1^\top S a_1 \leq \frac{n}{4}C\right) - 2e^{-\frac{n}{8}}. \tag{B.3}$$

Hence it is sufficient to upper-bound $\zeta_1 = a_1^\top S a_1$ with high probability.

**Lemma B.8.** *For any $\varepsilon \geq 0$, we have*

$$\mathbb{P}\left(\zeta_1 \leq \mathrm{Tr}(S) + \varepsilon\right) \geq \begin{cases} 1 - e^{-\frac{\varepsilon^2}{8\left\|S\right\|_F^2}} & \textit{if } 0 \leq \varepsilon \leq \frac{\left\|S\right\|_F^2}{\max_l s_l} \\ 1 - e^{-\frac{\varepsilon}{8\max_l s_l}} & \textit{otherwise.} \end{cases}$$

*Proof.* $\zeta_1 = \sum_{l=1}^r s_l a_1[l]^2$ is subexponential with parameters $\left(\sqrt{\sum_{l=1}^r (2 \cdot s_l)^2}, \max_{l \leq r} 4 \cdot s_l\right)$ as a linear combinations of independent $\chi^2$ random variables $a_j[l]^2$ which are subexponential with parameters $(2, 4)$ [Wai19, Sec. 2.1.3]. The announced subexponential tail bound follows by a direct application of [Wai19, Prop. 2.9]. $\qquad\square$

Combining (B.3) with Lemma B.8 yields the high-probability upper bound

$$\forall \varepsilon \geq 0, \quad \mathbb{P}\left(\mu \leq 4\frac{\mathrm{Tr}(S) + \varepsilon}{n}\right) \geq \begin{cases} 1 - e^{-\frac{\varepsilon^2}{8\left\|S\right\|_F^2}} - 2e^{-\frac{n}{8}} & \text{if } \varepsilon \leq \frac{\left\|S\right\|_F^2}{\max_l s_l} \\ 1 - e^{-\frac{\varepsilon}{8\max_l s_l}} - 2e^{-\frac{n}{8}} & \text{otherwise} \end{cases}$$

and so $\quad \forall \gamma \geq 0, \quad \mathbb{P}\left(\mu \leq \frac{4\,\mathrm{Tr}(S)}{n}(1 + \gamma)\right) \geq \begin{cases} 1 - e^{-\frac{\mathrm{Tr}(S)^2}{8\left\|S\right\|_F^2}\gamma^2} - 2e^{-\frac{n}{8}} & \text{if } \gamma \leq \frac{\left\|S\right\|_F^2}{\mathrm{Tr}(S)[\max_l s_l]} \\ 1 - e^{-\frac{\mathrm{Tr}(S)\gamma}{8\max_l s_l}} - 2e^{-\frac{n}{8}} & \text{otherwise} \end{cases}$

by the change of variables $\mathrm{Tr}(S)\gamma = \varepsilon$, which is exactly the upper bound of Prop. B.5.

## B.4 High-probability bounds for the sparse spectrum case

In this subsection we provide a high-probability counterpart to the expectation lower estimate from Prop. 3.3, where we showed $\mathbb{E}\left[\min_{j \leq n} u_j^\top Q u_j + v_j^\top R v_j\right] \gtrsim n^{-\frac{2}{r}-1}$ when $n \to \infty$ with $r = \mathrm{rank}(S)$ and $s \in \mathbb{R}^r$ fixed.

**Proposition B.9.** *Suppose $Q, R$ are fixed and $U, V$ are independently distributed uniformly on the set of $n \times n$ orthonormal matrices. Let $s_1 \geq \dots \geq s_r > 0 = s_{r+1} = \dots = s_{2n}$ the eigenvalues of $S$. Then denoting $\mu = \min_{j \leq n} u_j^\top Q u_j + v_j^\top R v_j$, for any $\mathcal{S} \subset \{1, \dots, r\}$,*

$$\forall 0 \leq \gamma \leq 1, \; \mathbb{P}\left(\mu \geq \frac{1}{2e}\frac{|\mathcal{S}|}{n}n^{-\frac{2}{|\mathcal{S}|}}\left[\prod_{l \in \mathcal{S}} s_l\right]^{\frac{1}{|\mathcal{S}|}} \cdot (1 - \gamma)\right) \geq 1 - e^{-\frac{|\mathcal{S}|}{2}\gamma} - 2e^{-\frac{n}{8}}.$$

The remainder of this subsection is dedicated to proving the above proposition. As we are exactly in the same setting as for [Prop. B.5], all the lemmas of [Sec. B.3] apply. We also reuse notations from that subsection: $a_j = \begin{pmatrix} \sqrt{N_j}\, u_j \\ \sqrt{M_j}\, v_j \end{pmatrix}$ where $N_1, ..., N_n, M_1, ..., M_n \sim \chi_n^2$ i.i.d. and independent of $U$ and $V$, and $\zeta_j = a_j^\top S a_j$ for $j \leq n$. We also assume $S$ diagonal w.l.o.g.

Given (B.2), in order to prove the proposition it suffices to derive a different lower tail bound for $\min_{j \leq n} \zeta_j$ that is more adapted to the sparse spectrum case.

**Lemma B.10.** *Let any $\mathcal{S} \subset \{1, ..., r\}$, $r_\mathcal{S} = |\mathcal{S}|$ and $G_\mathcal{S} = \left[ \prod_{l \in \mathcal{S}} s_l \right]^{1/r_\mathcal{S}}$. Then*

$$\forall \gamma \in \mathbb{R}, \ \mathbb{P}\left( \min_{j \leq n} \zeta_j \geq \frac{r_\mathcal{S}}{e} n^{-\frac{2}{r_\mathcal{S}}} \left[ \prod_{l \in \mathcal{S}} s_l \right]^{\frac{1}{r_\mathcal{S}}} \cdot (1 - \gamma) \right) \geq 1 - e^{-\frac{r_\mathcal{S}}{2}\gamma}.$$

*Proof.* Let any $x \in \mathbb{R}$. By Markov's inequality,

$$\mathbb{P}\left( \min_j \zeta_j \leq x \right) = \mathbb{P}\left( \max_j (-\zeta_j) \geq -x \right) \leq \inf_{t > 0} \frac{1}{e^{-tx}} \mathbb{E} e^{t \max_j (-\zeta_j)} = \inf_{t > 0} e^{tx} \cdot \mathbb{E} \max_j e^{t(-\zeta_j)}.$$

Now for any $\alpha \geq 1$, by monotonicity and concavity of $y \mapsto y^{1/\alpha}$,

$$\mathbb{E} \max_{j \leq n} e^{t(-\zeta_j)} = \mathbb{E}\left[ \left( \max_{j \leq n} e^{\alpha t(-\zeta_j)} \right)^{1/\alpha} \right] \leq \mathbb{E}\left[ \left( \sum_{j \leq n} e^{\alpha t(-\zeta_j)} \right)^{1/\alpha} \right] \leq \left[ \mathbb{E} \sum_{j \leq n} e^{\alpha t(-\zeta_j)} \right]^{1/\alpha}$$

$$= \left[ n \mathbb{E} e^{\alpha t(-\zeta_1)} \right]^{1/\alpha}.$$

Hence, taking the infimum over $\alpha$ and substituting into the above inequality,[8]

$$\mathbb{P}\left( \min_j \zeta_j \leq x \right) \leq \inf_{t > 0} e^{tx} \cdot \inf_{\alpha \geq 1} \left[ n \mathbb{E} e^{\alpha t(-\zeta_1)} \right]^{1/\alpha}$$

$$\log \mathbb{P}\left( \min_j \zeta_j \leq x \right) \leq \inf_{t > 0} tx + \inf_{\alpha \geq 1} \frac{1}{\alpha} \left( \log n + \log \mathbb{E} e^{\alpha t(-\zeta_1)} \right)$$

$$= \inf_{t > 0} t \left[ x + \inf_{\beta \geq t} \frac{1}{\beta} \left( \log n + \log \mathbb{E} e^{\beta(-\zeta_1)} \right) \right].$$

Let, as in the proof of [Lemma B.4],

$$g(\beta) := \frac{1}{\beta} \left( \log n + \log \mathbb{E} e^{-\beta \zeta_1} \right) = \frac{1}{\beta} \left( \log n - \frac{1}{2} \sum_{l=1}^r \log(1 + 2 s_l \beta) \right) \quad \text{and} \quad u := \frac{e}{2} n^{\frac{2}{r}} \left[ \prod_{l=1}^r s_l \right]^{-\frac{1}{r}}.$$

One can check, by lower-bounding $\log(1 + 2 s_l u)$ by $\log(2 s_l u)$ for each term, that $g(u) \leq \frac{1}{u}(-r/2)$. Hence,

$$\log \mathbb{P}\left( \min_j \zeta_j \leq x \right) \leq \inf_{t > 0} t \left[ x + \inf_{\beta \geq t} g(\beta) \right] \leq \inf_{0 < t \leq u} t \left( x + g(u) \right)$$

$$\leq u(x + g(u)) \leq ux - \frac{r}{2} = \frac{r}{2} \cdot \left( \frac{u}{r/2} x - 1 \right).$$

Hence we have shown that for all $x \in \mathbb{R}$,

$$\mathbb{P}\left( \min_j \zeta_j \leq x \right) \leq \exp\left( -\frac{r}{2} \left( 1 - \frac{u}{r/2} x \right) \right).$$

---

[8]In fact one can check, by inverting $\inf_t$ and $\inf_\alpha$ and optimizing first over $t$, that the infimum over the joint $(t, \alpha)$ is always attained at some $(t^*, 1)$; that is, the true optimum is always at $\alpha = 1$. What introducing the extra degree of freedom $\alpha$ affords us is a way to obtain a tractable upper bound, whereas upper-bounding the $\inf_t$ with $\alpha = 1$ fixed seems more difficult.

The inequality of the lemma with $\mathcal{S} = \{1, ..., r\}$ follows by the change of variable $\gamma = 1 - \frac{u}{r/2}x$ and substituting the expression of $u$.

The inequality for arbitrary $\mathcal{S} \subset \{1, ..., r\}$ follows by exactly the same considerations as above applied to

$$g_{\mathcal{S}}(\beta) = \frac{1}{\beta}\left(\log n - \frac{1}{2}\sum_{l \in \mathcal{S}}\log(1 + 2s_l\beta)\right) \quad \text{and} \quad u_{\mathcal{S}} = \frac{e}{2}n^{\frac{2}{r_{\mathcal{S}}}}\left[\prod_{l \in \mathcal{S}}s_l\right]^{-\frac{1}{r_{\mathcal{S}}}}$$

instead of $g$ and $u$, noting that $g(\beta) \leq g_{\mathcal{S}}(\beta)$ for any $\beta > 0$. $\qquad\square$

Combining (B.2) with Lemma B.10 yields the high-probability lower bound

$$\forall \gamma \in \mathbb{R}, \ \ \mathbb{P}\left(\mu \geq \frac{1}{2n}\frac{|\mathcal{S}|}{e}n^{-\frac{2}{|\mathcal{S}|}}\left[\prod_{l \in \mathcal{S}}s_l\right]^{\frac{1}{|\mathcal{S}|}}\cdot(1-\gamma)\right) \geq 1 - e^{-\frac{|\mathcal{S}|}{2}\gamma} - 2e^{-\frac{n}{8}}$$

for any $\mathcal{S} \subset \{1, ..., r\}$, which is exactly the inequality of Prop. B.5.

# C  A symmetric expression for Alt-GDA

This appendix contains results stated in Sec. 3.2 and used in Sec. D.3 and that may be of independent interest.

As announced in Sec. 3.2, we analyze a symmetrized formulation of Alt-GDA, whose definition we recall here for ease of reference: let $(x^0, y^{1/2}) \in \mathbb{R}^d \times \mathbb{R}^d$ and

$$\begin{cases}\forall k \in \mathbb{N}, & x^{k+1} = x^k - \eta\nabla_x f(x^k, y^{k+1/2}) \\ \forall k \in \mathbb{N}+1/2, & y^{k+1} = y^k + \eta\nabla_y f(x^{k+1/2}, y^k)\end{cases} \quad \text{and} \quad \begin{cases}\forall k \in \mathbb{N}, & x^{k+1/2} = \frac{x^{k+1}+x^k}{2} \\ \forall k \in \mathbb{N}+1/2, & y^{k+1/2} = \frac{y^{k+1}+y^k}{2}.\end{cases}$$
$$\text{(C.1)}$$

Note that this symmetrized formulation is indeed equivalent to standard one (used e.g. in [Gid+19; BGP20; ZWLG22; GLWM22]), via the correspondence $\forall k \in \mathbb{N}, \tilde{x}^k = x^k$ and $\tilde{y}^k = y^{k+1/2}$. We emphasize that, as one can easily check from the definition, the symmetrized iterates $(z^k)_k = (x^k, y^k)_k$ converge if and only if the classical ones $(\tilde{z}^k)_k = (x^k, y^{k+1/2})_{k \in \mathbb{N}}$ do, and if they converge exponentially, then they have the same convergence rate.

The following proposition shows that the evolution of the symmetrized Alt-GDA iterates can be expressed approximately in terms only of the (skewed) gradient field $g(z)$.

**Proposition C.1.** *Assume $\eta \leq \left\|\nabla^2_{xx}f\right\|_\infty^{-1} \wedge \left\|\nabla^2_{yy}f\right\|_\infty^{-1}$. Then the symmetrized Alt-GDA iterates defined by (C.1) satisfy*

$$\forall k \in \mathbb{N} \cup (\mathbb{N}+1/2), \ z^{k+1} = z^k - \eta g(z^k) + \frac{\eta^2}{2}A(z^k)g(z^k) + O\left(\eta^3\left\|g(z^k)\right\|\right)$$

*where $A(z) = \begin{bmatrix} 0 & \nabla^2_{xy}f \\ -\nabla^2_{xy}f^\top & 0 \end{bmatrix}(z)$. More precisely the "$O(\cdot)$" term is a vector with norm at most $10\eta^3\left\|g(z^k)\right\|\left(\left\|\nabla^3 f\right\|_\infty(1 + \left\|\nabla^2_{xy}f\right\|_\infty^2\eta^2) + \left\|\nabla^2 f\right\|_\infty\left\|g(z^k)\right\|\right).$*

*Remark* C.1 (High-resolution ODE for the symmetrized iterates). The symmetrized update operator with the $O(\eta^3)$ term neglected, $z \mapsto z - \eta g(z) + \frac{\eta^2}{2}A(z)g(z)$, leads to a different high-resolution ODE than the one derived by [GLWM22, Eq. (12)] for the standard formulation. The one derived in that reference is

$$\frac{d\tilde{z}}{dt} = -g(\tilde{z}) - \frac{\eta}{2}\begin{bmatrix} Q & P \\ P^\top & -R \end{bmatrix}(\tilde{z})g(\tilde{z}) \qquad \text{where} \quad \begin{bmatrix} Q & P \\ -P^\top & R \end{bmatrix}(z) := \nabla g(z)$$

while the high-resolution ODE corresponding to the symmetrized update $z^{k+1} \approx T(z^k)$ can be shown – simply by applying the high-resolution ODE for explicit Euler steps [Lu22, Eq. (20)] to the vector field $-g(z) + \frac{\eta}{2}A(z)g(z)$ – to be

$$\frac{dz}{dt} = -g(z) - \frac{\eta}{2}M(z)g(z) + \frac{\eta}{2}A(z)g(z) = -g(z) - \frac{\eta}{2}\begin{bmatrix} Q & 0 \\ 0 & R \end{bmatrix}(z)g(z).$$

There is no contradiction here, as the classical Alt-GDA iterates $\tilde{z}^k$ are not identical pointwise to the symmetrized ones $z^k$; instead the correspondence involves a time-shift of $+\frac{1}{2}$ for one of the variables. Informally, the former high-resolution ODE tracks the evolution of the Alt-GDA iterates using a piecewise-constant (forward in time) interpolation scheme, with the arbitrary choice of measuring at time-steps where $y$ was updated more recently than $x$, and the latter tracks the same iterates but using a piecewise-linear (trapezoidal) interpolation scheme.

*Proof.* As usual in this paper we let for concision $z^k = (x^k, y^k)$ for all $k \in \mathbb{N} \cup (\mathbb{N} + \frac{1}{2})$. Denote $L_{xx} = \left\| \nabla_{xx}^2 f \right\|_\infty$, $L_{yy} = \left\| \nabla_{yy}^2 f \right\|_\infty$, $L_{xy} = \left\| \nabla_{xy}^2 f \right\|_\infty$, $L_2 = L_{xx} \vee L_{yy} \vee L_{xy}$ and $L_3 = \left\| \nabla^3 f \right\|_\infty$. Throughout this proof we will write $\varepsilon$ or $\varepsilon'$ to denote elements of $[-1, 1]$ or of the unit ball of $\mathbb{R}^n$ or $\mathbb{R}^m$ or $\mathbb{R}^{n+m}$, and that may change from line to line.

Let $k \in \mathbb{N}$. By Taylor expansion with remainder in Lagrange form,

$$
\begin{aligned}
x^{k+1} - x^k &= -\eta \nabla_x f(x^k, y^{k+1/2}) \\
&= -\eta \left[ \nabla_x f(z^k) + \nabla_{xy}^2 f(z^k) \cdot (y^{k+1/2} - y^k) + \varepsilon \left( \frac{L_3}{2} \left\| y^{k+1/2} - y^k \right\|^2 \right) \right].
\end{aligned}
$$

Now

$$
y^{k+1/2} - y^k = y^{k+1/2} - \frac{y^{k+1/2} + y^{k-1/2}}{2} = \frac{y^{k+1/2} - y^{k-1/2}}{2}
$$

$$
\begin{aligned}
\text{and} \quad y^{k+1/2} - y^{k-1/2} &= \eta \nabla_y f(x^k, y^{k-1/2}) = \eta \nabla_y f(z^k) + \varepsilon \left( \eta L_{yy} \left\| y^k - y^{k-1/2} \right\| \right) \\
&= \eta \nabla_y f(z^k) + \varepsilon \left( \eta L_{yy} \frac{1}{2} \left\| y^{k+1/2} - y^{k-1/2} \right\| \right)
\end{aligned}
$$

$$
\text{since} \quad y^{k-1/2} - y^k = y^{k-1/2} - \frac{y^{k+1/2} + y^{k-1/2}}{2} = -\frac{y^{k+1/2} - y^{k-1/2}}{2} = -\left( y^{k+1/2} - y^k \right).
$$

Moreover, the above computation also shows that

$$
\left\| y^{k+1/2} - y^{k-1/2} \right\| \leq \eta \left\| \nabla_y f(z^k) \right\| + \eta L_{yy} \frac{1}{2} \left\| y^{k+1/2} - y^{k-1/2} \right\|
$$

$$
\text{and so} \quad \left\| y^{k+1/2} - y^{k-1/2} \right\| \leq \frac{1}{1 - \eta L_{yy}/2} \eta \left\| \nabla_y f(z^k) \right\| \tag{C.2}
$$

for any $\eta \leq 2/L_{yy}$. In particular, since we assume $\eta \leq 1/L_{yy}$ then $\left\| y^{k+1/2} - y^{k-1/2} \right\| \leq 2\eta \left\| \nabla_y f(z^k) \right\|$. Thus, using this last fact to express the error terms in terms of $\left\| g(z^k) \right\| = \left\| \nabla f(z^k) \right\|$, by substituting into the Taylor expansion of $x^{k+1} - x^k$ we obtain

$$
\begin{aligned}
\forall k \in \mathbb{N}, \ x^{k+1} - x^k &= -\eta \nabla_x f(z^k) - \frac{1}{2} \eta^2 \nabla_{xy}^2 f(z^k) \cdot \nabla_y f(z^k) \\
&\quad + \varepsilon \left( \eta^2 L_{xy} L_{yy} \frac{1}{2} \left\| y^{k+1/2} - y^{k-1/2} \right\| \right) + \varepsilon' \left( \eta \frac{L_3}{8} \left\| y^{k+1/2} - y^{k-1/2} \right\|^2 \right) \\
&= -\eta \nabla_x f(z^k) - \frac{1}{2} \eta^2 \nabla_{xy}^2 f(z^k) \cdot \nabla_y f(z^k) + \eta^3 \varepsilon \left( L_{xy} L_{yy} \left\| g(z^k) \right\| + \frac{L_3}{2} \left\| g(z^k) \right\|^2 \right). \tag{C.3}
\end{aligned}
$$

By the symmetric calculations, we have

$$
\begin{aligned}
\forall k \in \mathbb{N} + \frac{1}{2}, \ y^{k+1} - y^k &= \eta \nabla_y f(z^k) - \frac{1}{2} \eta^2 \nabla_{yx}^2 f(z^k) \nabla_x f(z^k) \\
&\quad + \eta^3 \varepsilon \left( L_{xy} L_{yy} \left\| g(z^k) \right\| + \frac{L_3}{2} \left\| g(z^k) \right\|^2 \right). \tag{C.4}
\end{aligned}
$$

Now let us compute an expansion for the increments between non-updating time-steps. Let $k \in \mathbb{N}$ and let us compute $y^{k+1} - y^k$:

$$
y^{k+1} - y^k = \frac{y^{k+\frac{3}{2}} + y^{k+1/2}}{2} - \frac{y^{k+1/2} + y^{k-1/2}}{2} = \frac{1}{2}\left(y^{k+\frac{3}{2}} - y^{k-1/2}\right)
$$

$$
= \frac{1}{2}\left(\eta \nabla_y f\left(x^{k+1}, y^{k+1/2}\right) + \eta \nabla_y f\left(x^k, y^{k-1/2}\right)\right)
$$

$$
= \frac{1}{2}\eta\left[2\nabla_y f(z^k) + \nabla_{yx}^2 f(z^k) \cdot (x^{k+1} - x^k) + \nabla_{yy}^2 f(z^k) \cdot (y^{k+1/2} - y^k) + \nabla_{yy}^2 f(z^k) \cdot (y^{k-1/2} - y^k)\right]
$$

$$
+ \varepsilon\left(\eta\frac{L_3}{2}\left\|x^{k+1} - x^k\right\|^2 + \eta\frac{L_3}{2}\left\|y^{k+1/2} - y^k\right\|^2\right) + \varepsilon'\left(\eta\frac{L_3}{2}\left\|y^k - y^{k-1/2}\right\|^2\right)
$$

$$
= \eta\nabla_y f(z^k) + \frac{1}{2}\eta\nabla_{yx}^2 f(z^k) \cdot (x^{k+1} - x^k) + \frac{1}{2}\eta\nabla_{yy}^2 f(z^k) \cdot \underbrace{\left(y^{k+1/2} + y^{k-1/2} - 2y^k\right)}_{=0}
$$

$$
+ \varepsilon\left(\eta\frac{L_3}{2}\left\|x^{k+1} - x^k\right\|^2 + \eta\frac{L_3}{4}\left\|y^{k+1/2} - y^{k-1/2}\right\|^2\right)
$$

$$
= \eta\nabla_y f(z^k) + \frac{1}{2}\eta\nabla_{yx}^2 f(z^k) \cdot (x^{k+1} - x^k) + \varepsilon\left(\eta^3 L_3 \left(2 + L_{xy}^2\eta^2\right)\left\|g(z^k)\right\|^2\right).
$$

Here in order to bound the error term in $\left\|x^{k+1} - x^k\right\|$ we used that

$$
\left\|x^{k+1} - x^k\right\| = \eta\left\|\nabla_x f(x^k, y^{k+1/2})\right\| \leq \eta\left\|\nabla_x f(z^k)\right\| + \eta L_{xy}\left\|y^{k+1/2} - y^k\right\|
$$

$$
= \eta\left\|\nabla_x f(z^k)\right\| + \frac{1}{2}\eta L_{xy}\left\|y^{k+1/2} - y^{k-1/2}\right\| \leq \eta(1 + L_{xy}\eta)\left\|g(z^k)\right\|
$$

$$
\left\|x^{k+1} - x^k\right\| \leq \eta^2\left(2 + 2L_{xy}^2\eta^2\right)\left\|g(z^k)\right\|^2
$$

since $\left\|y^{k+1/2} - y^{k-1/2}\right\| \leq 2\eta\left\|\nabla_y f(z^k)\right\|$ as shown previously (C.2). Hence, substituting the expansion of $x^{k+1} - x^k$ from the previous paragraph, we have

$$
\forall k \in \mathbb{N}, \; y^{k+1} - y^k = \eta\nabla_y f(z^k) + \frac{1}{2}\eta\nabla_{yx}^2 f(z^k) \cdot \left[-\eta\nabla_x f(z^k) - \frac{1}{2}\eta^2\nabla_{xy}^2 f(z^k) \cdot \nabla_y f(z^k)\right]
$$

$$
+ \eta^3\varepsilon\left(L_3\left(2 + L_{xy}^2\eta^2\right)\left\|g(z^k)\right\|^2 + L_{xy}L_{yy}\left\|g(z^k)\right\| + \frac{L_3}{2}\left\|g(z^k)\right\|^2\right)
$$

$$
= \eta\nabla_y f(z^k) - \frac{1}{2}\eta^2\nabla_{yx}^2 f(z^k) \cdot \nabla_x f(z^k) + \eta^3\varepsilon\left(L_3\left(\frac{5}{2} + L_{xy}^2\eta^2\right)\left\|g(z^k)\right\|^2 + \frac{5}{4}L_2^2\left\|g(z^k)\right\|\right).
$$
(C.5)

By the symmetric calculations, we have

$$
\forall k \in \mathbb{N} + 1/2, \; x^{k+1} - x^k = -\eta\nabla_x f(z^k) - \frac{1}{2}\eta^2\nabla_{xy}^2 f(z^k) \cdot \nabla_y f(z^k)
$$

$$
+ \eta^3\varepsilon\left(L_3\left(\frac{5}{2} + L_{xy}^2\eta^2\right)\left\|g(z^k)\right\|^2 + \frac{5}{4}L_2^2\left\|g(z^k)\right\|\right).
$$
(C.6)

The announced expansion for $z^{k+1} - z^k$, both at time-steps $k \in \mathbb{N}$ and $k \in \mathbb{N} + 1/2$, follows immediately from (C.3), (C.4), (C.5) and (C.6). $\qquad\square$

The above Prop. C.1 is already a good indication of the fact that local convergence of Alt-GDA can be analyzed via the spectrum of $\nabla\left[\mathrm{id} - \eta g + \frac{\eta^2}{2}A \cdot g\right](z^*) = I - \eta M + \frac{\eta^2}{2}AM$. In fact, since the error term in the above proposition is $O(\eta^3\left\|g(z^k)\right\|)$, one can follow the potential-based approach developed in [Lu22, Sec. 4] (via continuous-time) or [WC22, Sec. H] (for a special case), to lower-bound the decrease of $\left\|g(z^k)\right\|$ or $\left\|z^k - z^*\right\|$ at each iteration. However that approach may require $\eta$ to be smaller than actually necessary. The following proposition offers a more direct path to linking convergence of Alt-GDA to spectral properties of the aforementioned matrix.

**Proposition C.2.** *Assume $\eta \le \left\| \nabla_{xx}^2 f \right\|_\infty^{-1} \wedge \left\| \nabla_{yy}^2 f \right\|_\infty^{-1}$. Consider $z^k = (x^k, y^k)_{k \in \mathbb{N} \cup (\mathbb{N}+1/2)}$ the symmetrized Alt-GDA iterates defined by* (C.1).

*For any $k \in \mathbb{N} \cup (\mathbb{N}+1/2)$, $z^{k+1/2}$ is entirely determined by $z^k$. More precisely, there exist well-defined operators $T_{xy}^{1/2}, T_{yx}^{1/2}$ such that* $\begin{cases} \forall k \in \mathbb{N}, & z^{k+1/2} = T_{xy}^{1/2}(z^k) \\ \forall k \in \mathbb{N} + 1/2, & z^{k+1/2} = T_{yx}^{1/2}(z^k) \end{cases}$.

*Moreover, for $z^*$ such that $g(z^*) = 0$, it holds $T_{xy}^{1/2}(z^*) = T_{yx}^{1/2}(z^*) = z^*$ and*

$$\nabla \left[ T_{xy}^{1/2} \circ T_{yx}^{1/2} \right](z^*), \ \nabla \left[ T_{yx}^{1/2} \circ T_{xy}^{1/2} \right](z^*) = I - \eta M + \frac{\eta^2}{2} AM + O(\eta^3).$$

*More precisely the "$O(\cdot)$" term is a matrix with operator norm – and so spectral radius – at most $2\eta^3 \left\| A \right\| \left( \left\| A \right\| \vee \left\| S \right\| \right)^2$, where $\left\| \cdot \right\|$ denotes operator norm.*

Thanks to the above proposition, one can analyze the local convergence of the integer-time-step iterates $(z^k)_{k \in \mathbb{N}}$, say, by applying the usual analysis to its well-defined update operator $T_{yx}^{1/2} \circ T_{xy}^{1/2}$; that is to say, by bounding the spectral radius of its Jacobian at $z^*$.

For comparison, the update operator $\widetilde{T}_{xy}$ for the standard formulation is given by $\tilde{z}^{k+1} = \begin{pmatrix} \tilde{x}^k - \nabla_x f(\tilde{x}^k, \tilde{y}^k) \\ \tilde{y}^k + \nabla_y f(\tilde{x}^{k+1}, \tilde{y}^k) \end{pmatrix}$. Denoting $M = \begin{bmatrix} Q & P \\ -P^\top & R \end{bmatrix}$ as usual the Jacobian of $g$ at a fixed point $z^*$, the Jacobian of $\widetilde{T}_{xy}$ at $z^*$ writes [ZWLG22, Sec. A.3]

$$\nabla \widetilde{T}_{xy}(z^*) = \begin{bmatrix} I & 0 \\ \eta P^\top & I - \eta R \end{bmatrix} \begin{bmatrix} I - \eta Q & -\eta P \\ 0 & I \end{bmatrix} = \begin{bmatrix} I - \eta Q & -\eta P \\ \eta P^\top - \eta^2 P^\top Q & I - \eta R - \eta^2 P^\top P \end{bmatrix}$$

which cannot be written only in terms of the symmetric and antisymmetric parts of $M$. Interestingly, in order to heuristically obtain an expression for a "Jacobian" that is symmetric in the $x$ and $y$ players, a natural idea is to simply consider the average

$$\frac{\nabla T_{xy}(z^*) + \nabla T_{yx}(z^*)}{2} = \begin{bmatrix} I - \eta Q - \frac{\eta^2}{2} PP^\top & -\eta P + \frac{\eta^2}{2} PR \\ \eta P^\top - \frac{\eta^2}{2} P^\top Q & I - \eta R - \frac{\eta^2}{2} P^\top P \end{bmatrix} = I - \eta M + \frac{\eta^2}{2} AM,$$

which yields the same matrix as in Prop. C.2 (ignoring the $O(\eta^3)$ term).

*Proof.* Denote $L_{xx} = \left\| \nabla_{xx}^2 f \right\|_\infty$, $L_{yy} = \left\| \nabla_{yy}^2 f \right\|_\infty$, $L_{xy} = \left\| \nabla_{xy}^2 f \right\|_\infty$, $L_2 = L_{xx} \vee L_{yy} \vee L_{xy}$ and $L_3 = \left\| \nabla^3 f \right\|_\infty$.

Let $k \in \mathbb{N}$. We have by definition

$$\begin{cases} x^{k+1/2} = \frac{x^k - \eta \nabla_x f(x^k, y^{k+1/2}) + x^k}{2} \\ y^{k+1/2} = y^{k-1/2} + \eta \nabla_y f(x^k, y^{k-1/2}) \\ y^{k-1/2} = y^k - (y^{k+1/2} - y^k) \end{cases} \quad \text{and so} \quad \begin{cases} x^{k+1/2} = x^k - \frac{1}{2}\eta \nabla_x f(x^k, y^{k+1/2}) \\ y^{k+1/2} = 2y^k - y^{k+1/2} + \eta \nabla_y f(x^k, 2y^k - y^{k+1/2}) \end{cases}$$

$$\text{i.e.,} \quad \begin{cases} x^{k+1/2} = x^k - \frac{1}{2}\eta \nabla_x f(x^k, y^{k+1/2}) \\ y^{k+1/2} = y^k + \frac{1}{2}\eta \nabla_y f(x^k, 2y^k - y^{k+1/2}). \end{cases}$$

It remains to check that the second equation determines $y^{k+1/2}$ completely for any given $x^k$ and $y^k$. And indeed, thanks to our assumption that $\frac{\eta}{2} L_{yy} < 1$, the mapping $y \mapsto y^k + \frac{1}{2}\eta \nabla_y f(x^k, 2y^k - y)$ is a contraction, so in particular it has a unique fixed point. This shows that $z^{k+1/2}$ is entirely determined by $z^k$ for $k \in \mathbb{N}$, and the case $k \in \mathbb{N} + 1/2$ is similar.

Moreover the above also shows that $T_{xy}^{1/2}$ is characterized by

$$(x^{k+1/2}, y^{k+1/2}) = T_{xy}^{1/2}(x^k, y^k) \iff \begin{cases} x^{k+1/2} &= x^k - \frac{1}{2}\eta \nabla_x f(x^k, y^{k+1/2}) \\ y^{k+1/2} &= y^k + \frac{1}{2}\eta \nabla_y f(x^k, 2y^k - y^{k+1/2}) \end{cases}$$

and symmetrically for $T_{yx}^{1/2}$. So by substituting, we indeed find that $T_{xy}^{1/2}(x^*, y^*) = T_{yx}^{1/2}(x^*, y^*) = (x^*, y^*)$. In order to compute the Jacobian, let any small $\delta = (\delta_x, \delta_y)$, and consider $T_{xy}^{1/2}(z^* + \delta) - $

$T_{xy}^{1/2}(z^*) =: \Delta = (\Delta_x, \Delta_y)$. Then, denoting $y'$ the unique solution of $y' = y^* + \delta_y + \frac{1}{2}\eta\nabla_y f(x^* + \delta_x, 2y^* + 2\delta_y - y')$, we have

$$y' - y^* = \delta_y + \frac{\eta}{2}\left[\nabla_y f(x^* + \delta_x, 2y^* + 2\delta_y - y') - \nabla_y f(z^*)\right]$$

$$\|y' - y^*\| \le \|\delta_y\| + \frac{\eta}{2}\left[L_{xy}\|\delta_x\| + L_{yy}\left(2\|\delta_y\| + \|y^* - y'\|\right)\right]$$

$$\|y' - y^*\| \le \frac{1}{1 - \eta L_{yy}/2}\cdot\left(\|\delta_y\| + \frac{\eta}{2}\left[L_{xy}\|\delta_x\| + L_{yy}2\|\delta_y\|\right]\right) = O\left(\|\delta_x\| + \|\delta_y\|\right)$$

since we assume that $\eta L_{yy}/2 \le 1/2$, and so

$$y' - y^* = \delta_y + \frac{\eta}{2}\left[\nabla_{yx}^2 f(z^*)\delta_x + \nabla_{yy}^2 f(z^*)(2\delta_y + y^* - y') + O\left(\|\delta_x\|^2 + \|\delta_y\|^2\right)\right]$$

$$y' - y^* = \left[I + \frac{\eta}{2}\nabla_{yy}^2 f(z^*)\right]^{-1}\left(\delta_y + \frac{\eta}{2}\left[\nabla_{yx}^2 f(z^*)\delta_x + 2\nabla_{yy}^2 f(z^*)\delta_y\right]\right) + O\left(\|\delta_x\|^2 + \|\delta_y\|^2\right).$$

This directly gives an expansion for $\Delta_y = y' - y^*$, and we have

$$\Delta_x = \delta_x - \frac{1}{2}\eta\left[\nabla_x f(x^* + \delta_x, y') - \nabla_x f(z^*)\right]$$

$$= \delta_x - \frac{1}{2}\eta\left[\nabla_{xx}^2 f(z^*)\delta_x + \nabla_{xy}^2 f(z^*)(y' - y^*)\right] + O(\|\delta_x\|^2 + \|\delta_y\|^2)$$

$$= \delta_x - \frac{\eta}{2}\nabla_{xx}^2 f(z^*)\delta_x - \frac{\eta}{2}\nabla_{xy}^2 f(z^*)\left[I + \frac{\eta}{2}\nabla_{yy}^2 f(z^*)\right]^{-1}\left(\delta_y + \frac{\eta}{2}\left[\nabla_{yx}^2 f(z^*)\delta_x + 2\nabla_{yy}^2 f(z^*)\delta_y\right]\right)$$

$$\qquad + O(\|\delta_x\|^2 + \|\delta_y\|^2)$$

$$= \left\{I - \frac{\eta}{2}\nabla_{xx}^2 f(z^*) - \frac{\eta^2}{4}\nabla_{xy}^2 f(z^*)\left[I + \frac{\eta}{2}\nabla_{yy}^2 f(z^*)\right]^{-1}\nabla_{yx}^2 f(z^*)\right\}\delta_x$$

$$\qquad + \left\{-\frac{\eta}{2}\nabla_{xy}^2 f(z^*)\left[I + \frac{\eta}{2}\nabla_{yy}^2 f(z^*)\right]^{-1}\left(I + \eta\nabla_{yy}^2 f(z^*)\right)\right\}\delta_y + O(\|\delta_x\|^2 + \|\delta_y\|^2).$$

Writing for concision $\begin{bmatrix}\nabla_{xx}^2 f & \nabla_{xy}^2 f \\ \nabla_{yx}^2 f & \nabla_{yy}^2 f\end{bmatrix}(z^*) = \begin{bmatrix}Q & P \\ P^\top & -R\end{bmatrix}$ as usual in this paper, the above expansions write

$$\Delta_x = \left\{I - \frac{\eta}{2}Q - \frac{\eta^2}{4}P\left[I - \frac{\eta}{2}R\right]^{-1}P^\top\right\}\delta_x + \left\{-\frac{\eta}{2}P\left[I - \frac{\eta}{2}R\right]^{-1}(I - \eta R)\right\}\delta_y + O(\|\delta_x\|^2 + \|\delta_y\|^2)$$

$$\Delta_y = \left\{\left[I - \frac{\eta}{2}R\right]^{-1}\frac{\eta}{2}P^\top\right\}\delta_x + \left\{\left[I - \frac{\eta}{2}R\right]^{-1}(I - \eta R)\right\}\delta_y + O\left(\|\delta_x\|^2 + \|\delta_y\|^2\right)$$

which implies by definition of the Jacobian that

$$\nabla T_{xy}^{1/2}(z^*) = \begin{bmatrix}I - \frac{\eta}{2}Q - \frac{\eta^2}{4}P\left[I - \frac{\eta}{2}R\right]^{-1}P^\top & -\frac{\eta}{2}P\left[I - \frac{\eta}{2}R\right]^{-1}(I - \eta R) \\ \left[I - \frac{\eta}{2}R\right]^{-1}\frac{\eta}{2}P^\top & \left[I - \frac{\eta}{2}R\right]^{-1}(I - \eta R)\end{bmatrix}.$$

By symmetry, the Jacobian for $T_{yx}^{1/2}$ can be obtained from the above expression by swapping $\eta$ for $-\eta$, $Q$ for $-R$, $P^\top$ for $P$, and swapping the lines resp. columns of the block matrix, yielding

$$\nabla T_{yx}^{1/2}(z^*) = \begin{bmatrix}\left[I - \frac{\eta}{2}Q\right]^{-1}(I - \eta Q) & -\left[I - \frac{\eta}{2}Q\right]^{-1}\frac{\eta}{2}P \\ \frac{\eta}{2}P^\top\left[I - \frac{\eta}{2}Q\right]^{-1}(I - \eta Q) & I - \frac{\eta}{2}R - \frac{\eta^2}{4}P^\top\left[I - \frac{\eta}{2}Q\right]^{-1}P\end{bmatrix}.$$

By straightforward but tedious computations, and noting that $\left[I - \frac{\eta}{2}R\right]^{-1}$ commutes with $R$, we arrive at the following expression for $\nabla\left[T_{xy}^{1/2}\circ T_{yx}^{1/2}\right](z^*) = \nabla T_{xy}^{1/2}(z^*)\cdot\nabla T_{yx}^{1/2}(z^*)$ (since $z^*$ is a fixed point of both operators):

$$\nabla\left[T_{xy}^{1/2}\circ T_{yx}^{1/2}\right](z^*)$$

$$= \begin{bmatrix}I - \eta Q - \frac{\eta^2}{2}PP^\top\left[I - \frac{\eta}{2}Q\right]^{-1}(I - \eta Q) & -\eta P + \frac{\eta^2}{2}PR + \frac{\eta^3}{4}PP^\top\left[I - \frac{\eta}{2}Q\right]^{-1}P \\ \eta P^\top\left[I - \frac{\eta}{2}Q\right]^{-1}(I - \eta Q) & I - \eta R - \frac{\eta^2}{2}P^\top\left[I - \frac{\eta}{2}Q\right]^{-1}P\end{bmatrix}$$

$$= \begin{bmatrix}I - \eta Q - \frac{\eta^2}{2}PP^\top & -\eta P + \frac{\eta^2}{2}PR \\ \eta P^\top - \frac{\eta^2}{2}P^\top Q & I - \eta R - \frac{\eta^2}{2}P^\top P\end{bmatrix} + \boldsymbol{E} = I - \eta M + \frac{\eta^2}{2}AM + \boldsymbol{E}$$

where the error term $\boldsymbol{E}$ is

$$\boldsymbol{E} = \begin{bmatrix} -\frac{\eta^2}{2}PP^\top \left\{ [I - \frac{\eta}{2}Q]^{-1}(I - \eta Q) - I \right\} & \frac{\eta^3}{4}PP^\top [I - \frac{\eta}{2}Q]^{-1}P \\ \eta P^\top \left\{ [I - \frac{\eta}{2}Q]^{-1}(I - \eta Q) - (I - \frac{\eta}{2}Q) \right\} & -\frac{\eta^2}{2}P^\top \left\{ [I - \frac{\eta}{2}Q]^{-1} - I \right\}P \end{bmatrix} = O(\eta^3)$$

thanks to our assumption that $\frac{\eta}{2}L_{xx} < 1$. More precisely, by bounding the operator norm of each block and summing the bounds, and using that $\left\| [I - \frac{\eta}{2}Q]^{-1} \right\| \le \sum_{k=0}^\infty \left\| \frac{\eta}{2}Q \right\|^k = \frac{1}{1 - \eta\|Q\|/2} \le \frac{1}{1 - \eta L_{xx}/2} \le 2$, one can check that

$$\|R\| \le \left\| [I - \frac{\eta}{2}Q]^{-1} \right\| \left( \frac{\eta^2}{2}\|P\|^2 \cdot \frac{\eta}{2}\|Q\| + \frac{\eta^3}{4}\|P\|^3 + \eta\|P\| \cdot \frac{\eta^2}{4}\|Q\|^2 + \frac{\eta^2}{2}\|P\|^2 \cdot \frac{\eta}{2}\|Q\| \right)$$

$$\le \frac{1}{1 - \eta L_{xx}/2} \cdot \eta^3 \|P\|(\|P\| \vee \|Q\|)^2 \le 2\eta^3\|P\|(\|P\| \vee \|Q\|)^2.$$

Note that by definition of $A = \begin{bmatrix} 0 & P \\ -P^\top & 0 \end{bmatrix}$ and $S = \begin{bmatrix} Q & 0 \\ 0 & R \end{bmatrix}$, we have $\|A\| = \|P\|$ and $\|S\| = \|Q\| \vee \|R\|$.

This proves the claimed expansion for $\nabla \left[ T_{xy}^{1/2} \circ T_{yx}^{1/2} \right](z^*)$, and the expansion for $\nabla \left[ T_{yx}^{1/2} \circ T_{xy}^{1/2} \right](z^*)$ follows by symmetry. $\qquad\square$

# D   Details for Sec. 3.2

In this section we prove the expansions of the convergence rates of discrete-time algorithms (with non-asymptotic bounds in $\eta$ and $\alpha$) reported in Table 1. That is, we derive approximate expressions for $\rho(\nabla T(z^*))$ the spectral radius of the update operator's Jacobian at optimum, for various update rules (GDA, EG, etc.), when the skewed gradient field's Jacobian $M$ has a small symmetric part.

We will repeatedly make use of the following estimate for the spectrum of a perturbed normal matrix. The expansion itself is a special case of [Tao08, Eqs. (5) and (7)] for $M_0$ normal, explicitly pointed out in that reference. Deriving the explicit bound on the error term involves rather tedious calculations however, so the proof is deferred to Sec. D.5.

**Proposition D.1.** *Let $M_0, M_1, M_2 \in \mathbb{R}^{d \times d}$ or $\mathbb{C}^{d \times d}$ and $M_\alpha = M_0 + \alpha M_1 + \frac{\alpha^2}{2}M_2$. Assume $M_0$ is normal and has distinct eigenvalues; denote its eigenvalue decomposition as $M_0 w_j = \lambda_j w_j$ with $(w_j)_j$ unitary, and let $\gamma_0 = \min_{k \ne j} |\lambda_k - \lambda_j|$. Then for all $\alpha$ such that $\left\| \alpha M_1 + \frac{\alpha^2}{2}M_2 \right\| \le \frac{\gamma_0}{4\sqrt{2d}}$,*

$$\mathrm{Sp}(M_\alpha) = \left\{ \lambda_j + \alpha \overline{w}_j^\top M_1 w_j + \frac{\alpha^2}{2}\overline{w}_j^\top M_2 w_j + \alpha^2 \sum_{k \ne j} \frac{\left( \overline{w}_j^\top M_1 w_k \right)\left( \overline{w}_k^\top M_1 w_j \right)}{\lambda_j - \lambda_k} + \boldsymbol{r}_j, \ 1 \le j \le d \right\}$$

*where $\|\cdot\|$ denotes operator norm and for each $j$, $|\boldsymbol{r}_j| \le \alpha^3 \cdot 8d\gamma_0^{-1}\|M_1\| \left( \|M_2\| + 4d\gamma_0^{-1}\|M_1\|^2 \right)$.*

In the remainder of this section, consider $S$ real symmetric and $A$ real antisymmetric in $\mathbb{R}^{d \times d}$, and let $M = M_\alpha = \alpha S + A$. Assume $A$ has simple eigenvalues and denote its eigenvalue decomposition as $A w_j = i\sigma_j w_j$ with $\sigma_j \in \mathbb{R}$. This corresponds exactly to Sec. 3 with the correspondence $d := n + n$ and $\mathrm{Sp}(A) = \{i\sigma_j, j \le d\} := \{is\tilde{\sigma}_j, s \in \{-1, 1\}, j \le n\}$ with $(\tilde{\sigma}_j)_{j \le n}$ the singular values of $P$. Furthermore, let for concision $\gamma_A = \min_{k \ne j} |\sigma_k - \sigma_j|$ and $L_S = \|S\|, L_A = \|A\|$.

Throughout the derivations of this section, we will write $\varepsilon, \varepsilon', \varepsilon_j$ or $\varepsilon'_j$ to denote elements of $[-1, 1]$ or of the unit ball of $\mathbb{R}^d$, and that may change from line to line. Likewise, $\zeta, \zeta', \zeta_j, \zeta'_j$ will denote elements of $\{z \in \mathbb{C}; |z| \le 1\}$ or of the unit ball of $\mathbb{C}^d$, and that may change from line to line.

## D.1   Simultaneous GDA

The Sim-GDA update rule is $z^{k+1} = T(z^k) = z^k - \eta g(z^k)$, so the update operator's Jacobian at optimum is

$$\nabla T(z^*) = I - \eta M = (I - \eta A) - \alpha \eta S.$$

Observe that $I - \eta A$ is normal with eigenvalue/eigenvector pairs $(1 - i\eta\sigma_j, w_j)$. Hence let us apply Prop. D.1 with $M_0 = I - \eta A$ and $M_1 = -\eta S$ (and $M_2 = 0$). We get that

$$\forall \alpha \leq \frac{1}{\eta L_S} \frac{\eta \gamma_A}{4\sqrt{2d}} = \frac{\gamma_A}{L_S \cdot 4\sqrt{2d}},$$

$$\mathrm{Sp}(\nabla T(z^*)) = \left\{ 1 - \eta i \sigma_j - \alpha\eta\overline{w}_j^\top S w_j + \alpha^2 \sum_{k \neq j} \frac{\eta^2 \left|\overline{w}_j^\top S w_k\right|^2}{-i\eta\sigma_j + i\eta\sigma_k} + \alpha^3\eta \cdot \zeta_j \left( 32 d^2 \gamma_A^{-2} L_S^3 \right), 1 \leq j \leq d \right\}.$$

It seems unlikely that the term in $\alpha^2$ will ever lead to friendly expressions, so we will bound it uniformly; let us nonetheless note that it is pure imaginary. Namely, since

$$\left| \alpha^2 \sum_{k \neq j} \frac{\eta^2 \left|\overline{w}_j^\top S w_k\right|^2}{-i\eta\sigma_j + i\eta\sigma_k} \right| \leq \alpha^2\eta \sum_{k \neq j} \frac{\left|\overline{w}_k^\top S w_j\right|^2}{\left|\sigma_k - \sigma_j\right|} \leq \alpha^2\eta\gamma_A^{-1} \underbrace{\sum_k \left|\overline{w}_k^\top S w_j\right|^2}_{=\|S w_j\|^2 \leq L_S^2}, \qquad \text{(D.1)}$$

then we may write that term as $i\alpha^2\eta \cdot \varepsilon_j \left( \gamma_A^{-1} L_s^2 \right)$. For the spectral radius we get

$$\rho(\nabla T(z^*))^2 = \max_{\lambda \in \mathrm{Sp}(\nabla T(z^*))} |\lambda|^2$$

$$= \max_{j \leq d} \left| 1 - \alpha\eta\overline{w}_j^\top S w_j + \alpha^3\eta \cdot \varepsilon_j' 32 d^2 \gamma_A^{-2} L_S^3 \right|^2 + \left| \eta\sigma_j + \alpha^2\eta \cdot \varepsilon_j \gamma_A^{-1} L_S^2 + \alpha^3\eta \cdot \varepsilon_j'' 32 d^2 \gamma_A^{-2} L_S^3 \right|^2$$

$$= \max_{j \leq d} 1 - 2\alpha\eta \left( \overline{w}_j^\top S w_j \right) + \eta^2\sigma_j^2 + O(\alpha^3\eta + \alpha^2\eta^2)$$

and more precisely one can check that the $O(\cdot)$ term is absolutely bounded by $\alpha^3\eta \cdot 128 d^2 \gamma_A^{-2} L_S^3 (1 + 5\eta L_A) + \alpha^2\eta^2 \cdot 2\gamma_A^{-1} L_S^2 L_A$, for all $\alpha \leq \frac{\gamma_A}{L_S \cdot 4\sqrt{2d}}$.

## D.2 Proximal Point

The PP update rule is $z^{k+1} = T(z^k)$ with $T^{-1}(z) = z + \eta g(z)$ so the update operator's Jacobian at optimum is $\nabla T(z^*) = (I + \eta M)^{-1}$. So $\mathrm{Sp}(\nabla T(z^*)) = \left\{ \lambda^{-1}, \lambda \in \mathrm{Sp}(I + \eta M) \right\}$ and so, by the exact same calculations as for Sim-GDA with $\eta$ replaced by $-\eta$,

$$\rho(\nabla T(z^*))^2 = \left[ \min_{j \leq d} 1 + 2\alpha\eta \left( \overline{w}_j^\top S w_j \right) + \eta^2\sigma_j^2 + O(\alpha^3\eta + \alpha^2\eta^2) \right]^{-1}$$

for all $\alpha \leq \frac{\gamma_A}{L_S \cdot 4\sqrt{2d}}$, and we have the same absolute bound on the $O(\cdot)$ term as for Sim-GDA.

## D.3 Alternating GDA

As we show in App. C, for any $\eta \leq \left\| \nabla_{xx}^2 f \right\|_\infty^{-1} \wedge \left\| \nabla_{yy}^2 f \right\|_\infty^{-1}$, the (symmetrized) Alt-GDA update operator's Jacobian at optimum is

$$\nabla T(z^*) = \nabla \overline{T}(z^*) + \boldsymbol{E} \quad \text{for some } \boldsymbol{E} \in \mathbb{R}^{d \times d} \text{ with } \|\boldsymbol{E}\| \leq 2\eta^3 L_A (L_A \vee \alpha L_S)^2,$$

$$\text{where } \nabla \overline{T}(z^*) = I - \eta M + \frac{\eta^2}{2} AM = \left( I - \eta A + \frac{\eta^2}{2} A^2 \right) + \alpha \left( -\eta S + \frac{\eta^2}{2} AS \right).$$

Observe that $I - \eta A + \frac{\eta^2}{2} A^2$ is normal with eigenvalue/eigenvector pairs $\left( 1 - i\eta\sigma_j - \frac{\eta^2}{2}\sigma_j^2, w_j \right)$. Hence let us apply Prop. D.1 with $M_0 = I - \eta A + \frac{\eta^2}{2} A^2$ and $M_1 = -\eta S + \frac{\eta^2}{2} AS$ (and $M_2 = 0$). Further observe that

$$\overline{w}_j^\top A S w_k = \overline{w}_j^\top \left( \sum_l i\sigma_l w_l \overline{w}_l^\top \right) S w_k = i\sigma_j \cdot \overline{w}_j^\top S w_k.$$

So we get that, for all $\alpha \leq \frac{\gamma_A}{4\sqrt{2d}\cdot L_s\left(1+\frac{\eta}{2}L_A\right)}$,

$$\mathrm{Sp}(\nabla\overline{T}(z^*))$$

$$= \left\{ 1 - \eta i\sigma_j - \frac{\eta^2}{2}\sigma_j^2 + \alpha\overline{w}_j^\top\left(-\eta S + \frac{\eta^2}{2}AS\right)w_j + \alpha^2\sum_{k\neq j}\frac{\eta^2\left(\overline{w}_j^\top\left(S - \frac{\eta}{2}AS\right)w_k\right)\left(\overline{w}_k^\top\left(S - \frac{\eta}{2}AS\right)w_j\right)}{-i\eta\sigma_j + i\eta\sigma_k + \frac{\eta^2}{2}(-\sigma_j^2+\sigma_k^2)}\right.$$

$$\left. + \alpha^3\eta\cdot\boldsymbol{\zeta}_j\left(32d^2\gamma_A^{-2}L_S^3\left(1+\frac{\eta}{2}L_A\right)^3\right), 1\leq j\leq d\right\}$$

$$= \left\{ 1 - \eta i\sigma_j - \frac{\eta^2}{2}\sigma_j^2 - \alpha\eta\left(\overline{w}_j^\top Sw_j\right) + \alpha\frac{\eta^2}{2}i\sigma_j\cdot\left(\overline{w}_j^\top Sw_j\right) + i\cdot\alpha^2\eta\cdot\boldsymbol{\varepsilon}_j\left(\gamma_A^{-1}L_S^2\right)\right.$$

$$\left. + \alpha^3\eta\cdot\boldsymbol{\zeta}_j\left(32d^2\gamma_A^{-2}L_S^3\left(1+\frac{\eta}{2}L_A\right)^3\right) + \alpha^2\eta^3\cdot\boldsymbol{\zeta}_j'\left(\gamma_A^{-1}L_S^2L_A^2/4\right), 1\leq j\leq d\right\}.$$

Here for the second equality, we computed the coefficient for the term in $\alpha^2$ as

$$\sum_{k\neq j}\frac{\eta^2\left(\overline{w}_j^\top\left(S-\frac{\eta}{2}AS\right)w_k\right)\left(\overline{w}_k^\top\left(S-\frac{\eta}{2}AS\right)w_j\right)}{-i\eta\sigma_j + i\eta\sigma_k + \frac{\eta^2}{2}(-\sigma_j^2+\sigma_k^2)}$$

$$= \sum_{k\neq j}\eta\frac{\left(\overline{w}_j^\top Sw_k - \frac{\eta}{2}i\sigma_j\overline{w}_j^\top Sw_k\right)\left(\overline{w}_k^\top Sw_j - \frac{\eta}{2}i\sigma_k\overline{w}_k^\top Sw_j\right)}{i(\sigma_k-\sigma_j)+\frac{\eta}{2}(\sigma_k^2-\sigma_j^2)} = \sum_{k\neq j}\eta\left|\overline{w}_j^\top Sw_k\right|^2\frac{\left(1-\frac{\eta}{2}i\sigma_j\right)\left(1-\frac{\eta}{2}i\sigma_k\right)}{i(\sigma_k-\sigma_j)\left(1-i\frac{\eta}{2}(\sigma_k+\sigma_j)\right)}$$

$$= \sum_{k\neq j}\frac{\eta\left|\overline{w}_j^\top Sw_k\right|^2}{i(\sigma_k-\sigma_j)}\left(1+\frac{-\frac{\eta^2}{4}\sigma_j\sigma_k}{1-i\frac{\eta}{2}(\sigma_k+\sigma_j)}\right) = i\eta\cdot\boldsymbol{\varepsilon}_j(\gamma_A^{-1}L_S^2) + \eta^3\cdot\boldsymbol{\zeta}_j'\left(\gamma_A^{-1}L_S^2\cdot L_A^2/4\right)$$

where the last equality follows from the same bound as in (D.1). For the spectral radius we get

$$\rho(\nabla\overline{T}(z^*))^2 = \max_{j\leq d}\left|1-\frac{\eta^2}{2}\sigma_j^2 - \alpha\eta\left(\overline{w}_j^\top Sw_j\right)\right|^2 + \left|-\eta\sigma_j + \alpha\frac{\eta^2}{2}\sigma_j\cdot\left(\overline{w}_j^\top Sw_j\right) + O(\alpha^2\eta)\right|^2$$

$$+ O(\alpha^3\eta + \alpha^2\eta^3)$$

$$= \max_{j\leq d}1 + \frac{\eta^4}{4}\sigma_j^4 - \eta^2\sigma_j^2 - 2\alpha\eta\left(\overline{w}_j^\top Sw_j\right) + \alpha\eta^3\sigma_j^2\left(\overline{w}_j^\top Sw_j\right)$$

$$+ \eta^2\sigma_j^2 - \alpha\eta^3\sigma_j^2\left(\overline{w}_j^\top Sw_j\right) + O(\alpha^3\eta + \alpha^2\eta^2)$$

$$= \max_{j\leq d}1 - 2\alpha\eta\left(\overline{w}_j^\top Sw_j\right) + \frac{\eta^4}{4}\sigma_j^4 + O(\alpha^3\eta+\alpha^2\eta^2).$$

The two cancellations that occur in the last line (of $\pm\eta^2\sigma_j^2$ and of $\pm\alpha\eta^3\sigma_j^2\left(\overline{w}_j^\top Sw_j\right)$) are consistent with the intuition that Alt-GDA is a good symplectic integrator. More precisely, one can check that the final $O(\cdot)$ term is absolutely bounded by $\alpha^3\eta\cdot512d^2\gamma_A^{-2}L_S^3(1+\eta L_A)^6+\alpha^2\eta^2\cdot4d\gamma_A^{-1}L_S^2L_A(1+\eta L_A)^4$, for all $\alpha\leq\frac{\gamma_A}{4\sqrt{2d}\cdot L_s\left(1+\frac{\eta}{2}L_A\right)}$.

Finally, by smoothness of the eigenvalues of perturbed matrices (Lemma D.2),

$$\rho(\nabla T(z^*))^2 = \rho(\nabla\overline{T}(z^*)+\boldsymbol{E})^2 = \rho(\nabla\overline{T}(z^*)^2 + O(\eta^3).$$

One could further derive explicit bounds on the $O(\eta^3)$ term in that last expression by controlling the distance to normality of $\nabla\overline{T}(z^*)$ – since it is continuous in $\alpha$ and normal for $\alpha=0$ – and adapting the proof of Lemma D.5 to matrices that are close to normal.

### D.4  Extra-Gradient

The EG update rule is $z^{k+1} = T(z^k) = z^k - \eta g(z^k - \eta g(z^k))$ so the update operator's Jacobian at optimum is

$$\nabla T(z^*) = I - \eta M(I-\eta M) = I - \eta A - \alpha\eta S + \eta^2 M^2$$

$$= \left(I - \eta A + \eta^2 A^2\right) + \alpha\left(-\eta S + \eta^2(AS+SA)\right) + \frac{1}{2}\alpha^2\left(2\eta^2 S^2\right).$$

Similarly as in the previous subsection, observe that $I - \eta A + \eta^2 A^2$ is normal with eigenvalue/eigenvector pairs $\left(1 - i\eta\sigma_j - \eta^2\sigma_j^2, w_j\right)$, and that

$$\overline{w}_j^\top (AS + SA) w_k = i(\sigma_j + \sigma_k) \cdot \overline{w}_j^\top S w_k.$$

Let us apply Prop. D.1 to $M_0 = I - \eta A + \eta^2 A^2$, $M_1 = -\eta S + \eta^2(AS + SA)$ and $M_2 = 2\eta^2 S^2$. The proposition yields a bound for all $\alpha$ such that $\alpha \left\| \left| S - \eta(AS + SA) + \alpha\eta S^2 \right| \right\| \leq \frac{\gamma_A}{4\sqrt{2d}}$, for which a simpler sufficient condition is $\alpha \leq 1 \wedge \frac{\gamma_A}{4\sqrt{2d} \cdot L_S(1 + \eta L_S(1 + 2\eta L_A))}$. Using that $\|\|M_1\|\| \leq \eta L_S(1 + 2\eta L_A)$ to upper-bound the term in $\alpha^3$, we get that for all such $\alpha$,

$$
\begin{aligned}
\mathrm{Sp}(\nabla T(z^*)) = \Bigg\{ &1 - i\eta\sigma_j - \eta^2\sigma_j^2 + \alpha\left(-\eta\overline{w}_j^\top S w_j + \eta^2 2i\sigma_j \overline{w}_j^\top S w_j\right) + \alpha^2\eta^2 \overline{w}_j^\top S^2 w_j \\
&+ \alpha^2 \sum_{k \neq j} \frac{\eta^2 \overline{w}_j^\top (S - \eta(AS + SA)) w_k \cdot \overline{w}_k^\top (S - \eta(AS + SA)) w_j}{-i\eta\sigma_j + i\eta\sigma_k + \eta^2(-\sigma_j^2 + \sigma_k^2)} \\
&+ \alpha^3 \eta \cdot \boldsymbol{\zeta}_j \cdot 8d\gamma_A^{-1} L_S(1 + 2\eta L_A)\left(2\eta L_S^2 + 4d\gamma_A^{-1} L_S^2(1 + 2\eta L_A)^2\right), 1 \leq j \leq d \Bigg\} \\
= \Bigg\{ &1 - i\eta\sigma_j - \eta^2\sigma_j^2 - \alpha\eta\left(\overline{w}_j^\top S w_j\right) + 2i\alpha\eta^2\sigma_j\left(\overline{w}_j^\top S w_j\right) \\
&+ i \cdot \alpha^2\eta \cdot \boldsymbol{\varepsilon}_j\left(\gamma_A^{-1} L_S^2\right) + \alpha^2\eta^2 \cdot \boldsymbol{\varepsilon}_j'\left(2\gamma_A^{-1} L_S^2 L_A\right) + \alpha^2\eta^2 \overline{w}_j^\top S^2 w_j \\
&+ \alpha^3 \eta \cdot \boldsymbol{\zeta}_j\left(512d^2\gamma_A^{-2} L_S^3(1 + \eta L_A)^3\right), 1 \leq j \leq d \Bigg\}.
\end{aligned}
$$

Here for the second equality we computed the terms in $\alpha^2$ as

$$
\begin{aligned}
&\alpha^2 \sum_{k \neq j} \frac{\eta^2 \overline{w}_j^\top (S - \eta(AS + SA)) w_k \cdot \overline{w}_k^\top (S - \eta(AS + SA)) w_j}{-i\eta\sigma_j + i\eta\sigma_k + \eta^2(-\sigma_j^2 + \sigma_k^2)} \\
&= \alpha^2 \sum_{k \neq j} \eta \frac{\left(\overline{w}_j^\top S w_k\right)(1 - i\eta(\sigma_j + \sigma_k)) \cdot \left(\overline{w}_k^\top S w_j\right)(1 - i\eta(\sigma_j + \sigma_k))}{i(\sigma_k - \sigma_j)(1 - i\eta(\sigma_k + \sigma_j))} \\
&= \alpha^2 \sum_{k \neq j} \frac{\eta \left|\overline{w}_j^\top S w_k\right|^2}{i(\sigma_k - \sigma_j)}(1 - i\eta(\sigma_k + \sigma_j)) = i \cdot \alpha^2\eta \cdot \boldsymbol{\varepsilon}_j\left(\gamma_A^{-1} L_S^2\right) + \alpha^2\eta^2 \cdot \boldsymbol{\varepsilon}_j'\left(\gamma_A^{-1} L_S^2 \cdot 2L_A\right)
\end{aligned}
$$

where the last equality follows from the same bound as in (D.1). For the spectral radius we get

$$
\begin{aligned}
\rho(\nabla T(z^*))^2 &= \max_{j \leq d} \left|1 - \eta^2\sigma_j^2 - \alpha\eta\left(\overline{w}_j^\top S w_j\right) + O(\alpha^2\eta^2)\right|^2 + \left|-\eta\sigma_j + 2\alpha\eta^2\sigma_j\left(\overline{w}_j^\top S w_j\right) + O(\alpha^2\eta)\right|^2 \\
&\quad + O(\alpha^3\eta) \\
&= \max_{j \leq d} 1 + \eta^4\sigma_j^4 - 2\eta^2\sigma_j^2 - 2\alpha\eta\left(\overline{w}_j^\top S w_j\right) + 2\alpha\eta^3\sigma_j^2\left(\overline{w}_j^\top S w_j\right) \\
&\quad + \eta^2\sigma_j^2 - 4\alpha\eta^3\sigma_j^2\left(\overline{w}_j^\top S w_j\right) + O(\alpha^3\eta + \alpha^2\eta^2) \\
&= \max_{j \leq d} 1 - 2\alpha\eta\left(\overline{w}_j^\top S w_j\right) - \eta^2\sigma_j^2 - 2\alpha\eta^3\sigma_j^2\left(\overline{w}_j^\top S w_j\right) + \eta^4\sigma_j^4 + O(\alpha^3\eta + \alpha^2\eta^2)
\end{aligned}
$$

and more precisely one can check that the final $O(\cdot)$ term is absolutely bounded by $\alpha^3\eta \cdot 2^{12}d^2\gamma_A^{-2} L_S^3(1 + \eta L_A)^7 + \alpha^2\eta^2 \cdot 15d\gamma_A^{-1} L_S^2 L_A(1 + \eta L_A)^2$.

### D.5 Proof of Prop. D.1

The following lemma summarizes the eigenvalue derivative formulas up to order 2 for perturbed matrices with distinct eigenvalues.

**Lemma D.2** ([Tao08])**.** *Let $M_\alpha \in \mathbb{C}^{d \times d}$ for $\alpha \in (-1, 1)$ a smooth curve of matrices such that $M_0$ has distinct eigenvalues. Then there exists an open interval $I \ni 0$ such that $M_\alpha$ has distinct eigenvalues*

*for all $\alpha \in I$; in particular they are diagonalizable. Denote their eigenvalue decompositions as $M_\alpha = \sum_{k=1}^{d} \lambda_k(\alpha) v_k(\alpha) \overline{w}_k(\alpha)^\top$; that is, the eigenvalues of $M_\alpha$ are $(\lambda_k(\alpha))_k$, the associated eigenvectors are $(v_k(\alpha))_k$, and $(w_k(\alpha))_k$ is a dual basis to the basis of eigenvectors, i.e., $\overline{w}_k(\alpha)^\top v_j(\alpha) = \mathbb{1}_{j=k}$.*

*The eigenvalues of $M_\alpha$ are smooth over $I$ and their first two derivatives at any $\alpha \in I$ are given by, using $\dot{\bullet}$ to denote differentiation w.r.t. $\alpha$ and leaving the dependency on $\alpha$ implicit,*

$$\dot{\lambda}_k = w_k^* \dot{M} v_k \quad and \quad \ddot{\lambda}_k = w_k^* \ddot{M} v_k + 2 \sum_{j \neq k} \frac{(w_k^* \dot{M} v_j)(w_j^* \dot{M} v_k)}{\lambda_k - \lambda_j}$$

*where we denoted $w_k^* = \overline{w}_k^\top$. Furthermore the eigenvectors $v_k$ and dual basis vectors $w_k$ can also be chosen smooth and their derivatives at any $\alpha \in I$ are given by*

$$\dot{v}_k = \sum_{j \neq k} \frac{w_j^* \dot{M} v_k}{\lambda_k - \lambda_j} v_j + c_k v_k \quad and \quad \dot{w}_k^* = \sum_{j \neq k} \frac{w_k^* \dot{M} v_j}{\lambda_k - \lambda_j} w_j^* - c_k w_k^*$$

*for some scalars $c_k$ that reflect the normalization of the eigenvectors.*

By applying a Taylor expansion with remainder in Lagrange form to the eigenvalues $\lambda_k(\alpha)$ of the matrix $M_\alpha = M_0 + \alpha M_1 + \frac{\alpha^2}{2} M_2$ of [Prop. D.1](#) – since the eigenvalues are smooth by the above lemma –, we have that for all $\alpha$ in some neighborhood of zero,

$$\lambda_k(\alpha) = \lambda_k(0) + \alpha \dot{\lambda}_k(0) + \frac{\alpha^2}{2} \ddot{\lambda}_k(0) + \frac{\alpha^3}{6} \dddot{\lambda}_k(\xi) \quad \text{for some } 0 < \xi < \alpha. \tag{D.2}$$

By substituting the expressions from [Lemma D.2](#) for the first two eigenvalue derivatives, we already get the terms in $\alpha$ and $\alpha^2$ in [Prop. D.1](#). In order to control the last term in $\alpha^3$, we need to compute the third eigenvalue derivatives $\dddot{\lambda}_k(\xi)$. Note that $M_\xi$ is never assumed normal for any $\xi > 0$, which is why we do not use normality in [Lemma D.2](#) nor in [Lemma D.3](#) below.

**Lemma D.3.** *Under the conditions of [Lemma D.2](#),*

$$\dddot{\lambda}_k = w_k^* \dddot{M} v_k + 3 \sum_{j \neq k} \frac{1}{\lambda_k - \lambda_j} \left[ (w_k^* \dot{M} v_j)(w_j^* \ddot{M} v_k) + (w_k^* \ddot{M} v_j)(w_j^* \dot{M} v_k) \right]$$

$$+ 6 \sum_{j,l \neq k} \frac{(w_k^* \dot{M} v_j)(w_j^* \dot{M} v_l)(w_l^* \dot{M} v_k)}{(\lambda_k - \lambda_j)(\lambda_k - \lambda_l)} - 6 \sum_{j \neq k} \frac{(w_k^* \dot{M} v_j)(w_j^* \dot{M} v_k)}{(\lambda_k - \lambda_j)^2} w_k^* \dot{M} v_k.$$

*In particular,*

$$\max_k \left| \dddot{\lambda}_k \right| \leq \chi \left\| \dddot{M} \right\| + 6 d \gamma^{-1} \chi^2 \left\| \dot{M} \right\| \left\| \ddot{M} \right\| + 6 d^2 \gamma^{-2} \chi^3 \left\| \dot{M} \right\|^3$$

*where $\gamma = \min_{j \neq k} |\lambda_j - \lambda_k|$, $\chi = \max_k \|v_k\| \|w_k\|$ and $\|\cdot\|$ denotes operator norm.*

*Proof.* By differentiating the identity $M v_k = \lambda_k v_k$, we have that

$$\dot{M} v_k + M \dot{v}_k = \dot{\lambda}_k v_k + \lambda_k \dot{v}_k$$

$$\ddot{M} v_k + 2 \dot{M} \dot{v}_k + M \ddot{v}_k = \ddot{\lambda}_k v_k + 2 \dot{\lambda}_k \dot{v}_k + \lambda \ddot{v}_k$$

$$\dddot{M} v_k + 3 \ddot{M} \dot{v}_k + 3 \dot{M} \ddot{v}_k + M \dddot{v}_k = \dddot{\lambda}_k v_k + 3 \ddot{\lambda}_k \dot{v}_k + 3 \dot{\lambda}_k \ddot{v}_k + \lambda_k \dddot{v}_k.$$

Also note that $w_k^* M = \lambda_k w_k^*$. By multiplying the third identity by $w_k^*$ on the left, we have that

$$w_k^* \dddot{M} v_k + 3 w_k^* \ddot{M} \dot{v}_k + 3 w_k^* \dot{M} \ddot{v}_k + w_k^* M \dddot{v}_k = \dddot{\lambda}_k + 3 \ddot{\lambda}_k w_k^* \dot{v}_k + 3 \dot{\lambda}_k w_k^* \ddot{v}_k + \lambda_k w_k^* \dddot{v}_k$$

$$w_k^* \dddot{M} v_k + 3 w_k^* \ddot{M} \dot{v}_k + 3 w_k^* \dot{M} \ddot{v}_k = \dddot{\lambda}_k + 3 \ddot{\lambda}_k w_k^* \dot{v}_k + 3 \dot{\lambda}_k w_k^* \ddot{v}_k$$

since $(w_k^* M) \dddot{v}_k = \lambda_k w_k^* \dddot{v}_k$. Now let us compute $\ddot{v}_k$. By multiplying the identity for the second derivatives by $w_j^*$ on the left for any $j \neq k$, we have that

$$w_j^* \ddot{M} v_k + 2 w_j^* \dot{M} \dot{v}_k + \underbrace{w_j^* M}_{= \lambda_j w_j^*} \ddot{v}_k = \ddot{\lambda}_k w_j^* v_k + 2 \dot{\lambda}_k w_j^* \dot{v}_k + \lambda_k w_j^* \ddot{v}_k$$

$$w_j^* \ddot{v}_k = \frac{1}{\lambda_j - \lambda_k} \left( 2 \dot{\lambda}_k w_j^* \dot{v}_k - w_j^* \ddot{M} v_k - 2 w_j^* \dot{M} \dot{v}_k \right).$$

Hence we can compute

$$w_k^* \dot{M} \ddot{v}_k - \dot{\lambda}_k w_k^* \ddot{v}_k = w_k^* \dot{M} \sum_j (w_j^* \ddot{v}_k) v_j - \dot{\lambda}_k w_k^* \ddot{v}_k$$

$$= \sum_{j \neq k} \left( w_k^* \dot{M} v_j \right) \frac{1}{\lambda_j - \lambda_k} \left( 2\dot{\lambda}_k w_j^* \dot{v}_k - w_j^* \ddot{M} v_k - 2 w_j^* \dot{M} \dot{v}_k \right) + \underbrace{(w_k^* \dot{M} v_k)(w_k^* \ddot{v}_k) - \dot{\lambda}_k w_k^* \ddot{v}_k}_{=0}$$

since $\dot{\lambda}_k = w_k^* \dot{M} v_k$ by Lemma D.2. Substituting back into the identity for the third derivative left-multiplied by $w_k^*$, we find that

$$\dddot{\lambda}_k = w_k^* \dddot{M} v_k + 3 w_k^* \ddot{M} \dot{v}_k - 3 \ddot{\lambda}_k w_k^* \dot{v}_k + 3 \sum_{j \neq k} \frac{w_k^* \dot{M} v_j}{\lambda_j - \lambda_k} \left( 2 \dot{\lambda}_k w_j^* \dot{v}_k - w_j^* \ddot{M} v_k - 2 w_j^* \dot{M} \dot{v}_k \right).$$

The claimed expression for $\dddot{\lambda}_k$ will follow by substituting the expressions for $\dot{v}_k$, $\dot{\lambda}_k$ and $\ddot{\lambda}_k$ from Lemma D.2 and simplifying. Namely, letting for concision $\delta_{jk} = \lambda_j - \lambda_k$ and $M_{jk} = w_j^* M v_k$, $\dot{M}_{jk} = w_j^* \dot{M} v_k$ and similarly for $\ddot{M}_{jk}$, $\dddot{M}_{jk}$ for any $j, k \leq d$, we have

$$\dot{\lambda}_k = \dot{M}_{kk}, \qquad \dot{v}_k = \sum_{j \neq k} \frac{\dot{M}_{jk}}{\delta_{kj}} v_j + c_k v_k, \qquad \ddot{\lambda}_k = \ddot{M}_{kk} + 2 \sum_{j \neq k} \frac{\dot{M}_{kj} \dot{M}_{jk}}{\delta_{kj}}$$

and in particular $w_j^* \dot{v}_k = \frac{\dot{M}_{jk}}{\delta_{kj}}$ for all $j \neq k$ and so

$$\dddot{\lambda}_k = \dddot{M}_{kk} + 3 \sum_{j \neq k} \frac{\ddot{M}_{kj} \dot{M}_{jk}}{\delta_{kj}} + 3 \ddot{M}_{kk} c_k - 3 \left( \ddot{M}_{kk} + 2 \sum_{j \neq k} \frac{\dot{M}_{kj} \dot{M}_{jk}}{\delta_{kj}} \right) c_k$$

$$- 3 \sum_{j \neq k} \frac{\dot{M}_{kj} \ddot{M}_{jk}}{\delta_{jk}} + 6 \sum_{j \neq k} \frac{\dot{M}_{kj}}{\delta_{jk}} \left( \dot{M}_{kk} \frac{\dot{M}_{jk}}{\delta_{kj}} - \sum_{l \neq k} \frac{\dot{M}_{jl} \dot{M}_{lk}}{\delta_{kl}} - \dot{M}_{jk} c_k \right)$$

$$= \dddot{M}_{kk} + 3 \sum_{j \neq k} \frac{\ddot{M}_{kj} \dot{M}_{jk} + \dot{M}_{kj} \ddot{M}_{jk}}{\delta_{kj}} + 6 \sum_{j \neq k} \frac{\dot{M}_{kj}}{\delta_{jk}} \left( \dot{M}_{kk} \frac{\dot{M}_{jk}}{\delta_{kj}} - \sum_{l \neq k} \frac{\dot{M}_{jl} \dot{M}_{lk}}{\delta_{kl}} \right).$$

In order to simplify the last term, we simply write it as

$$\sum_{j \neq k} \frac{\dot{M}_{kj}}{\delta_{jk}} \left( \dot{M}_{kk} \frac{\dot{M}_{jk}}{\delta_{kj}} - \sum_{l \neq k} \frac{\dot{M}_{jl} \dot{M}_{lk}}{\delta_{kl}} \right) = \sum_{j,l \neq k} \frac{\dot{M}_{kj} \dot{M}_{jl} \dot{M}_{lk}}{\delta_{kj} \delta_{kl}} - \sum_{j \neq k} \frac{\dot{M}_{kj} \dot{M}_{jk}}{\delta_{kj}^2} \dot{M}_{kk}.$$

By substituting, we obtain the announced expression for $\dddot{\lambda}_k$. $\qquad \square$

In order to bound the term in $\dddot{\lambda}_k(\xi)$ in (D.2), we want to control

$$\gamma(\alpha) := \min_{j \neq k} |\lambda_j(\alpha) - \lambda_k(\alpha)| \quad \text{and} \quad \chi(\alpha) := \max_k \|v_k(\alpha)\| \, \|w_k(\alpha)\|$$

the uniform eigengap resp. maximal eigenvalue condition number, uniformly in $\alpha$ for $\alpha$ in a neighborhood of zero. Now assuming $M_0$ has distinct eigenvalues then $\gamma(0) > 0$, and assuming $M_0$ is normal – as is the case for Prop. D.1 – then $\chi(0) = 1$. So we want to bound the perturbation of the eigenvalues and eigenvectors uniformly.

For the eigenvalues, thanks to the assumption that $M_0$ is normal, we easily get the following bound.

**Lemma D.4.** *Under the conditions of Lemma D.2, if additionally $M_0$ is normal, then it holds*

$$\max_k |\lambda_k(\alpha) - \lambda_k(0)| \leq |\!|\!| M_\alpha - M_0 |\!|\!|$$

*for all $\alpha \in I'$ the maximal open interval containing 0 such that $\forall \alpha \in I', |\!|\!| M_\alpha - M_0 |\!|\!| < \frac{1}{2} \gamma(0)$.*

*In particular, we have for all $\alpha \in I'$*

$$\gamma(\alpha) \geq \gamma(0) - 2 |\!|\!| M_\alpha - M_0 |\!|\!| > 0.$$

Note that $I' \subset I$ the interval from Lemma D.2, since $\gamma(\alpha) > 0$ if and only if $M_\alpha$ has distinct eigenvalues.

*Proof.* Let $\lambda$ any eigenvalue of $M_\alpha$ and $v$ such that $M_\alpha v = \lambda v$ and $\|v\| = 1$. Since $M_0$ is normal, it is orthonormally diagonalizable, so we may write it as $M_0 = U_0 \Lambda_0 \overline{U}_0^\top$ with $U_0$ unitary and $\Lambda_0 = \mathrm{Diag}((\lambda_k(0)))$. Then, letting $\tilde{v} = \overline{U}_0^\top v$ and $\Delta = M_\alpha - M_0$ and $\widetilde{\Delta} = \overline{U}_0^\top \Delta U_0$,

$$M_\alpha v = (M_0 + \Delta)v = U_0^\top (\Lambda_0 + \widetilde{\Delta})\overline{U}_0^\top v = \lambda v, \text{ i.e., } (\Lambda_0 + \widetilde{\Delta})\tilde{v} = \lambda \tilde{v}$$

and so $\min_j |\lambda - \lambda_{0j}|^2 \leq \sum_j |\tilde{v}[j]|^2 |\lambda - \lambda_{0j}|^2 = \left\| (\Lambda_0 + \widetilde{\Delta})\tilde{v} \right\|^2 \leq \left\| \widetilde{\Delta}\tilde{v} \right\|^2 \leq \|\Delta\|^2$

since $U_0$ is unitary and $\|v\| = 1$.

This shows that for any $\lambda \in \mathrm{Sp}(M_\alpha)$, there exists a $\lambda_k(0) \in \mathrm{Sp}(M_0)$ which is close to it, i.e.,

$$\forall k, \exists j \text{ s.t } |\lambda_k(\alpha) - \lambda_j(0)| \leq \|M_\alpha - M_0\|.$$

A fortiori, we can ensure that the smooth parametrization of the eigenvalues $(\lambda_k(\alpha))_k$ satisfies the inequality announced in the lemma, by restraining $\alpha$ to some $I'$ small enough so that $\arg\min_j |\lambda_k(\alpha) - \lambda_j(0)| = k$ for all $\alpha \in I'$. More explicitly, this can be achieved by choosing $I'$ such that $\sup_{\alpha \in I'} \|M_\alpha - M_0\| < \frac{1}{2} \min_{j \neq k} |\lambda_j(0) - \lambda_k(0)| = \frac{1}{2}\gamma(0)$. Hence the choice of $I'$ announced in the lemma. $\qquad \square$

For the eigenvectors, also using the assumption that $M_0$ is normal, we get the following bound.

**Lemma D.5.** *Under the conditions of Lemma D.2, assume additionally that $M_0$ is normal, and choose the normalization of the eigenvectors $(v_k(\alpha))_k$ such that $\|v_k(\alpha)\| = 1$ and $v_k^*(0) \cdot v_k(\alpha) \in \mathbb{R}_+$ for all $k$. Then*

$$\forall k, \ \|v_k(\alpha) - v_k(0)\| \leq \frac{2\sqrt{2}\|M_\alpha - M_0\|}{\gamma(0)}$$

$$\text{and} \quad \|w_k^*(\alpha) - v_k^*(0)\| \leq \sqrt{d}\,\|w_k(\alpha)\| \frac{2\sqrt{2}\|M_\alpha - M_0\|}{\gamma(0)}$$

*for all $\alpha \in I'$ the interval from Lemma D.4.*

*In particular, we have*

$$\chi(\alpha) \leq \frac{\gamma(0)}{\gamma(0) - 2\sqrt{2d}\|M_\alpha - M_0\|}$$

*for all $\alpha \in I''$ the maximal open interval containing $0$ such that $\forall \alpha \in I'', \|M_\alpha - M_0\| < \frac{1}{2\sqrt{2d}}\gamma(0)$.*

*Proof.* We will write for concision $v_k = v_k(\alpha)$ and $v_{0k} = v_k(0) = w_k(0)$ since $M_0$ is normal, and $\lambda_k = \lambda_k(\alpha)$, $\lambda_{0k} = \lambda_k(0)$. Fix $\alpha \in I'$ and let $\Delta = M_\alpha - M_0$; by Lemma D.4 we have $\max_k |\lambda_k - \lambda_{0k}| \leq \|\Delta\| \leq \frac{1}{2}\gamma(0)$.

Fix $k$. Subtracting $M_0 v_{0k} = \lambda_{0k} v_{0k}$ from $(M_0 + \Delta)v_k = \lambda_k v_k$, we have

$$M_0(v_k - v_{0k}) + \Delta v_k = \lambda_k v_k - \lambda_{0k} v_{0k} = \lambda_{0k}(v_k - v_{0k}) + (\lambda_k - \lambda_{0k})v_{0k}$$
$$(M_0 - \lambda_{0k}I)(v_k - v_{0k}) = -\Delta v_k + (\lambda_k - \lambda_{0k})v_{0k}$$
$$\sum_{j \neq k} |\lambda_{0j} - \lambda_{0k}|^2 |v_{0j}^* v_k|^2 = \|(M_0 - \lambda_{0k}I)(v_k - v_{0k})\|^2 \leq \|\Delta\|^2 (\|v_k\| + 1)^2 = 4\|\Delta\|^2$$

where $v_{0j}^* := \overline{v}_{0j}^\top$, using the unitary basis $(v_{0j})_j$ to compute the norm on the left-hand side. The left-hand side is further lower-bounded by $\gamma(0)^2 \sum_{j \neq k} |v_{0j}^* v_k|^2 = \gamma(0)^2 \left(1 - |v_{0k}^* v_k|^2\right)$, so

$$1 - |v_{0k}^* v_k|^2 \leq \frac{4\|\Delta\|^2}{\gamma(0)^2}.$$

Consequently, by our choice of normalization: $v_k^*(0)v_k(\alpha) \in \mathbb{R}_+$, we have

$$\|v_k - v_{0k}\|^2 = 2 - 2\Re(v_{0k}^* v_k) = 2 - 2\,|v_{0k}^* v_k| \le 2\left(1 - \sqrt{1 - \frac{4\|\Delta\|^2}{\gamma(0)^2}}\right) \le \frac{8\|\Delta\|^2}{\gamma(0)^2}$$

using that $\forall y \in [0,1]$, $1 - \sqrt{1 - y^2} \le y^2$. This shows the first inequality of the lemma.

Let us now show the control on $w_k := w_k^*(\alpha)$ – the second inequality of the lemma. Using the unitary basis $(v_{0j})_j$ to compute the norm, we have $\|w_k^* - v_{0k}^*\|^2 = \sum_{j \ne k} |w_k^* v_{0j}|^2 + |w_k^* v_{0k} - 1|^2$ and since by definition $w_k^* v_j = \mathbb{1}_{j=k}$, we can bound each term as

$$\forall j \ne k, \ |w_k^* v_{0j}| \le |w_k^* v_j| + |w_k^*(v_{0j} - v_j)| \le 0 + \|w_k^*\| \frac{2\sqrt{2}\|\Delta\|}{\gamma(0)}$$

$$\text{and} \quad |w_k^* v_{0k} - 1| = |w_k^* v_k - 1 + w_k^*(v_{0k} - v_k)| \le \|w_k^*\|\,\|v_{0k} - v_k\| \le \|w_k^*\| \frac{2\sqrt{2}\|\Delta\|}{\gamma(0)}.$$

So in total, $\|w_k^* - v_{0k}^*\|^2 \le d \cdot \|w_k^*\|^2 \frac{8\|\Delta\|^2}{\gamma(0)^2}$, as announced.

The second part of the lemma follows by noting that $\|v_k\|\,\|w_k\| = 1 \cdot \|w_k\|$ is bounded by

$$\|w_k\| \le \|v_{0k}\| + \|w_k - v_{0k}\| \le 1 + \sqrt{d}\,\|w_k\| \frac{2\sqrt{2}\|\Delta\|}{\gamma(0)}$$

$$\implies \quad \|w_k\| \le \frac{1}{1 - \sqrt{d} \cdot \frac{2\sqrt{2}\|\Delta\|}{\gamma(0)}} = \frac{\gamma(0)}{\gamma(0) - 2\sqrt{2}\sqrt{d}\|\Delta\|}$$

for all $\alpha$ such that the denominator in the second line is positive, as announced. $\qquad\square$

By combining the three above lemmas, we have that under the conditions of Lemma D.2, if additionally $M_0$ is normal, then for all $\alpha$ in a neighborhood of zero such that $\|M_\alpha - M_0\| \le \frac{\gamma(0)}{4\sqrt{2d}}$, one can check that $\gamma(\alpha) \ge \frac{1}{2}\gamma(0)$ and $\chi(\alpha) \le 2$ and so

$$\frac{1}{6}\max_k \left|\dddot{\lambda}_k\right| \le \frac{1}{3}\left\|\dddot{M}\right\| + 8d\gamma(0)^{-1}\left\|\dot{M}\right\| \cdot \left\|\ddot{M}\right\| + 32d^2\gamma(0)^{-2}\left\|\dot{M}\right\|^3.$$

In particular, for a matrix $M_\alpha = M_0 + \alpha M_1 + \frac{\alpha^2}{2}M_2$ with $M_0$ normal, the right-hand side translates exactly to the bound on $|\boldsymbol{r}_j|$ announced in Prop. D.1. So that proposition follows directly from (D.2) and the above discussion.

## E Local convergence of equality-constrained Mirror Flow

This appendix contains results used in App. F.

Let $\mathcal{X}, \mathcal{Y}$ convex subsets of $\mathbb{R}^n$ resp. $\mathbb{R}^m$ and $\phi_x : \mathcal{X} \to \mathbb{R}$, $\phi_y : \mathcal{Y} \to \mathbb{R}$ strictly convex and differentiable. Let $\mathcal{Z} = \mathcal{X} \times \mathcal{Y}$ and $\phi : \mathcal{Z} \to \mathbb{R}$ with $\phi(x,y) = \phi_x(x) + \phi_y(y)$. Consider a twice continuously differentiable min-max objective $f : \mathcal{X} \times \mathcal{Y} \to \mathbb{R}$, denote $g(z) = \mathrm{Diag}(I_n, -I_m)\nabla f(z)$ and $M(z) = \nabla g(z)$. Throughout this appendix we make the following assumption.

**Assumption 1.** *The constraint set is defined by equalities: $\mathcal{Z} = \{z \in \mathbb{R}^{n+m}; Az = b\} = z_b + \mathrm{Ker}\,A$, where $z_b$ is any solution of $Az_b = b$. Furthermore, $\phi$ is strictly convex and three times differentiable.*

**Definition E.1** ([AW20, Proposition 1])**.** For an initial point $z^0 \in \mathcal{Z}$, the mirror flow (MF) with link function $\phi$ is the unique curve $z(t)$ such that $z(0) = z^0$ and

$$\frac{dz}{dt} = -\Phi_z^{-1} P_z g(z) =: -g_{\text{eff}}(z),$$

where $\Phi_z$ denotes the Hessian of $\phi$ at $z$ and $P_z := I - A^\top \left[A\Phi_z^{-1}A^\top\right]^{-1} A\Phi_z^{-1}$.

One can check that $P_z^2 = P_z$ and that $P_z \Phi_z = \Phi_z P_z^\top$. Furthermore, $P_z A^\top = 0$, i.e., $\operatorname{Im} P_z^\top \subset \operatorname{Ker} A$, and in fact $\operatorname{Im} P_z^\top = \operatorname{Ker} A$ since $(\operatorname{Im} P_z^\top)^\perp = \operatorname{Ker} P_z = \operatorname{Im}(I - P_z) \subset \operatorname{Im} A^\top = (\operatorname{Ker} A)^\perp$. In particular the MF preserves the constraint set, since $\frac{d}{dt}(Az - b) = -AP^\top \Phi_z^{-1} g(z) = 0$. As a consequence, $g_{\mathrm{eff}}$ can be seen as an operator from the affine space $\mathcal{Z} = z_b + \operatorname{Ker} A$ to itself.

**Lemma E.1.** *The Jacobian of $g_{\mathrm{eff}} : \mathbb{R}^{n+m} \to \mathbb{R}^{n+m}$ at a local NE $z^*$ is equal to $\Phi_{z^*}^{-1} P_{z^*} \cdot M(z^*)$.*

*Furthermore, the Jacobian of $g_{\mathrm{eff}} : \mathcal{Z} \to \mathcal{Z}$ (seen as an operator between affine spaces) at $z^*$ is*

$$M_{\mathrm{eff}}(z^*) = \Phi_{z^*}^{-1} P_{z^*} \cdot M(z^*) P_{z^*}^\top$$

*which is a linear operator from $\operatorname{Ker} A$ to itself.*

*Proof.* In this proof, write for concision $P = P_z$ and $\Phi^{-1} = \Phi_z^{-1}$. Using Einstein's summation notation, pose

$$P_i{}^j = P = I - A^\top \left[ A\Phi^{-1} A^\top \right]^{-1} A\Phi^{-1} \qquad \text{and} \qquad Q^{ij} = Q := \Phi^{-1} P.$$

In particular $g_{\mathrm{eff}}(z) = Qg$. Now

$$\nabla \{Qg\}^i{}_j = \frac{\partial [Qg]^i}{\partial z^j} = (Q\nabla g)^i{}_j + \frac{\partial Q^{ik}}{\partial z^j} g_k. \tag{E.1}$$

Using formula (59) from [PP+08]: $\frac{\partial Y^{-1}}{\partial x} = -Y^{-1} \frac{\partial Y}{\partial x} Y^{-1}$, we have that

$$\frac{\partial (\Phi^{-1})^{il}}{\partial z^j} = -(\Phi^{-1})^{ia} K_{abj} (\Phi^{-1})^{bl}$$

$$\frac{\partial P_l{}^k}{\partial z^j} = - \left( A^\top [A\Phi^{-1} A^\top]^{-1} A \right)_l{}^s K_{stj} \left( \Phi^{-1} A^\top [A\Phi^{-1} A^\top]^{-1} A\Phi^{-1} \right)^{tk}$$

$$\qquad - (A^\top)_{la} \frac{\partial \{[A\Phi^{-1} A^\top]^{-1}\}^{ab}}{\partial z^j} (A\Phi^{-1})_b{}^k$$

where $K = \nabla^3 \phi(z)$, and after calculating and simplifying,

$$\frac{\partial P_l{}^k}{\partial z^j} = (I - P)_l{}^s K_{stj} (\Phi^{-1} P)^{tk}.$$

So

$$\frac{\partial Q^{ik}}{\partial z^j} = (\Phi^{-1})^{il} \frac{\partial P_l{}^k}{\partial z^j} + \frac{\partial (\Phi^{-1})^{il}}{\partial z^j} P_l{}^k$$

$$= -(\Phi^{-1} P)^{is} K_{stj} (\Phi^{-1} P)^{tk} = -Q^{is} K_{stj} Q^{tk}$$

and finally $\quad \dfrac{\partial Q^{ik}}{\partial z^j} g_k = -Q^{is} K_{stj} (Qg)^t.$

Now $g_{\mathrm{eff}}(z^*) = 0$, so $Qg = 0$ at $z^*$. The first part of the lemma follows by substituting into (E.1).

To check the second part of the lemma, simply use that $\Phi_{z^*}^{-1} P_{z^*} = P_{z^*}^\top \Phi_{z^*}^{-1}$ and that $P_{z^*}^\top$ is a projector onto the kernel of $A$, as remarked under Definition E.1. $\qquad \square$

The next lemma is a special case of the stable manifold theorem [Per13, Section 2.7].

**Lemma E.2.** *Let $\mathcal{Z} = z_b + T\mathcal{Z}$ an affine subspace of $\mathbb{R}^d$. Let a continuously differentiable operator $g : \mathcal{Z} \to \mathcal{Z}$ and denote its Jacobian as $\nabla g : \mathcal{Z} \to T\mathcal{Z} \times T\mathcal{Z}$. Suppose there exists $z^*$ such that $g(z^*) = 0$ and $\min_{\lambda \in \mathrm{Sp}(\nabla g(z^*))} \Re(\lambda) > 0$.*

*Then for any $\varepsilon > 0$, there exists $C > 0$ and a relative neighborhood of $z^*$ such that, for any $z^0$ in that neighborhood, the flow $\frac{dz}{dt} = -g(z)$ converges exponentially to $z^*$ with*

$$\|z(t) - z^*\|^2 \leq C \|z^0 - z^*\|^2 \exp \left( -2 \left[ \min_{\lambda \in \mathrm{Sp}(\nabla g(z^*))} \Re(\lambda) - \varepsilon \right] t \right).$$

**Proposition E.3.** *If $\widetilde{M} := \Phi_{z^*}^{-\frac{1}{2}} P_{z^*} \cdot M(z^*) \cdot P_{z^*}^\top \Phi_{z^*}^{-\frac{1}{2}}$ is invertible and satisfies the equivalent conditions of Thm. 2.1, then MF converges locally exponentially to $z^*$ at a rate $\tilde{\mu}_{\widetilde{M}}$.*

*Proof.* $\widetilde{M}$ is similar to $M_{\text{eff}}(z^*)$. In particular they have the same eigenvalues. So the proposition follows immediately from the above two lemmas. □

*Remark* E.1. The matrix $\widetilde{M}$ can be interpreted as

$$\widetilde{M} = \underbrace{\Phi_{z^*}^{-\frac{1}{2}} P_{z^*} \Phi_{z^*}^{\frac{1}{2}}}_{=:R_{z^*}} \cdot \Phi_{z^*}^{-\frac{1}{2}} M(z^*) \Phi_{z^*}^{-\frac{1}{2}} \cdot \underbrace{\Phi_{z^*}^{\frac{1}{2}} P_{z^*}^{\top} \Phi_{z^*}^{-\frac{1}{2}}}_{=R_{z^*}^{\top}}.$$

Note that $R_{z^*}$ is an orthogonal projection. Furthermore, the central factor consists in a transformation of $M(z^*)$ that is compatible with the block structure:

- If we write $M(z^*) = S + A$ with $S$ symmetric and $A$ antisymmetric, then $\Phi_{z^*}^{-\frac{1}{2}} M(z^*) \Phi_{z^*}^{-\frac{1}{2}} = \Phi_{z^*}^{-\frac{1}{2}} S \Phi_{z^*}^{-\frac{1}{2}} + \Phi_{z^*}^{-\frac{1}{2}} A \Phi_{z^*}^{-\frac{1}{2}}$ and the first term is symmetric and the second term is antisymmetric.

- If we write $M(z^*) = \begin{bmatrix} Q & P \\ -P^{\top} & R \end{bmatrix}$ with $Q, R$ symmetric, then

$$\Phi_{z^*}^{-\frac{1}{2}} M(z^*) \Phi_{z^*}^{-\frac{1}{2}} = \begin{bmatrix} \Phi_{x^*}^{-\frac{1}{2}} & 0 \\ 0 & \Phi_{y^*}^{-\frac{1}{2}} \end{bmatrix} \begin{bmatrix} Q & P \\ -P^{\top} & R \end{bmatrix} \begin{bmatrix} \Phi_{x^*}^{-\frac{1}{2}} & 0 \\ 0 & \Phi_{y^*}^{-\frac{1}{2}} \end{bmatrix} = \begin{bmatrix} \Phi_{x^*}^{-\frac{1}{2}} Q \Phi_{x^*}^{-\frac{1}{2}} & \Phi_{x^*}^{-\frac{1}{2}} P \Phi_{y^*}^{-\frac{1}{2}} \\ -\left(\Phi_{x^*}^{-\frac{1}{2}} P \Phi_{y^*}^{-\frac{1}{2}}\right)^{\top} & \Phi_{y^*}^{-\frac{1}{2}} R \Phi_{y^*}^{-\frac{1}{2}} \end{bmatrix}$$

  is of the same form, where $\Phi_{x^*}$ is the Hessian of $\phi_x$ at $x^*$ and similarly for $\Phi_{y^*}$.

We state an analogous result for Mirror Descent-Ascent (MDA) and its variants, Mirror Prox (MP) and Bregman PP. Its proof, omitted for brevity, involves exactly the same ideas as for MF.

**Proposition E.4.** *Under the assumptions of Prop. E.3, if additionally $\phi$ is strongly convex, then MDA, MP and Bregman PP converge locally exponentially to $z^*$ for any small enough step-size at a rate $\eta \tilde{\mu}_{\widetilde{M}} + O(\eta^2)$.*

# F  Details for Sec. 4

In this appendix we provide details for Fig. 2, derive the first-order term (in $\eta$) in the local convergence rate of MP and EG, and discuss a counter-example. For our purpose it is sufficient to consider only the continuous-time flows of those algorithms.

## F.1  Details for Fig. 2

In the numerical experiment reported in Fig. 2, we used the same random payoff functions as in [WC22, Section 4]:

$$f(x, y) = \Re \sum_{-K \leq k \leq K} \sum_{-L \leq l \leq L} c_{kl} e^{2\pi i (kx + ly)} \quad \text{for } (c_{kl}) \in \mathbb{C}^{(2K+1) \times (2L+1)}, \qquad \text{(F.1)}$$

with $K = L = 2$ and $\Re(c_{kl}), \Im(c_{kl})$ drawn independently from the standard Gaussian distribution. We measured the distance between the solution $z^* = (a^*, x^*, b^*, y^*)$ and the iterates $z^k$ by

$$\|z - z^*\|^2 := \|a - a^*\|^2 + \|b - b^*\|^2 + \|x - x^*\|^2 + \|y - y^*\|^2.$$

For $z$ in a neighborhood of $z^*$, $\|z - z^*\|$ is an upper bound on the duality gap of the original MNE problem up to a constant dependent only on $f$ [WC22, Proposition 3.1 and Claim C.3].

## F.2  First-order term in the local convergence rate of MP

Denote the joint variable as $z = (a, b) \in \Delta_N \times \Delta_M$ and let $z^*$ such that $(\mu^*, \nu^*) = \left(\sum_I a_I^* \delta_{x_I^*}, \sum_J b_J^* \delta_{y_J^*}\right)$ is the MNE. The continuous-time flow of MP is MF (Definition E.1) with $\phi(z) = \sum_I z_I \log z_I$, $A = \begin{bmatrix} \mathbf{1}_N^{\top} & 0 \\ 0 & \mathbf{1}_M^{\top} \end{bmatrix} \in \mathbb{R}^{2 \times (N+M)}$ and $b = \begin{pmatrix} 1 \\ 1 \end{pmatrix}$. So by Prop. E.4, the first-order term in the local convergence rate of MP is $\eta \tilde{\mu}_{M_{\text{MP}}}$ for some $M_{\text{MP}}$ which we now compute.

Following the notations of App. E, we have $\Phi_{z^*} = \text{Diag}\left(\left(\frac{1}{a_I^*}\right)_I, \left(\frac{1}{b_J^*}\right)_J\right)$, and by straightforward calculations $P_{z^*} = \text{Diag}\left(I_N - \mathbf{1}_N (a^*)^{\top}, I_M - \mathbf{1}_M (b^*)^{\top}\right)$, so that $R_{z^*} = \Phi_{z^*}^{-1/2} P_{z^*} \Phi_{z^*}^{1/2} =$

$\text{Diag}\left(I_N - \sqrt{a^*}\sqrt{a^*}^\top, I_M - \sqrt{b^*}\sqrt{b^*}^\top\right)$. In order to project out the constraints explicitly, let $\Pi_a$ any matrix in $\mathbb{R}^{(N-1)\times N}$ with orthonormal rows, i.e., $\Pi_a \Pi_a^\top = I_{N-1}$, and such that $\Pi_a^\top \Pi_a = I_N - \sqrt{a^*}\sqrt{a^*}^\top$, and likewise for $\Pi_b \in \mathbb{R}^{(M-1)\times M}$. For example the rows of $\Pi_a$ may be obtained by completing the unit vector $\sqrt{a^*}$ into an orthonormal basis and removing $\sqrt{a^*}$ from the basis. Then $R_{z^*}$ as a linear projector from $\mathbb{R}^{N+M}$ to $\{\sqrt{a^*}\}^\perp \times \{\sqrt{b^*}\}^\perp \simeq \mathbb{R}^{N+M-2}$ can be written as the matrix $R_{z^*} = \text{Diag}(\Pi_a, \Pi_b)$.[9] So $\widetilde{M}$ as a linear operator over $\{\sqrt{a^*}\}^\perp \times \{\sqrt{b^*}\}^\perp$ can be written as the matrix, denoting $D_a = \text{Diag}(\sqrt{a^*})$ and $D_b = \text{Diag}(\sqrt{b^*})$ for concision,

$$
\begin{aligned}
M_{\text{MP}} &:= R_{z^*} \Phi_{z^*}^{-1/2} M(z^*) \Phi_{z^*}^{-1/2} R_{z^*}^\top \\
&= \text{Diag}(\Pi_a, \Pi_b) \text{Diag}\left(\sqrt{a^*}, \sqrt{b^*}\right) \begin{bmatrix} \mathbf{0} & P \\ -P^\top & \mathbf{0} \end{bmatrix} \text{Diag}\left(\sqrt{a^*}, \sqrt{b^*}\right) \text{Diag}\left(\Pi_a^\top, \Pi_b^\top\right) \\
&= \begin{bmatrix} \mathbf{0} & \Pi_a D_a P D_b \Pi_b^\top \\ -\Pi_b D_b P^\top D_a \Pi_a^\top & \mathbf{0} \end{bmatrix}.
\end{aligned}
$$

### F.3 First-order term in the local convergence rate of EG (conic particle methods)

As in the main text, we write "EG" to refer to the Conic Particle Mirror Prox algorithm of [WC22].

Denote the joint variable as $z = (a, x, b, y) \in \Delta_N \times (\mathbb{T}^1)^N \times \Delta_M \times (\mathbb{T}^1)^M$ and let any $z^*$ such that $(\mu^*, \nu^*) = \left(\sum_I a_I^* \delta_{x_I^*}, \sum_J b_J^* \delta_{y_J^*}\right)$ is the MNE. The flow is given by the system of ODEs

$$
\begin{aligned}
\frac{da}{dt} &= -\Phi_a^{-1} P_a \nabla_a F(z) & \text{and} && \frac{db}{dt} &= \Phi_b^{-1} P_b \nabla_b F(z) & \text{(F.2)} \\
\frac{dx}{dt} &= -\gamma \text{Diag}\left(\frac{1}{a_I}\right) \nabla_x F(z) & && \frac{dy}{dt} &= \gamma \text{Diag}\left(\frac{1}{b_J}\right) \nabla_y F(z).
\end{aligned}
$$

Here $F(z) = \sum_I \sum_J a_I b_J f(x_I, y_J)$, $\gamma$ is a constant parameter, $\Phi_a = \text{Diag}\left(\frac{1}{a_I}\right)$ and $P_a = I_N - \mathbf{1}_N a^\top$, and $\Phi_b$ and $P_b$ are defined similarly.

In general, the flow (F.2) does not match the structure of Mirror Flow because of the factor "$\text{Diag}(\frac{1}{a_I})$" in the equation for $\frac{dx}{dt}$ [GWS21, Section 2.4]. However, by adapting the reasoning of App. E – namely only Lemma E.1 needs to be adapted – it is easy to show that the statement of Prop. E.3 holds also for this dynamics. By the same adaptation one can show that the statement of Prop. E.4 holds also for EG. Hence the first-order term in the local convergence rate of EG is $\eta \tilde{\mu}_{M_\gamma}$ for some $M_\gamma$ which we now compute.

The statement of Prop. E.3 applies to $\widetilde{M} = R_{z^*} \cdot \Phi_{z^*}^{-1/2} M(z^*) \Phi_{z^*}^{-1/2} \cdot R_{z^*}^\top$ where $\Phi_{z^*} = \text{Diag}\left(\left(\frac{1}{a_I^*}\right)_I, \frac{1}{\gamma} a^*, \left(\frac{1}{b_J^*}\right)_J, \frac{1}{\gamma} b^*\right)$ and $P_{z^*} = \text{Diag}(P_{a^*}, I_N, P_{b^*}, I_M)$ so that $R_{z^*} = \Phi_{z^*}^{-1/2} P_{z^*} \Phi_{z^*}^{1/2} = \text{Diag}\left(I_N - \sqrt{a^*}\sqrt{a^*}^\top, I_N, I_M - \sqrt{b^*}\sqrt{b^*}^\top, I_M\right)$. In order to project out the constraints explicitly, let $\Pi_a \in \mathbb{R}^{(N-1)\times N}$, $\Pi_b \in \mathbb{R}^{(M-1)\times M}$ the same matrices as in the previous subsection. Then $R_{z^*}$ as a linear projector from $\mathbb{R}^{2N+2M}$ to $\{\sqrt{a^*}\}^\perp \times \mathbb{R}^N \times \{\sqrt{b^*}\}^\perp \times \mathbb{R}^M \simeq \mathbb{R}^{2N+2M-2}$ can be written as the matrix $R_{z^*} = \text{Diag}\left(\Pi_a, I_N, \Pi_b, I_M\right)$. Moreover by [WC22, Claim C.2], dropping superscript *'s for concision only in this equation,

$$
M(z^*) = \begin{bmatrix}
\mathbf{0} & \mathbf{0} & P & \partial_y P \text{Diag}(b) \\
\mathbf{0} & \text{Diag}(a)\text{Diag}(\partial_{xx}^2 P b) & \text{Diag}(a)\partial_x P & \text{Diag}(a)\partial_{xy}^2 P \text{Diag}(b) \\
-P^\top & -(\text{Diag}(a)\partial_x P)^\top & \mathbf{0} & \mathbf{0} \\
-(\partial_y P \text{Diag}(b))^\top & -(\text{Diag}(a)\partial_{xy}^2 P \text{Diag}(b))^\top & \mathbf{0} & -\text{Diag}(b)\text{Diag}(\partial_{yy}^2 P^\top a)
\end{bmatrix}
$$

where $[\partial_x P]_{IJ} = \partial_x f(x_I^*, y_J^*)$, and likewise for $\partial_y P$, $\partial_{xx}^2 P$, $\partial_{yy}^2 P$, $\partial_{xy}^2 P$. So, finally, $\widetilde{M} = R_{z^*} \Phi_{z^*}^{-1/2} M(z^*) \Phi_{z^*}^{-1/2} R_{z^*}^\top$ as a linear operator over $\{\sqrt{a^*}\}^\perp \times \mathbb{R}^N \times \{\sqrt{b^*}\}^\perp \times \mathbb{R}^M$ can be written

---

[9]The expression "$\text{Diag}(\Pi_a, \Pi_b)$", as well as "$\text{Diag}(\Pi_a, I_N, \Pi_b, I_M)$" in Sec. F.3, constitutes a slight abuse of our notation "Diag", since $\Pi_a$ and $\Pi_b$ are not square matrices. To remove any ambiguity: by $\text{Diag}(\Pi_a, \Pi_b)$ we mean the matrix $\begin{bmatrix} \Pi_a & \mathbf{0}_{(N-1)\times M} \\ \mathbf{0}_{(M-1)\times N} & \Pi_b \end{bmatrix}$, and similarly for $\text{Diag}(\Pi_a, I_N, \Pi_b, I_M)$.

as the matrix, denoting $D_a = \mathrm{Diag}(\sqrt{a^*})$ and $D_b = \mathrm{Diag}(\sqrt{b^*})$ for concision,

$$
M_\gamma := \begin{bmatrix}
\mathbf{0} & \mathbf{0} & \Pi_a D_a P D_b \Pi_b^\top & \sqrt{\gamma}\,\Pi_a D_a [\partial_y P] D_b \\
\mathbf{0} & \gamma \mathrm{Diag}(\partial_{xx}^2 P b^*) & \sqrt{\gamma} D_a [\partial_x P] D_b \Pi_b^\top & \gamma D_a [\partial_{xy}^2 P] D_b \\
-(*)^\top & -(*)^\top & \mathbf{0} & \mathbf{0} \\
-(*)^\top & -(*)^\top & \mathbf{0} & -\gamma \mathrm{Diag}(\partial_{yy}^2 P^\top a^*)
\end{bmatrix}.
$$

## F.4 An example where the first-order term in the local convergence rate of EG is zero

Consider the payoff function defined by (F.1) with $c_{20} = c_{02} = -i$, $c_{11} = 2$ and $c_{kl} = 0$ otherwise, i.e., $f(x, y) = \sin(4\pi x) + \sin(4\pi y) + 2\cos(2\pi x + 2\pi y)$. As shown in [WC22, Example 4.1], the MNE is unique and given by $a^* = b^* = \left(\frac{1}{2}, \frac{1}{2}\right)$, $x^* = \left(\frac{3}{8}, \frac{7}{8}\right)$, and $y^* = \left(\frac{1}{8}, \frac{5}{8}\right)$. So we can compute $M_\gamma$ explicitly in this case: we find

$$
P = \begin{pmatrix} -2 & 2 \\ 2 & -2 \end{pmatrix}, \ \partial_x P = \partial_y P = 0, \ \partial_{xx}^2 P \begin{pmatrix} 1/2 \\ 1/2 \end{pmatrix} = \partial_{yy}^2 P \begin{pmatrix} 1/2 \\ 1/2 \end{pmatrix} = \begin{pmatrix} 16\pi^2 \\ 16\pi^2 \end{pmatrix}, \ \partial_{xy} P = \begin{pmatrix} 8\pi^2 & -8\pi^2 \\ -8\pi^2 & 8\pi^2 \end{pmatrix}
$$

and $D_a = D_b = \frac{1}{\sqrt{2}} I$, and so (for a certain choice of $\Pi_a$ and $\Pi_b$, each of which is anyway determined up to a sign)

$$
M_\gamma = \left[\begin{array}{ccc|ccc}
0 & 0 & 0 & -2 & 0 & 0 \\
0 & \gamma(4\pi)^2 & 0 & 0 & \gamma(2\pi)^2 & -\gamma(2\pi)^2 \\
0 & 0 & \gamma(4\pi)^2 & 0 & -\gamma(2\pi)^2 & \gamma(2\pi)^2 \\
\hline
2 & 0 & 0 & 0 & 0 & 0 \\
0 & -\gamma(2\pi)^2 & \gamma(2\pi)^2 & 0 & \gamma(4\pi)^2 & 0 \\
0 & \gamma(2\pi)^2 & -\gamma(2\pi)^2 & 0 & 0 & \gamma(4\pi)^2
\end{array}\right].
$$

This matrix clearly does not satisfy condition $(iii)$ of Thm. 2.1, so $\tilde\mu_{M_\gamma} = 0$.

## F.5 Proof of Prop. 4.1

For ease of reference, we restate the proposition below.

**Proposition 4.1.** *Let $S_2$ symmetric and $A_0, A_1, A_2$ antisymmetric real matrices of the form*

$$
S_2 = \left[\begin{array}{c|c} \mathbf{0} & \\ \hline & * \\ \hline & \mathbf{0} \\ & * \end{array}\right], \ A_0 = \left[\begin{array}{c|c} & * \ \ \mathbf{0} \\ & \mathbf{0} \ \ \mathbf{0} \\ \hline * \ \ \mathbf{0} & \\ \mathbf{0} \ \ \mathbf{0} & \end{array}\right], \ A_1 = \left[\begin{array}{c|c} & \mathbf{0} \ \ * \\ & * \ \ \mathbf{0} \\ \hline \mathbf{0} \ \ * & \\ * \ \ \mathbf{0} & \end{array}\right], \ A_2 = \left[\begin{array}{c|c} & \mathbf{0} \ \ \mathbf{0} \\ & \mathbf{0} \ \ * \\ \hline \mathbf{0} \ \ \mathbf{0} & \\ \mathbf{0} \ \ * & \end{array}\right]
$$

*and $M_\gamma = \gamma S_2 + A_0 + \sqrt{\gamma} A_1 + \gamma A_2$ for all $\gamma > 0$. Then $\tilde\mu_{M_\gamma} = O(\gamma^2)$ as $\gamma \to 0$.*

Pose $\alpha = \sqrt{\gamma}$, $M_0 = A_0 + A_\varepsilon$, $M_1 = A_1$ and $M_2 = 2(A_2 + S_2)$, where $A_\varepsilon$ is any antisymmetric matrix such that $|\!|\!| A_\varepsilon |\!|\!| \leq \varepsilon$ and $M_0$ has distinct eigenvalues. We will prove the proposition by applying the spectral expansions of Sec. D.5 to the matrix curve $M_\alpha = M_0 + \alpha M_1 + \frac{\alpha^2}{2} M_2$.

Adopting the notations of that section, we have the expansion for the eigenvalues $\lambda_k(\alpha)$ of $M_\alpha$:

$$
\lambda_k(\alpha) = \lambda_k(0) + \alpha \dot\lambda_k(0) + \frac{\alpha^2}{2} \ddot\lambda_k(0) + \frac{\alpha^3}{3!} \dddot\lambda_k(0) + \frac{\alpha^4}{4!} \ddddot\lambda_k(0) + O(\alpha^5),
$$

with $\{\lambda_k(0)\}_k = \mathrm{Sp}(M_0) \subset i\mathbb{R}$ by antisymmetry, $\dot\lambda_k(0) = v_{0k}^* A_1 v_{0k} \in i\mathbb{R}$ – where $v_{0k}$ are the eigenvectors of $M_0$ – by normality of $M_0$ and antisymmetry of $A_1$, and

$$
\ddot\lambda_k(0) = 2 v_{0k}^*(A_2 + S_2) v_{0k} + 2 \sum_{j \neq k} \underbrace{\frac{(v_{0k}^* A_1 v_{0j})(v_{0j}^* A_1 v_{0k})}{\lambda_k(0) - \lambda_j(0)}}_{\in i\mathbb{R}}
$$

since $v_{0k}^* A_1 v_{0j} = -(v_{0j}^* A_1 v_{0k})^*$, and

$$
\dddot\lambda_k(0) = 3 \sum_{j \neq k} \frac{1}{\lambda_k(0) - \lambda_j(0)} \left[ (v_{0k}^* A_1 v_{0j})(v_{0j}^* M_2 v_{0k}) + (v_{0k}^* M_2 v_{0j})(v_{0j}^* A_1 v_{0k}) \right]
$$

$$
+ 6 \underbrace{\sum_{j, l \neq k} \frac{(v_{0k}^* A_1 v_{0j})(v_{0j}^* A_1 v_{0l})(v_{0l}^* A_1 v_{0k})}{(\lambda_k(0) - \lambda_j(0))(\lambda_k(0) - \lambda_l(0))}}_{\in i\mathbb{R}} - 6 \sum_{j \neq k} \underbrace{\frac{(v_{0k}^* A_1 v_{0j})(v_{0j}^* A_1 v_{0k})}{(\lambda_k(0) - \lambda_j(0))^2} v_{0k}^* A_1 v_{0k}}_{\in i\mathbb{R}}
$$

as one can check by computing the convex conjugate of each of the underbraced expressions.

Now, let $\mathcal{K}$ the set of indices corresponding to the non-zero eigenvalues of $A_0$, and $\mathcal{K}^{\complement} = \{1, ..., d\} \setminus \mathcal{K}$. Note that the eigenvectors of $A_0$ are of the form

$$\forall k \in \mathcal{K}, \ \tilde{v}_k = \begin{pmatrix} * \\ 0 \\ * \\ 0 \end{pmatrix} \quad \text{and} \quad \forall h \in \mathcal{K}^{\complement}, \ \tilde{v}_h = \begin{pmatrix} 0 \\ * \\ 0 \\ * \end{pmatrix}.$$

Furthermore, note that

$$A_1 \begin{pmatrix} * \\ 0 \\ * \\ 0 \end{pmatrix} = \begin{pmatrix} 0 \\ * \\ 0 \\ * \end{pmatrix}, \qquad S_2 \begin{pmatrix} * \\ 0 \\ * \\ 0 \end{pmatrix} = \mathbf{0}, \qquad A_2 \begin{pmatrix} * \\ 0 \\ * \\ 0 \end{pmatrix} = \mathbf{0},$$

$$A_1 \begin{pmatrix} 0 \\ * \\ 0 \\ * \end{pmatrix} = \begin{pmatrix} * \\ 0 \\ * \\ 0 \end{pmatrix}, \qquad A_2 \begin{pmatrix} 0 \\ * \\ 0 \\ * \end{pmatrix} = \begin{pmatrix} 0 \\ * \\ 0 \\ * \end{pmatrix}, \qquad A_2 \begin{pmatrix} 0 \\ * \\ 0 \\ * \end{pmatrix} = \begin{pmatrix} 0 \\ * \\ 0 \\ * \end{pmatrix}.$$

Using this structure, one can check that for all $k \in \mathcal{K}$, $\Re \ddot{\lambda}_k(0) = o_\varepsilon(1)$ and $\Re \dddot{\lambda}_k(0) = o_\varepsilon(1)$.

Thus, the spectrum of $M_\alpha$ consists of eigenvalues (corresponding to indices $h \in \mathcal{K}^{\complement}$) with $\Re \ddot{\lambda}_h(0)$ non-zero a priori, in which case $\Re \lambda_h(\alpha) = \Theta(\alpha^2) = \Theta(\gamma)$, and of eigenvalues (corresponding to indices $k \in \mathcal{K}$) with $\Re \lambda_k(\alpha) = O(\alpha^4) + o_\varepsilon(1) = O(\gamma^2) + o_\varepsilon(1)$. By letting $\varepsilon \to 0$ and using that eigenvalues are continuous, this shows that $\tilde{\mu}_{M_\gamma} = O(\gamma^2)$ as $\gamma \to 0$.