# OpenReview forum: "Local Convergence of Gradient Methods for Min-Max Games: Partial Curvature Generically Suffices"
_NeurIPS.cc/2023/Conference — NeurIPS 2023 poster_

### Official Review · Reviewer_L3Mt · 2023-06-29

**Soundness:** 4 excellent
**Presentation:** 3 good
**Contribution:** 3 good
**Rating:** 6
**Confidence:** 4

**Summary:**

This paper studies the convergence of Gradient based methods to local Nash equilibria. The properties of the “potential” and the “interaction” parts of the game are analyzed, and conditions when this leads to convergence of gradient methods are studied.

**Strengths:**

This paper provides insights into the convergence of minimax algorithms, and highlights important differences from the minimization setting.

**Weaknesses:**

The writing and presentation of the results can be improved.

 - Is the y-axis in Figure 1(a) the rate of convergence. What is the significance of including the observed and the predicted $\tilde{\mu}_M$?
Can a plot of the the actual iterated of the algorithm for any one particular realization of $P$ also be included?
[Like how GDA spirals out for a simple bilinear game, how does the trajectory change for this new game? Does the spiraling towards the solution happen in a skewed manner?
Maybe a plot which shows the trajectories (for one particular P) of GDA diverging for the bilinear game, but converging when the new term is added could help the reader visualize how these algorithms converge
]

 - The proofs on page 4 has a few terms which are overflowing to the left.

 - The result showing the dependence on the average of the eigenvalues is surprising. How crucial is this result on the choice of the matrices $U$ and $V$ being distributed uniformly on the set of $n \times n$ orthonormal matrices? Proposition 3.2 and 3.3 cover the cases when the matrices $S$ is not sparse and sparse, but can a complete characterization on the way $P$ is drawn also be done?

**Questions:**

See above

**Limitations:**

See above

---

> ### Author Rebuttal · Authors · 2023-08-09
>
> We address each listed Weakness separately.
> - In Figure 1(a), the datapoints shown by circles represent the observed convergence rate $r$ of GDA with step-size $\eta$, obtained by really running GDA for many iterations for a small $\eta$.
> On the other hand the lines represent the quantity $\tilde{\mu}\_M = \min\_{\lambda \in \mathrm{Sp}(M)} \Re(\lambda)$ where $M$ is the Jacobian matrix, obtained simply by computing the eigenvalues of $M$ for different values of the regularization parameter $\alpha$.
> The fact that $r/\eta$ and $\tilde{\mu}\_M$ coincide confirms the known dynamical systems result that $\tilde{\mu}\_M = \lim\_{\eta \to 0} r/\eta$ (the convergence rate of GF).
> 	- Yes, we will add plots of the actual iterates for a particular $P$, in dimension $n=m=2$. The spiraling towards the solution indeed happens in a skewed manner, with the "axes" matching the eigenvectors of the Jacobian $M$.
> - The overflow on page 4 will be fixed.
> - Our motivation for studying a randomized setting was to capture the behavior of the convergence rate in the generic situation where the singular vectors of $P$, $Q$ and $R$ are in general position (i.e. do not have a particular form of alignment).
> Hence we chose the simplest setting: $(U, V)$ uniformly distributed, and we have not studied any other distribution.
> 	- The precise way $P$ is drawn does not matter, only the way its singular matrices $U$ and $V$ are distributed.
> Indeed $\tilde{\mu}\_{M\_\alpha}$ only depends on $S$ and $U, V$ up to $O(\alpha^3)$ (as a consequence of the eigenvalue expansion of Proposition 3.1).

---

> > ### Comment · Reviewer_L3Mt · 2023-08-15
> > **Response to Rebuttal**
> >
> > Thank you for the response.

---

### Official Review · Reviewer_JUcy · 2023-07-06

**Soundness:** 3 good
**Presentation:** 2 fair
**Contribution:** 3 good
**Rating:** 6
**Confidence:** 2

**Summary:**

This paper focuses on min-max games with partial curvature, i.e., the symmetric part S of the Jacobian is p.s.d. and nonzero, and specifies the necessary and sufficient conditions for the convergence of gradient flow. The authors show that when the interaction term dominates, the convergence rate could depend on the average of the eigenvalues of S.
They also use the problem of computing mixed Nash equilibria of continuous games to illustrate the results.

**Strengths:**

1. The paper is well-organized.
2. The results are novel and interesting.
3. The analysis is detailed and solid.

**Weaknesses:**

1. Propositions 4.2 and 4.3 are based on a very restrictive setting (see Questions).
2. Some expressions are not easy to follow.

**Questions:**

Propositions 3.2 and 3.3 require that U and V are uniformly distributed on $O_n$. After taking the expectation over all these problem instances, the averaged convergence rate depends on the average of the eigenvalues of S.
However, in practice, we are faced with a specific instance and such an expectation step might be invalid. When the antisymmetric part of the Jacobian is given in advance, is the convergence rate still related to the average of the eigenvalues of S?

**Limitations:**

This paper only studies local convergence.

---

> ### Author Rebuttal · Authors · 2023-08-09
>
> The relation between convergence rate and average of the eigenvalues of $S$ is only valid under a randomized setting.
> For a fixed $S$, in the worst case, the convergence rate does depend on the minimum eigenvalues of $S$ (to be precise, on $\sigma\_{\min}(Q) + \sigma\_{\min}(R)$, by the eigenvalue expansion of Proposition 3.1).
> However building the worst-case instance requires a very special alignment between the singular vectors of $P$, $Q$ and $R$.
> The role of the randomness in our analysis is to reveal the generic behavior of the convergence rate, when the singular vectors of $P$, $Q$ and $R$ are in general position.

---

> > ### Comment · Reviewer_JUcy · 2023-08-15
> >
> >  I'd like to thank the authors for their rebuttal. I keep my current score.

---

### Official Review · Reviewer_oPMS · 2023-07-06

**Soundness:** 4 excellent
**Presentation:** 4 excellent
**Contribution:** 3 good
**Rating:** 7
**Confidence:** 4

**Summary:**

The authors study the convergence of gradient descent-type algorithms for saddle-point problems of the form $\displaystyle\min_{x \in \mathbb{R}^n } \displaystyle\min_{x \in \mathbb{R}^m} f(x,y)$. Let $M$ denote the Hessian of $f$ at a local saddle point: convergence is governed by the minimum real part of all eigenvalues of $M,$ denoted $\mu_M$. We have $\mu_M \ge 0,$ and this inequality is strict for almost all $M\in \mathbb{R}^{n\times m}$, as the authors verify in their Theorem 2.1. This observation motivates the use of regularization/overparametrization for such problem involving quadratic/higher order terms.

Decomposing $M$ into symmetric and antisymmetric parts, $M=S + \alpha A$, with $\alpha$ small, and write
\begin{equation}
A = \left(\begin{array}{cc}
0 & P \\\\
-P^T & 0
 \end{array}\right),
\quad
S = \begin{bmatrix} Q & 0 \\\\
0 & R \end{bmatrix}.
\end{equation}

For $m=n,$ the Taylor expansion formula of Proposition 3.1 shows that $\mu_M$ is well-approximated by the expression $\displaystyle\min_{1\le j \le n} \left( u_j^T Q u_j  + v_j^T R v_j \right)$, where the $u_i, v_i$ are the left and right singular vectors of $P = U \Sigma V^T.$ Assuming $U,V$ are drawn uniformly from $O(n),$ the authors show that this expression (in expectation and with concentration) is asymptotically the average of the spectrum of $S,$ under the assumption that the spectrum is "spread out" in the sense $\operatorname{tr} (S) / \lVert S \rVert_F$ is bounded by $\sqrt{\log n} $ times some constant greater than 2. As a counterpoint, the authors  consider the "sparse" case where $S$ has a fixed number $r$ of nonzero eigenvalues, and derive corresponding asymptotics and concentration bounds. Some experiments illustrating the theory for classical gradient methods are given.

**Strengths:**

The paper is well-written overall. I am happy with the mathematical details, having read the main paper and the first two sections of the supplementary material and found no errors.

**Weaknesses:**

The main limitation I see is one that the authors have already suggested in the conclusion: namely, it is not clear how the insights developed in this paper could extend to $N$-player games for $N>2$, since the assumption that $S$ is positive definite is needed, and the "interactions" among more than two players seem to be more subtle to define.

I also found the experimental section to be rather confusing. It really feels like an afterthought, and far more technical than necessary. What is the connection with the previous sections? If the point is just to illustrate that extra parameters can accelerate convergence, then be clear about that and move the many unnecessary technical details to the appendix. It is rather surprising that your "randomly-drawn" $f$ has the property that Theorem 2.1 fails, no? Is it by design that the family of such functions will not have this property? I also think more attention could be drawn to the fact that $M_{\text{MP}}$ and $M_{\gamma}$, although not the actual Hessian of the payoff function $f,$ play the same role in the dynamics as $M$ would for vanilla gradient flow. Please make this more clear.

**Questions:**

line 156: "In this section we assume $m=n$ for simplicity." I assume this is WLOG, since $Q$ and $R$ are arbitrary in your results, and we can just add variables to maintain $A\gg  S$?

line 208: "four" -> "three"?

line 225: "x gets updated" more precisely, updated with the gradient rule rather than by averaging.

---

> ### Author Rebuttal · Authors · 2023-08-09
>
> Regarding the second paragraph of "Weaknesses":
> Section 4 is not an experimental section, but an application of the previous considerations to a particular class of min-max problems which is of its own interest in game theory:
> $
>     \min\_{\mu \in \mathcal{P}(\mathcal{X})} \max\_{\nu \in \mathcal{P}(\mathcal{Y})}
>     \left\lbrace F(\mu, \nu) \coloneqq \mathbb{E}\_{x \sim \mu, y \sim \nu} [f(x,y)] \right\rbrace.
> $
> Here "$f$" represents a parameter of the problem -- and not a min-max objective function itself (contrary to the notation in Sections 1-3). We will make sure to better draw attention to this change of convention.
> Because this is an infinite-dimensional problem when $\mathcal{X}$, $\mathcal{Y}$ are continuous sets, algorithms to solve it are based on reparametrized formulations, which are the ones we actually analyze.
>
> The fact that the property from Theorem 2.1 fails for the "natural" reparametrization
> $
>     \min\_{a \in \Delta\_N} \max\_{b \in \Delta\_M}
>     \left\lbrace
>         F\_1(a,b) \coloneqq
>         \sum\_I \sum\_J a\_I b\_J f(x^*\_I, y^*\_J)
>     \right\rbrace
> $
> for randomly drawn $f$, is not very surprising in hindsight. Indeed it is known that Mirror Flow does not converge unless $N=M=1$ [MPP18], so it was to be expected that $\tilde{\mu}\_{M\_{\mathrm{MP}}}=0$.
> By contrast, for the overparametrized formulation
> $
>     \min\_{a, x} \max\_{b, y}
>     \left\lbrace
>         F\_2(a, x, b, y) \coloneqq
>         \sum\_I \sum\_J a\_I b\_J f(x\_I, y\_J)
>     \right\rbrace,
> $
> the condition from Theorem 2.1 does indeed hold for randomly drawn $f$, as discussed in the paragraph "Overparameterization induces partial curvature" of Section 4.
> We realize that some of this background information is lacking in Section 4, and we will add it.
>
> Regarding the Questions:
> - line 156:
> The assumption that $n=m$ in Section 3 can be relaxed straightforwardly to $|n-m| \leq 1$, but not beyond that.
> Indeed for our eigenvalue expansion result (Proposition 3.1) we needed to assume $A$ has distinct eigenvalues, which implies $|n-m| \leq 1$.
> We believe that an analogous behavior takes place in the general case $n \neq m$ (after changing the average eigenvalue of $S$ by a suitable, related, quantity) but we have not yet been able to prove it.
> - line 208: Indeed this is a typo, thank you.
> - line 225: Indeed, this will make the text easier to read, thank you.
>
> [MPP18] Panayotis Mertikopoulos, Christos Papadimitriou, and Georgios Piliouras. “Cycles in adversarial regularized learning”. In: Proceedings of the twenty-ninth annual ACM-SIAM symposium on discrete algorithms. SIAM. 2018, pp. 2703–2717.

---

> > ### Comment · Reviewer_oPMS · 2023-08-12
> >
> > Thanks for the explanations. The experimental section makes much more sense after reading your explanation. I would include something like this in the revision. You should emphasize very clearly to the reader that that the $m\ne n$ case is not addressed, as I was not the only reviewer who noticed that this was swept under the rug.
> >
> > I am inclined now to raise my rating to 7, but will monitor the responses of other reviewers before making a final decision.

---

### Official Review · Reviewer_KFbS · 2023-07-18

**Soundness:** 4 excellent
**Presentation:** 4 excellent
**Contribution:** 2 fair
**Rating:** 6
**Confidence:** 5

**Summary:**

This research investigates the local convergence properties of gradient dynamics in two-player zero-sum differentiable games towards Nash equilibria. Existing knowledge suggests that such dynamics converge locally when the symmetric part of the Jacobian at equilibrium, denoted by S, is positive definite (S ≻ 0), and divergence is likely when S equals zero (S = 0). The symmetric part S accounts for the potential function of the game.

The authors advance this understanding by demonstrating that gradient dynamics can also exhibit local convergence as soon as S is non-zero but fails to be strictly positive definite, which is referred to as partial curvature. This convergence is shown to occur provided the eigenvectors of the antisymmetric component of the Jacobian, denoted as A, occupy a general position relative to the nullspace of S. This result elucidates conditions under which convergence is guaranteed, thereby broadening the scenarios where gradient dynamics can be effectively employed in such games.

The paper further explores the rates of convergence in the case where the antisymmetric part dominates the symmetric part, represented mathematically as S ≪ A. It is proven that the convergence rates typically depend on the arithmetic mean of the eigenvalues of S, in contrast to the minimization problems analogy that suggests the dependence of rates on the smallest eigenvalue. This counterintuitive finding contributes to a more nuanced understanding of the behavior of gradient dynamics in these games.

To illustrate the theoretical findings, the problem of computing mixed Nash equilibria in continuous games is considered. The authors reveal that due to the effect of partial curvature, conic particle methods, which concurrently optimize over the weights and supports of mixed strategies, converge generically faster than their fixed-support counterparts. In the context of min-max games, this implies a strategic benefit in adding degrees of freedom exhibiting curvature, presenting yet another advantage of over-parameterization. This practical manifestation of their theoretical insights underscores the significant role of over-parameterization in enhancing the efficiency of Nash equilibria computations in such games.

**Strengths:**

The paper impressively contributes a significant degree of originality by extending the traditional notions of regularization and partial curvature in a fresh manner to elucidate the convergence of Gradient Descent Ascent (GDA) to a Nash Equilibrium in continuous differentiable games. The authors’ approach not only offers an intriguing new perspective but also opens up a novel theoretical pathway that future work may explore further.

Quality-wise, the paper exhibits a meticulous level of mathematical rigor in establishing the convergence conditions and rates. The well-structured proofs and clear theoretical arguments bolster the reliability of the results, reinforcing the significance of the paper’s main findings.

In terms of clarity, the authors deserve commendation for maintaining a high standard of exposition. They have done an excellent job in the paper's organization and presentation of complex concepts in a manner that is accessible and engaging. The lucid language, combined with logically organized sections, facilitates an intuitive understanding of the problem setup, the methodological developments, and the theoretical results.

Regarding significance, the findings of this paper hold substantial implications for the broader research community focused on game theory and optimization. By unveiling the factors that determine the local convergence of GDA in two-player zero-sum differentiable games, the paper paves the way for the development of more efficient and reliable methods for finding Nash equilibria. Moreover, the practical relevance of these findings is underscored by the authors' demonstration of the benefits of over-parameterization in mixed Nash equilibria computations.

Overall, the paper stands out for its innovative approach, rigorous methodology, clear exposition, and high impact on the field of continuous differentiable games. It makes a valuable addition to the literature by advancing our understanding of the roles of partial curvature and regularization in the context of gradient dynamics convergence to Nash equilibria.

**Weaknesses:**

Certainly, here is the markdown version:

1. **Clarity on Mirror Prox (MP), Exponential Gradient (EG) Methods, and Overparametrization:** The paper could enhance its clarity by offering more detailed explanations and motivations for the application of Mirror Prox and Exponential Gradient methods, particularly in the context of overparametrization. Clear elaboration on how overparametrization interacts with these methods and contributes to convergence could reinforce the paper's argument.

2. **Applicability to Non-Square Games:** It would be beneficial for the authors to clarify if their results extend to non-square games, i.e., when the dimensions of the two players' strategies are different ($n \neq m$). Currently, the paper does not explicitly address this case, which leaves a gap in the comprehension of the scope of the presented results.

3. **Practical Examples of Slight Curvature Setting:** The inclusion of practical examples or real-world scenarios where the slight curvature condition holds would significantly enhance the paper's impact. Such examples can illustrate the practical relevance of the theoretical findings and provide concrete contexts in which the paper's insights can be applied.

4. **Average Eigenvalues and Convex-Concave Settings:** The paper's discussion on the role of average eigenvalues of the symmetric part of the Jacobian matrix $S$ in determining convergence rates might appear unclear to some readers. It would be advantageous if the authors could elaborate on whether these results are most relevant in non-convex non-concave settings, or in convex-concave settings that are not strongly convex-concave. This would help readers better understand the implications of the paper's results in different game settings.

**Questions:**

Please answer my concerns at Weaknesses' section

---

> ### Author Rebuttal · Authors · 2023-08-09
>
> We thank the reviewer for their appreciation of the paper. We address each concern separately.
> 1. **Clarity on MP, EG, and Overparametrization:**
> The benefit of using extrapolated gradient methods such as MP or EG for last-iterate convergence is well-known, for a general min-max optimization context [LS19, Section 3 and references therein].
> For the problem of computing mixed Nash equilibria, it is shown in [WLZL21] that MP converges while its explicit analog (MDA) diverges [BP18], and in [WC22] that the method called "EG" in Section 4 converges while its explicit analog had not yet been analyzed.
> Therefore when writing Section 4 of the paper (on computing mixed Nash equilibria), we chose to focus directly on the methods that are known to converge.
> Nevertheless we agree that a reminder of this background information would be helpful.
> 	- Regarding overparametrization, our results suggest that it generically improves the local conditioning of min-max problems. Moreover, this improvement implies that it is not in fact necessary to resort to extrapolated gradient methods for convergence -- although they still have a faster rate than explicit methods.
>
> 2. **Applicability to Non-Square Games:**
> Our results apply to non-square games, with the exception of the convergence rate estimates when interaction dominates (Propositions 3.1-3.3 and Table 1).
> For those, it is straightforward to relax the assumption $n=m$ to $|n-m| \leq 1$, but not beyond that.
> We believe that an analogous behavior takes place in the general case $n \neq m$ (after changing the average eigenvalue of $S$ by a suitable, related, quantity) but we have not yet been able to prove it.
> We will add an explicit remark about this.
>
> 3. **Practical Examples of Slight Curvature Setting:**
> In addition to the example studied in Section 4, another interesting example is the Augmented Lagrangian method, which is commonly used for real-world constrained optimization problems. We will include a discussion of this setting in revision.
>
> 4. **Average Eigenvalues and Convex-Concave Settings:**
> The relation between average eigenvalues of the symmetric part of the Jacobian matrix $M=S+A$ and convergence rate, is agnostic to convexity-concavity.
> Indeed the relation goes via the algebraic quantity $\tilde{\mu}_M$, which determines the local convergence rate of gradient flow by a fully general dynamical system result.
> In fact our average-eigenvalue results of Section 3 do not require $S$ to be positive semi-definite,
> and so they also apply in non-convex non-concave settings.
> We chose not to emphasize this fact because we focus on convergence to local Nash equilibria throughout, which automatically implies local convexity-concavity, but we will add a brief remark on it.
>
> [BP18] James P. Bailey and Georgios Piliouras. “Multiplicative Weights Update in Zero-Sum Games”. In: Proceedings of the 2018 ACM Conference on Economics and Computation (2018).
>
> [LS19] Tengyuan Liang and James Stokes. “Interaction matters: A note on non-asymptotic local convergence of generative adversarial networks”. In: The 22nd International Conference on Artificial Intelligence and Statistics. PMLR. 2019, pp. 907–915.
>
> [WC22] Guillaume Wang and Lénaïc Chizat. “An Exponentially Converging Particle Method for the Mixed Nash Equilibrium of Continuous Games”. In: arXiv preprint arXiv:2211.01280 (2022).
>
> [WLZL21] Chen-Yu Wei, Chung-Wei Lee, Mengxiao Zhang, and Haipeng Luo. “Linear Last-iterate Convergence in Constrained Saddle-point Optimization”. In: International Conference on Learning Representations. 2021.

---

### Decision · Program_Chairs · 2023-09-21

**Decision:**

Accept (poster)

**Comment:**

All four reviewers are positive about this paper and think it should be accepted and I support their decisions. Please take into account the reviewers comments while preparing the final version.